# CTRL-ADAPTER: AN EFFICIENT AND VERSATILE FRAMEWORK FOR ADAPTING DIVERSE CONTROLS TO ANY DIFFUSION MODEL

**Han Lin**[*]   **Jaemin Cho**[*]   **Abhay Zala**   **Mohit Bansal**
UNC Chapel Hill
{hanlincs, jmincho, aszala, mbansal}@cs.unc.edu
https://ctrl-adapter.github.io

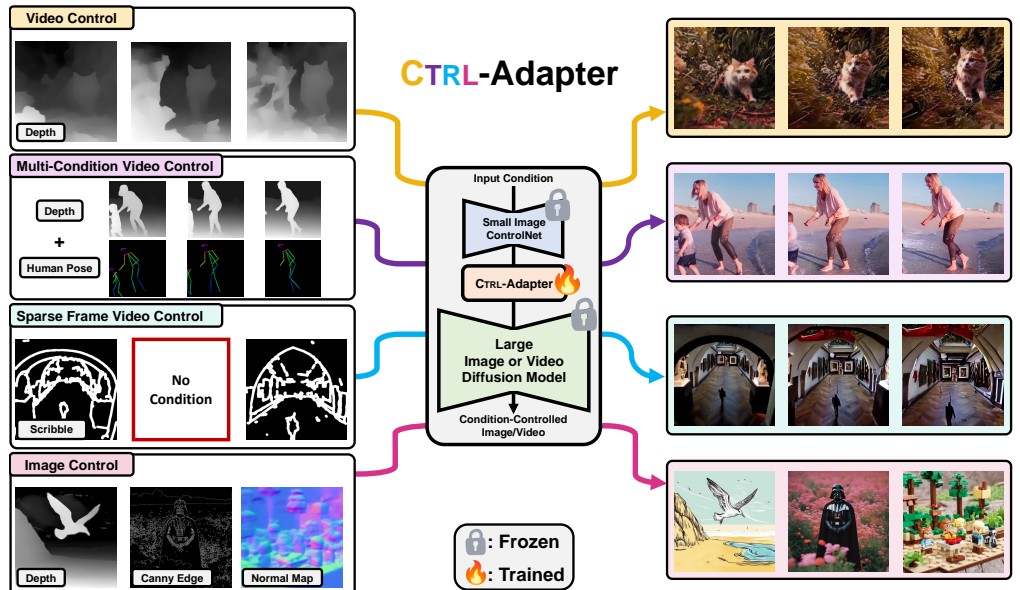

Figure 1: We propose **CTRL-Adapter**, an efficient and versatile framework for adding diverse controls to any diffusion model. CTRL-Adapter supports a variety of useful applications.

## ABSTRACT

ControlNets are widely used for adding spatial control to text-to-image diffusion models with different conditions, such as depth maps, scribbles/sketches, and human poses. However, when it comes to controllable video generation, ControlNets cannot be directly integrated into new backbones due to feature space mismatches, and training ControlNets for new backbones can be a significant burden for many users. Furthermore, applying ControlNets independently to different frames cannot effectively maintain object temporal consistency. To address these challenges, we introduce **CTRL-Adapter**, *an efficient and versatile framework that adds diverse controls to any image/video diffusion model through the adaptation of pretrained ControlNets.* CTRL-Adapter offers strong and diverse capabilities, including image and video control, sparse-frame video control, fine-grained patch-level multi-condition control (via an MoE router), zero-shot adaptation to unseen conditions, and supports a variety of downstream tasks beyond spatial control, including video editing, video style transfer, and text-guided motion control. With six diverse U-Net/DiT-based image/video diffusion models (SDXL, PixArt-$\alpha$, I2VGen-XL, SVD, Latte, Hotshot-XL), CTRL-Adapter matches the performance of pretrained ControlNets on COCO and achieves the state-of-the-art on DAVIS 2017 with significantly lower computation ($< 10$ GPU hours).

---

[*]Equal Contribution

# 1    INTRODUCTION

Recent diffusion models have achieved significant progress in generating high-fidelity images (Rombach et al., 2022; Podell et al., 2024; Saharia et al., 2022; Ramesh et al., 2022) and videos (Blattmann et al., 2023; Girdhar et al., 2023; Chen et al., 2024a; Lin et al., 2023; Long et al., 2024) from text descriptions. As it is often hard to describe every image/video detail only with text, there have been many works to control diffusion models in a more fine-grained manner by providing additional condition inputs such as bounding boxes (Li et al., 2023c; Yang et al., 2023), reference object images (Ruiz et al., 2023; Gal et al., 2023; Li et al., 2023a), and segmentation maps (Gafni et al., 2022; Avrahami et al., 2023; Zhang et al., 2023c). Among them, Zhang *et al.* (Zhang et al., 2023c) have released a variety of ControlNet checkpoints based on Stable Diffusion (Rombach et al., 2022) v1.5 (SDv1.5), and the user community has shared many ControlNets trained with different input conditions. Until now, ControlNet has become one of the most popular methods for controllable image generation.

However, there are challenges when using the existing pretrained image ControlNets for controllable video generation. First, pretrained ControlNet cannot be directly plugged into new backbone models, and the cost for training ControlNets for new backbone models is a big burden for many users due to high computational costs. For example, training a ControlNet for SDv1.5 takes 500-600 A100 GPU hours (Zhang et al., 2023b;a). Second, ControlNet was originally designed for controllable image generation; hence, applying pretrained image ControlNets directly to each video frame independently does not take the temporal consistency across frames into account.

To address this challenge, we design **CTRL-Adapter**, *a novel, flexible framework that enables the efficient reuse of pretrained ControlNets for diverse controls with any new image/video diffusion models, by adapting pretrained ControlNets (and improving temporal alignment for videos)*. We illustrate the overall capabilities of CTRL-Adapter framework in Fig. 1. As shown in Fig. 3 left (in Sec. 2), CTRL-Adapter trains adapter layers (Houlsby et al., 2019; Yi-Lin Sung, 2022) to map the features of a pretrained image ControlNet to a target image/video diffusion model, while keeping the parameters of the ControlNet and the backbone diffusion model frozen. As shown in Fig. 3 right, each CTRL-Adapter consists of four modules: spatial convolution, temporal convolution, spatial attention, and temporal attention. The temporal convolution/attention modules effectively fuse the ControlNet features into image/video diffusion models for better temporal consistency. Additionally, to ensure robust adaptation of ControlNets to backbone models of different noise scales and sparse frame control conditions, we propose skipping the visual latent variable from the ControlNet inputs. We also introduce inverse timestep sampling to effectively adapt ControlNets to new backbones equipped with continuous diffusion timestep samplers. For more accurate control beyond a single condition, we designed a novel and powerful Mixture-of-Experts (MoE) router, which allows fine-grained, patch-level composition of spatial feature maps from multiple control conditions via CTRL-Adapters.

As shown in Table 5, CTRL-Adapter allows many useful capabilities, including image control, video control, video control with sparse frames, multi-condition control, and compatibility with different backbone models, while previous methods only support a small subset of them (see details in Sec. 5). We demonstrate the effectiveness of CTRL-Adapter through extensive experiments and analyses. It exhibits strong performance when adapting ControlNets (pretrained with SDv1.5) to various video and image diffusion backbones, including image-to-video generation – I2VGen-XL (Zhang et al., 2023d) and Stable Video Diffusion (SVD) (Blattmann et al., 2023), text-to-video generation – Latte (Ma et al., 2024b) and Hotshot-XL (Mullan et al., 2023), and text-to-image generation – SDXL (Podell et al., 2024) and PixArt-$\alpha$ (Chen et al., 2024c). The ability of CTRL-Adapter to seamlessly adapt to DiT-based models such as Latte and PixArt-$\alpha$, which are structurally different from U-Net based ControlNets, demonstrates the flexibility of our framework design.

In Sec. 4.1 and Sec. 4.2, we first show that CTRL-Adapter matches the performance of a pretrained image ControlNet on COCO dataset (Lin et al., 2014) and outperforms previous methods in controllable video generation (achieving state-of-the-art performance on the DAVIS 2017 dataset (Pont-Tuset et al., 2017)) with significantly lower training costs (less than 10 GPU hours, see Fig. 2). Next, we demonstrate that CTRL-Adapter enables more accurate video generation with multiple conditions compared to a single condition. Our fine-grained patch-level MoE router consistently outperforms both the equal weights baseline and the global weights MoE router (Sec. 4.3). In addition, we show that skipping the visual latent variable from ControlNet inputs allows video control only with a few frames of (*i.e.*, sparse) conditions, eliminating the need for dense conditions across all frames

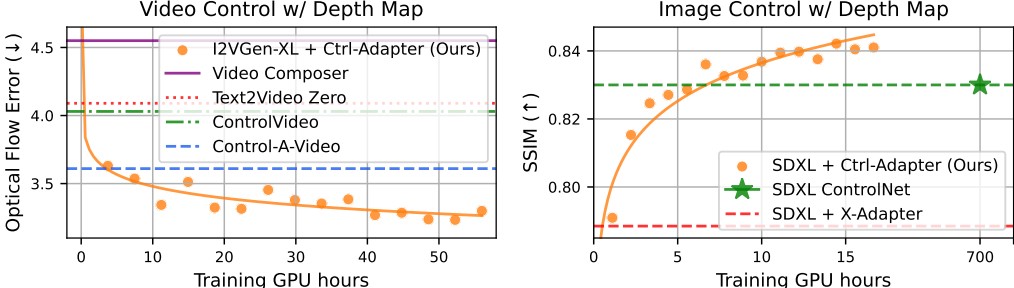

Figure 2: Training speed of CTRL-Adapter for video (left) and image (right) control with depth maps, measured on A100 80GB GPUs. For both video and image controls, CTRL-Adapter trained for 10 GPU hours outperforms strong baselines, including SDXL, which is trained for 700 GPU hours.

(Sec. 4.4). We also highlight zero-shot adaption – CTRL-Adapter trained with one condition can easily adapt to another ControlNet trained with a different condition (Sec. 4.5). In Sec. 4.6 and Sec. 4.7, we provide comparison of the training efficiency with ControlNet, and comparison between a unified CTRL-Adapter with individual CTRL-Adapters. Moreover, our CTRL-Adapter can be flexibly applied to a variety of downstream tasks beyond spatial control, including video editing, video style transfer, and text-guided object motion control (Appendix H.3). Lastly, we provide comprehensive ablations for CTRL-Adapter design choices (Appendix E), quantitative analysis including trade-off between visual quality and spatial control (Appendix F), and qualitative examples (Appendix G).

- We propose an efficient and versatile framework (CTRL-Adapter) that adds diverse controls to any image/video diffusion model through the adaptation of pretrained ControlNets, which matches the performance of training ControlNet from scratch with significantly lower training costs.
- We propose fine-grained, patch-level MoE routing to effectively compose ControlNet features, while previous works fuse the features of different control conditions only at the image level.
- CTRL-Adapter can be seamlessly adapted to both UNet-based and DiT-based image/video backbones (*e.g.*, PixArt-$\alpha$, Latte), diffusion models with continuous timestep samplers (via inverse noise sampling), different noise scales and sparse input frames (via latent-skipping).
- Through extensive experiments, we show that CTRL-Adapter matches the performance of pretrained ControlNets on both image and video generation backbones. In addition, Ctrl-Adapter can be flexibly applied to a variety of downstream tasks beyond spatial control, including video editing, video style transfer, and text-guided object motion control.

## 2 METHOD

### 2.1 PRELIMINARIES: LATENT DIFFUSION MODELS AND CONTROLNETS

**Latent Diffusion Models.** Many recent video generation works utilize latent diffusion models (LDMs) (Rombach et al., 2022) to learn the compact representations of videos. First, given a $F$-frame RGB video $\boldsymbol{x} \in \mathbb{R}^{F \times 3 \times H \times W}$, a video encoder (of a pretrained autoencoder) provides $C$-dimensional latent representation (*i.e.*, latents): $\boldsymbol{z} = \mathcal{E}(\boldsymbol{x}) \in \mathbb{R}^{F \times C \times H' \times W'}$, where height and width are spatially downsampled ($H' < H$ and $W' < W$). Next, in the forward process, a noise scheduler (*e.g.*, DDPM (Ho et al., 2020)) adds noise to the latents $\boldsymbol{z}$. Then, in the backward pass, a diffusion model $\mathcal{F}_{\boldsymbol{\theta}}(\boldsymbol{z}_t, t, \boldsymbol{c}_{\text{text/img}})$ learns to gradually denoise the latents, given a diffusion timestep $t$, and a text prompt $\boldsymbol{c}_{\text{text}}$ (*i.e.*, T2V) and/or an initial frame $\boldsymbol{c}_{\text{img}}$ (*i.e.*, I2V) if provided. The diffusion model is trained with objective:

$$\mathcal{L}_{\text{LDM}} = \mathbb{E}_{\boldsymbol{z}, \boldsymbol{\epsilon} \sim N(0, \boldsymbol{I}), t} \|\boldsymbol{\epsilon} - \boldsymbol{\epsilon}_{\boldsymbol{\theta}}(\boldsymbol{z}_t, t, \boldsymbol{c}_{\text{text/img}})\|_2^2$$

where $\boldsymbol{\epsilon}$ and $\boldsymbol{\epsilon}_{\boldsymbol{\theta}}$ represent the added noise to latents and the predicted noise by $\mathcal{F}_{\boldsymbol{\theta}}$ respectively. We apply the same objective for CTRL-Adapter training.

**ControlNets.** ControlNet (Zhang et al., 2023c) is designed to add spatial controls (*e.g.*, depth, sketch, segmentation maps, *etc.*) to image diffusion models. Specifically, given a pretrained backbone image diffusion model $\mathcal{F}_{\boldsymbol{\theta}}$ that consists of input/middle/output blocks, ControlNet has a similar

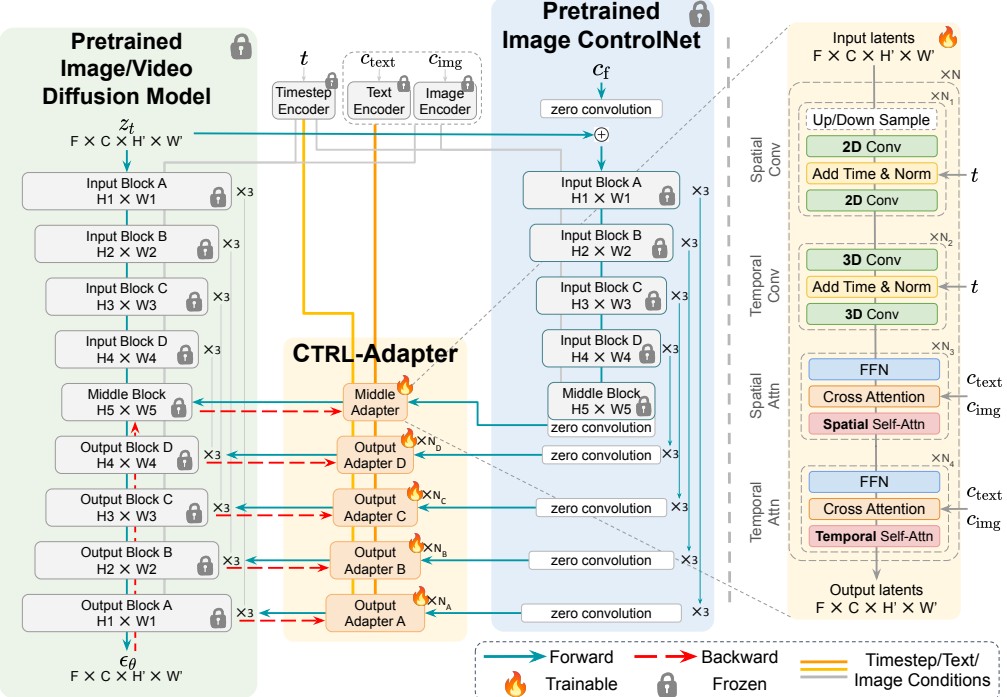

**Figure 3:** **Left:** CTRL-Adapter (colored orange) enables to reuse pretrained image ControlNets (colored blue) for new image/video diffusion models (colored green). **Right:** Architecture details of CTRL-Adapter. Temporal convolution and attention layers are skipped for image diffusion backbones. architecture $\mathcal{F}_{\theta'}$, where the input/middle blocks parameters of $\theta'$ are initialized from $\theta$, and the output blocks consist of $1 \times 1$ convolution layers initialized with zeros. ControlNet takes the diffusion timestep $t$, text prompt $c_{\text{text}}$, control image $c_f$ (e.g., depth map), and the noisy latents $z_t$ as inputs, and the output features are merged into the backbone model $\mathcal{F}_{\theta}$ for final image generation.

## 2.2 CTRL-ADAPTER

We introduce **CTRL-Adapter**, a novel framework that enables the efficient reuse of existing image ControlNets (SDv1.5) for spatial control with new diffusion models. *We mainly describe our method details in the video generation settings, since CTRL-Adapter can be flexibly adapted to image diffusion models by regarding images as single-frame videos.*

**Efficient adaptation of pretrained ControlNets.** As shown in Fig. 3 (left), we train an adapter module (colored orange) to map the middle/output blocks of a pretrained ControlNet (colored blue) to the corresponding middle/output blocks of the target video diffusion model (colored green). If the target backbone does not have the same number of output blocks, CTRL-Adapter maps the ControlNet features to the output block that handles the closest height and width of the latents. We keep all parameters in both the ControlNet and the target video diffusion model frozen. Therefore, training a CTRL-Adapter can be significantly more efficient than training a new video ControlNet.

**CTRL-Adapter architecture.** As shown in Fig. 3 (right), each block of CTRL-Adapter consists of four modules: spatial convolution, temporal convolution, spatial attention, and temporal attention. We set the values for $N_1, ..., N_4$ and $N$ as 1 by default. The temporal convolution and attention modules effectively fuse the ControlNet features to the video backbone models for better temporal consistency. Moreover, the spatial/temporal convolution modules incorporate the current denoising timestep $t$ by adding timestep embeddings into the adapter features and spatial/temporal attention modules incorporate the conditions (i.e., text prompt/initial frame) $c_{\text{text/img}}$ through cross-attention between image/text condition embeddings and the adapter features.. This design allows CTRL-Adapter to dynamically adjust its features according to different denoising stages and the objects generated. In addition, we skip the temporal convolution/attention modules when adapting to image diffusion models. See Appendix B.1 for architecture details of the four modules, and Appendix E for detailed ablation studies on the design choices of CTRL-Adapter.

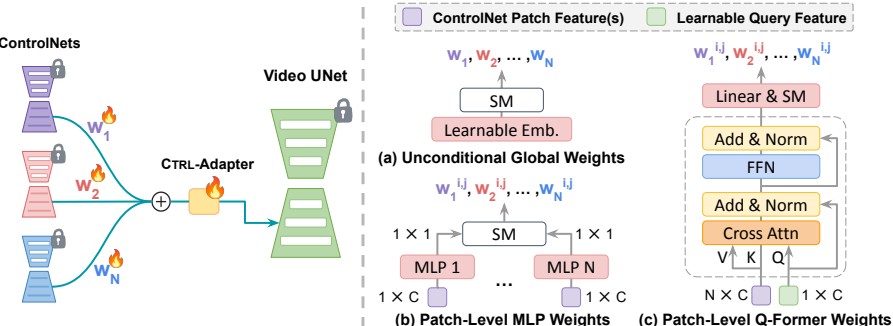

Figure 4: **Left:** Framework for multi-condition video generation by combining multiple ControlNets. $w_1, w_2, ..., w_N$ are the weights allocated to each ControlNet. **Right:** Three MoE router variants. (a) operates globally, while (b) and (c) operate on the fine-grained patch-level. $C$ and $N$ represent feature dimensions and number of ControlNet experts respectively. $w_k^{i,j}$ represents the router weights at position $(i, j)$ of the $k^{\text{th}}$ ControlNet 2D feature map. SM stands for Softmax.

**Adaptation to DiT-based image/video backbones.** Our CTRL-Adapter can also adapt U-Net based ControlNets to DiT-based image/video generation backbones. One important observation we made is that the spatial features encoded in the U-Net of ControlNets and the DiT blocks are structurally different (see Fig. 22). Specifically, the representation from U-Net blocks exhibits coarse-to-fine, hierarchical patterns (*e.g.*, earlier blocks output smaller size feature maps and control high-level information such as object presence, while later blocks output larger feature maps and control lower-level details like textures), while all DiT blocks handle the feature maps of same sizes. This indicates that mapping all middle/output blocks of ControlNet to DiT blocks might not be the optimal solution. Therefore, we choose to map the feature maps of the largest size in ControlNet (*i.e.*, block A) to the DiT blocks via CTRL-Adapters, which are followed by zero-convolutions for channel dimension matching. To improve computational efficiency for DiT-based video generation models (*i.e.*, Latte Ma et al. (2024b)), we only insert CTRL-Adapters into every other DiT block (*i.e.*, blocks 2, 4, 6..., 28, see (a) in Fig. 16). See Appendix E.2 for more discussion on CTRL-Adapter designs for DiT.

**Skipping the latent from ControlNet inputs: robust adaption to different noise scales & sparse frame conditions.** Although the original ControlNets take the latent $z_t$ as part of their inputs, we find that skipping $z_t$ from ControlNet inputs is effective for CTRL-Adapter in certain settings, as illustrated in Fig. 12. **(1) Different noise scales:** while SDv1.5 samples noise $\epsilon$ from $N(\mathbf{0}, \mathbf{I})$, some recent diffusion models Hoogeboom et al. (2023); Esser et al. (2024); Blattmann et al. (2023) sample noise $\epsilon$ of much bigger scale (*e.g.* SVD Blattmann et al. (2023) sample noise from $\sigma * N(\mathbf{0}, \mathbf{I})$, where $\sigma \sim \text{LogNormal}(0.7, 1.6)$; $\sigma \in [0, +\infty]$ and $\mathbb{E}[\sigma] = 7.24$). We find that adding larger-scale $z_t$ from the new backbone models to image conditions $c_f$ dilutes the $c_f$ and makes the ControlNet outputs less informative, whereas skipping $z_t$ enables the adaptation of such new backbone models. **(2) Sparse frame conditions:** when the image conditions are provided only for the subset of video frames (*i.e.*, $c_f = \emptyset$ for most frames $f$), ControlNet could rely on the information from $z_t$ and ignore $c_f$ during training. Skipping $z_t$ from ControlNet inputs also helps the CTRL-Adapter to more effectively handle such sparse frame conditions (see Table 10). With latent-skipping, the input to ControlNet $\mathcal{F}_{\boldsymbol{\theta}}$ becomes $c_f$ instead of $c_f + z_t$ (*i.e.*, ControlNet outputs $\mathcal{F}_{\boldsymbol{\theta}}(c_f)$ instead of $\mathcal{F}_{\boldsymbol{\theta}}(c_f + z_t)$).

**Inverse timestep sampling: robust adaptation to continuous diffusion timestep samplers.** While SDv1.5 samples discrete timesteps $t$ uniformly from $\{0, 1, ...1000\}$, some recent diffusion models Esser et al. (2024); Ma et al. (2024a); Rombach et al. (2021) sample timesteps from continuous distributions, *e.g.*, SVD Blattmann et al. (2023) samples timesteps from a LogNormal distribution. This gap between discrete and continuous distributions means that we cannot assign the same timestep $t$ to both the video diffusion model and the ControlNet. Therefore, we propose inverse timestep sampling, an algorithm that *creates a timestep mapping between the continuous and discrete time distributions* (see Algorithm 1 for PyTorch Ansel et al. (2024) code). The high-level idea of this algorithm is inspired by inverse transform sampling Estimation lemma (2010). Given the cumulative distribution functions (CDFs) of the continuous timestep distribution $F_{\text{cont.}}$ and the ControlNet timestep distribution $F_{\text{CNet}}$, we first uniformly sample a value $u$ between $[0, 1]$, and then returns the smallest timesteps $t_{\text{cont.}} \in [0, \infty] \subseteq \mathbb{R}, t_{\text{CNet}} \in \{0, 1, ..., 1000\} \subseteq \mathbb{N}$, such that $F_{\text{cont.}}(t_{\text{cont.}}) \geq u, F_{\text{CNet}}(t_{\text{CNet}}) \geq u$. This procedure naturally creates a mapping between two distributions. See Appendix B.2 for details.

Table 1: Evaluation of video generation with single control condition on DAVIS 2017 dataset. The best number in each column is **bolded**, and the second best is underscored.

| Method | Depth Map | | Canny Edge | |
|---|---|---|---|---|
| | FID ($\downarrow$) | Optical Flow Error ($\downarrow$) | FID ($\downarrow$) | Optical Flow Error ($\downarrow$) |
| Text2Video-Zero (Khachatryan et al., 2023) | 19.46 | 4.09 | 17.80 | 3.77 |
| ControlVideo (Zhang et al., 2024) | 27.84 | 4.03 | 25.58 | 3.73 |
| Control-A-Video (Chen et al., 2023) | 22.16 | 3.61 | 22.82 | 3.44 |
| VideoComposer (Wang et al., 2024) | 22.09 | 4.55 | - | - |
| *Hotshot-XL backbone* | | | | |
| SDXL ControlNet (von Platen et al., 2022) | 45.35 | 4.21 | 25.40 | 4.43 |
| SDv1.5 ControlNet + CTRL-Adapter (Ours) | 14.63 | 3.94 | 20.83 | 4.15 |
| *Latte backbone (DiT-Based)* | | | | |
| SDv1.5 ControlNet + CTRL-Adapter (Ours) | 16.92 | 3.98 | 17.87 | 2.73 |
| *I2VGen-XL backbone* | | | | |
| SDv1.5 ControlNet + CTRL-Adapter (Ours) | 7.43 | 3.20 | 6.42 | 3.37 |
| *SVD backbone* | | | | |
| SVD Temporal ControlNet (Rowles, 2023) | 4.91 | 4.84 | - | - |
| SDv1.5 ControlNet + CTRL-Adapter (Ours) | **3.82** | **2.96** | **3.96** | **2.39** |

Table 2: Evaluation of image generation with single control condition on COCO val2017 split. The best number in each column is **bolded**, and the second best is underscored.

| Method | Depth Map | | | Canny Edge | | Soft Edge / HED | |
|---|---|---|---|---|---|---|---|
| | FID ($\downarrow$) | MSE ($\downarrow$) | SSIM ($\uparrow$) | FID ($\downarrow$) | SSIM ($\uparrow$) | FID ($\downarrow$) | SSIM ($\uparrow$) |
| *SDv1.4 or v1.5 backbone* | | | | | | | |
| SDv1.5 ControlNet (Zhang et al., 2023c) | 21.25 | 87.57 | - | 18.90 | 0.4828 | 26.59 | 0.4719 |
| T2I-Adapter (Mou et al., 2023) | 21.35 | 89.82 | - | 18.98 | 0.4422 | - | - |
| GLIGEN (Li et al., 2023c) | 21.46 | 88.22 | - | 24.74 | 0.4226 | 28.57 | 0.4015 |
| Uni-ControlNet (Zhao et al., 2024) | 21.20 | 91.05 | - | 17.79 | 0.4911 | 17.86 | 0.5197 |
| *SDXL backbone* | | | | | | | |
| SDXL ControlNet (von Platen et al., 2022) | **17.91** | 86.95 | 0.8363 | **17.21** | 0.4458 | - | - |
| SDv1.5 ControlNet + X-Adapter (Ran et al., 2024) | 20.71 | 90.08 | 0.7885 | 19.71 | 0.3002 | - | - |
| SDv1.5 ControlNet + CTRL-Adapter (Ours) | 19.26 | 87.54 | **0.8534** | 21.04 | 0.5806 | 18.08 | 0.6454 |
| *PixArt-$\alpha$ backbone (DiT-Based)* | | | | | | | |
| PixArt-$\delta$ ControlNet (Chen et al., 2024b) | - | - | - | - | - | 20.41 | **0.6938** |
| SDv1.5 ControlNet + CTRL-Adapter (Ours) | 22.54 | **84.78** | 0.8496 | 18.75 | **0.6359** | **17.52** | 0.6812 |

## 2.3 MULTI-CONDITION GENERATION VIA CTRL-ADAPTER COMPOSITION

Multi-ControlNet (Zhang et al., 2023c) is proposed for spatial control beyond a single condition. However, this method naively combines different conditions with equal weights during inference time without training. For more effective control composition, we experiment with the following. For each variant, we randomly select $K \in \{1, 2, 3, 4\}$ control conditions to train CTRL-Adapter and the router in each training step. Comparisons of different variants are discussed in Sec. 4.3 and Appendix E.4.

- **(a) Unconditional Global Weights:** This variant replaces these fixed weights with unconditional global learnable weights via a lightweight MoE (Shazeer et al., 2017) router. Specifically, this router is a simple linear layer with an input dimension of 1 and an output dimension equal to the number of conditions. Each output dimension represents the weight allocated to a certain condition, with the constraint that the sum of all weights equals 1.
- **(b) Patch-Level MLP Weights:** This variant processes the patch-level features of each ControlNet into a scalar value independently, then uses softmax to assign ControlNet weights for each patch. Specifically, the router computes the weighted average of the feature maps from different control conditions associated with the same patch, with the weights learned by a 3-layer MLP. The input dimension is equal to the feature map embedding dimension, and the output dimension is equal to 1, producing a scalar value that represents the weight of each patch's feature map.
- **(c) Patch-Level Q-Former Weights:** This variant takes in all $N$ ControlNet features associated with a patch, using an architecture design inspired by Q-Former (Li et al., 2023b) to output expert weights. Compared with variant (b), where the weights allocated to each patch are independent, variant (c) operates in a more holistic way that allows the router to see all patches of a feature map and then determine the weights allocated to each patch.

## 3 EXPERIMENTAL SETUP

**ControlNets and Target Diffusion Models.** We use ControlNets trained with SD v1.5. For target diffusion models, we experiment with two I2V models – I2VGen-XL (Zhang et al., 2023d) and SVD (Blattmann et al., 2023), two T2V models – Latte (Ma et al., 2024b) and Hotshot-XL (Mullan et al., 2023), and two T2I models – SDXL (Podell et al., 2024) and PixArt-$\alpha$ (Chen et al., 2024c).

**Training and Evaluation Datasets.** We use 200K videos sampled from Panda-70M training set (Chen et al., 2024d) and 300K images from the LAION POP (Schuhmann & Bevan, 2023) dataset for video and image CTRL-Adapters training respectively. During training, we extract various control conditions (*e.g.*, depth map) on-the-fly to simplify the data-preparation process. Following previous works (Hu & Xu, 2023; Zhang et al., 2024), we evaluate video CTRL-Adapters on DAVIS 2017 (Pont-Tuset et al., 2017), and image CTRL-Adapters on COCO `val2017` split (Lin et al., 2014). Detailed training and inference setups for the experiments are provided in Appendix C and Appendix D.

**Evaluation Metrics.** We perform evaluation on two folds: **visual quality** and **spatial control**. Following previous works (Qin et al., 2023; Hu & Xu, 2023), we use FID (Heusel et al., 2017) to measure the visual quality of generated images/videos. For video datasets, following previous works (Hu & Xu, 2023; Li et al., 2024), we report the L2 distance between the optical flow error (Ranjan & Black, 2017) between the conditions extracted from input and generated videos. For image datasets, following Uni-ControlNet (Zhao et al., 2024), we report the Structural Similarity (SSIM) (Wang et al., 2004) and mean squared error (MSE) between generated images and ground truth images.

## 4 RESULTS AND ANALYSIS

### 4.1 VIDEO GENERATION WITH SINGLE CONDITION

We compare SDv1.5 ControlNet + CTRL-Adapter built on Hotshot-XL, I2VGen-XL, SVD, and Latte with video control methods including Text2Video-Zero (Khachatryan et al., 2023), Control-A-Video (Chen et al., 2023), ControlVideo (Zhang et al., 2024), and VideoComposer (Wang et al., 2024). As the spatial layers of Hotshot-XL are initialized with SDXL and remain frozen, the SDXL ControlNets are directly compatible with Hotshot-XL, so we include Hotshot-XL + SDXL ControlNet as a baseline. We also experiment with a temporal ControlNet (Rowles, 2023) trained with SVD.

Table 1 shows that in both depth map and canny edge input conditions, CTRL-Adapters on I2VGen-XL and SVD outperforms all previous strong video control methods in visual quality (FID) and spatial control (optical flow error) metrics. Note that it takes $< 10$ GPU hours for CTRL-Adapter to outperform the baselines (see Fig. 2). In Appendix H.1, we visualize the comparison between CTRL-Adapter and other baselines. We study visual quality-spatial control trade-off in Appendix F.1.

### 4.2 IMAGE GENERATION WITH SINGLE CONDITION

We compare SDv1.5 ControlNet + CTRL-Adapter with controllable image generation methods that use SDv1.4, SDv1.5, SDXL, and PixArt-$\alpha$ as backbones, including pre-trained SDv1.5/SDXL ControlNets (Zhang et al., 2023c; von Platen et al., 2022), T2I-Adapter (Mou et al., 2023), GLIGEN (Li et al., 2023c), Uni-ControlNet (Zhao et al., 2024), X-Adapter (Ran et al., 2024), and PixArt-$\delta$ ControlNet (Chen et al., 2024b).

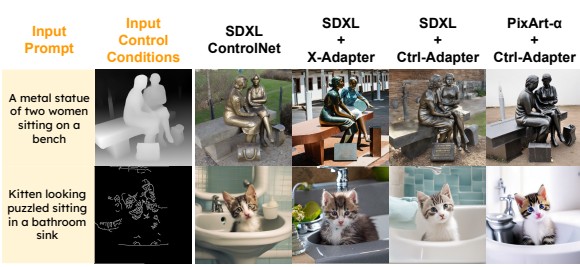

Figure 5: Images generated with single condition.

As shown in Table 2, CTRL-Adapter outperforms baselines with SDv1.4/v1.5 backbones in almost all metrics. When compared to the baselines with SDXL backbones, CTRL-Adapter outperforms X-Adapter in most metrics, and matches (in FID/MSE with depth map inputs) or outperforms SDXL ControlNet (in SSIM with depth map and canny edge inputs). Note that SDXL ControlNet was trained for much longer than CTRL-Adapter (700 *vs.* 44 A100 GPU hours) and it takes less than 10 GPU hours for CTRL-Adapter to outperform the SDXL depth ControlNet in SSIM (see Fig. 2). In addition, when applied to DiT-based backbone (*i.e.*, PixArt-$\alpha$), CTRL-Adapter achieves good improvement

Table 3: Comparison of different weighting methods (see Fig. 4 right part for details) for multi-condition video generation. The control sources are abbreviated as D (depth map), C (canny edge), N (surface normal), S (softedge), Seg (semantic segmentation map), L (line art), and P (human pose).

| | D+C | | D+P | | D+C+N+S | | D+C+N+S+Seg+L+P | |
|---|---|---|---|---|---|---|---|---|
| | FID ($\downarrow$) | Flow Error ($\downarrow$) | FID ($\downarrow$) | Flow Error ($\downarrow$) | FID ($\downarrow$) | Flow Error ($\downarrow$) | FID ($\downarrow$) | Flow Error ($\downarrow$) |
| Baseline: Equal Weights | 8.50 | 2.84 | 11.32 | 3.48 | 8.75 | 2.40 | 9.48 | 2.93 |
| (a) Unconditional Global Weights | 9.14 | 2.89 | 10.98 | 3.32 | 8.39 | 2.36 | 8.18 | 2.48 |
| (b) Patch-Level MLP Weights | 8.40 | **2.34** | 9.37 | **3.17** | 7.87 | **2.11** | 8.26 | **2.00** |
| (c) Patch-Level Q-Former Weights | **7.54** | 2.39 | **9.22** | 3.22 | **7.72** | 2.31 | **8.00** | 2.08 |

in FID (17.52 ours *vs.* 20.41 PixArt-$\delta$ ControlNet on soft edge) and competitive SSIM score. In Fig. 5, we visualize the comparison between CTRL-Adapter and other image control baselines. See Appendix H.2 for more visualizations.

## 4.3 VIDEO GENERATION WITH MULTIPLE CONTROL CONDITIONS

As described in Sec. 2.3, users can achieve multi-source control by simply combining the control features of multiple ControlNets via our CTRL-Adapter. Table 3 shows the result in two folds: firstly, patch-level MoE routers (*i.e.*, variants b and c in Fig. 4) consistently outperforms the equal weights baseline as well as the unconditional global weights (*i.e.*, variant a in Fig. 4), which proves the effectiveness of patch-level fine-grained control composition. Secondly, as shown in (b) and (c), control with more conditions almost always yields better spatial control and visual quality than control with a single condition. Fig. 28 and Fig. 29 show that multi-condition composition provides more accurate control compared to a single condition. Table 9 extends (a) by conditioning on image/text/timestep embeddings.

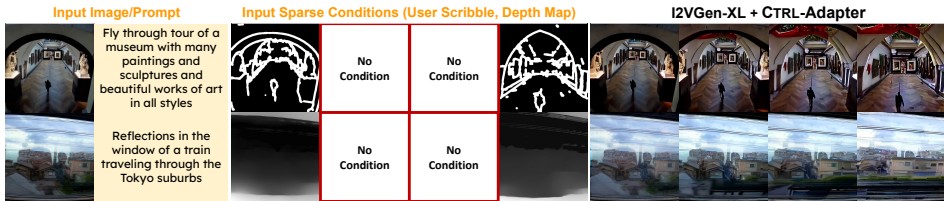

Figure 6: Video generation from sparse frame conditions with CTRL-Adapter on I2VGen-XL (which generates 16 frames in total). We only provide controls for the 1st, 6th, 11th, and 16th frames.

## 4.4 VIDEO GENERATION WITH SPARSE FRAMES AS CONTROL CONDITION

We experiment CTRL-Adapter with providing sparse frame conditions using I2VGen-XL as backbone. During each training step, we first randomly select an integer $k \in \{1, ..., N\}$, where $N$ is equal to the total number of output frames (*e.g.*, $N = 16$ for I2VGen-XL). Next, we randomly select $k$ key frames from $N$ total frames. We then extract these key frames' depth maps and user scribbles as control conditions. we do not give the latents $z$ and only give the $k$ frames to ControlNet. In Fig. 6, we can see that I2VGen-XL with our CTRL-Adapter can correctly generate videos that follow the control conditions for the given 4 sparse key frames and make reasonable interpolations on the frames without conditions. In Appendix E.3, we show that skipping the latent from ControlNet inputs is important in improving the sparse control capability.

## 4.5 ZERO-SHOT GENERALIZATION ON UNSEEN CONDITIONS

ControlNet can be understood as an image feature extractor that maps different types of controls to the unified representation space of backbone generation models. This begs an interesting question: *"Does CTRL-Adapter learn general feature mapping from one (smaller) backbone to another (larger) backbone?"*. To answer this question, we experiment by directly plugging CTRL-Adapter to ControlNets that are **not seen during training**. In Fig. 7, we observe the CTRL-Adapter trained on depth maps can adapt to normal map and soft edge ControlNets in a zero-shot manner. Quantitative analysis of different training strategies based on such observation is illustrated in Sec. 4.7.

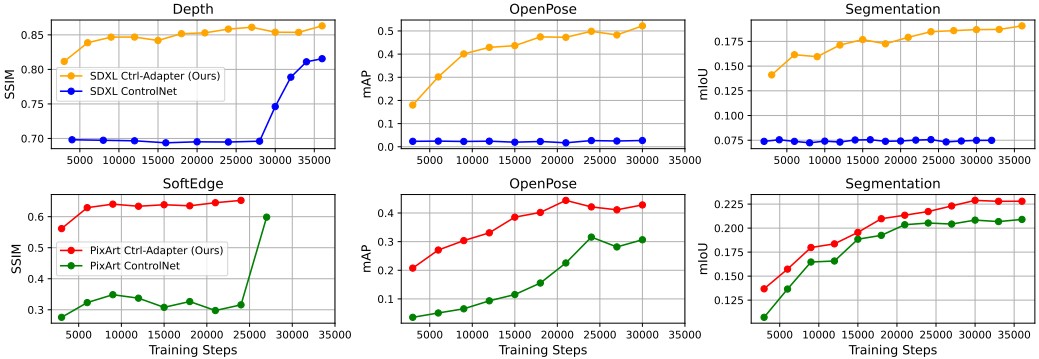

Figure 7: Zero-shot transfer of CTRL-Adapter trained only on depth maps to unseen conditions.

Figure 9: CTRL-Adapters are trained much faster than training ControlNets from scratch.

## 4.6 TRAINING EFFICIENCY

As a complement to Fig. 2, we present training efficiency comparison of our CTRL-Adapter and ControlNet trained from scratch under different control conditions, using the same hyperparameter settings for a fair comparison. Fig. 9 shows that our CTRL-Adapter achieves significantly faster training and higher final performance in different backbones and tasks. Additionally, the clock time *per training step* for our method is faster than that of ControlNet (CTRL-Adapter: 0.48s/step v.s. ControlNet: 0.60s/step for depth; CTRL-Adapter: 0.57s/step v.s. ControlNet: 0.68s/step for segmentation; CTRL-

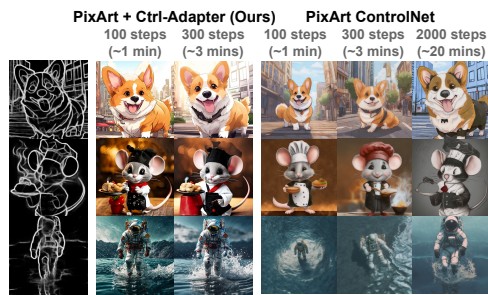

Figure 8: CTRL-Adapter achieves faster convergence than training ControlNet from scratch

Adapter: 0.82s/step v.s. ControlNet: 0.95s/step for openpose). Lastly, Fig. 8 shows that it takes only 3 mins for CTRL-Adapter to provide a model that accurately follows the edge condition, while it takes 20 mins for ControlNet.

## 4.7 A UNIFIED MULTI-TASK ADAPTER V.S. INDIVIDUAL TASK-SPECIFIC ADAPTERS

In our main results from the above sections, we train CTRL-Adapter for each control conditions. An interesting question to ask is: *can we have a single unified* CTRL-*Adapter that works for all control conditions?* We conduct an experiment comparing (1) training a single unified CTRL-Adapter (single adapter for all control conditions) v.s. (2) training an individual adapter for each control condition. Specifically, we use the same training settings with the only difference being that for individual CTRL-Adapter, we only extract one control condition (e.g. depth) from the input images, while for each training step of unified CTRL-Adapter, we randomly select one control condition from depth, canny, softedge, normal, openpose, lineart, segmentation with equal probability. For evaluation, we compare the generated image quality and control quality on 1000 randomly selected images from the COCO val2017 dataset. As shown in Table 4, unified CTRL-Adapter achieves comparable FID and SSIM scores. Such a result is consistent with the strong zero-shot transferability as observed in Sec. 4.5. Therefore, when a user has limited computational resource but still needs to work on multiple control conditions, we suggest training a unified CTRL-Adapter. Extended version of Table 4 is shown in Appendix F.4.

Table 4: Training a unified CTRL-Adapter with SDXL backbone achieves comparable FID/SSIM to training individual CTRL-Adapters; evaluated on 1K samples from COCO val2017.

| Method | Depth | | Canny | | Softedge | | Lineart | | Segmentation | | Normal | |
|---|---|---|---|---|---|---|---|---|---|---|---|---|
| | FID ($\downarrow$) | SSIM ($\uparrow$) | FID ($\downarrow$) | SSIM ($\uparrow$) | FID ($\downarrow$) | SSIM ($\uparrow$) | FID ($\downarrow$) | SSIM ($\uparrow$) | FID ($\downarrow$) | SSIM ($\uparrow$) | FID ($\downarrow$) | SSIM ($\uparrow$) |
| Individual CTRL-Adapters | **14.87** | **0.8398** | 14.00 | **0.5600** | **14.13** | **0.5123** | 11.26 | **0.5216** | 16.03 | **0.6732** | 14.94 | 0.8182 |
| Unified CTRL-Adapter | 15.13 | 0.8358 | **13.97** | 0.5454 | 14.25 | 0.4934 | 12.99 | 0.5117 | **15.68** | 0.6682 | 14.94 | **0.8143** |

## 5 RELATED WORKS: ADDING CONTROL TO DIFFUSION MODELS

There have been many works using different types of additional inputs to control the image/video diffusion models, such as bounding boxes (Li et al., 2023c; Yang et al., 2023), reference object image (Ruiz et al., 2023; Gal et al., 2023; Li et al., 2023a), segmentation map (Gafni et al., 2022; Avrahami et al., 2023; Zhang et al., 2023c), sketch (Zhang et al., 2023c), *etc.*, and combinations of multiple conditions (Kim et al., 2023; Qin et al., 2023; Zhao et al., 2024; Wang et al., 2024). As finetuning all the parameters of such image/video diffusion models is computationally expensive, several methods, such as ControlNet (Zhang et al., 2023c), have been proposed to add conditional control capability via parameter-efficient training (Zhang et al., 2023c; Ryu, 2022; Mou et al., 2023).

X-Adapter (Ran et al., 2024) learns an adapter module to reuse ControlNets pretrained with a smaller image diffusion model (*e.g.*, SDv1.5) for a bigger image diffusion model (*e.g.*, SDXL). While they focus solely on learning an adapter for image control, CTRL-Adapter features architectural designs (*e.g.*, temporal convolution/attention layers) for video generation as well. In addition, as shown in Fig. 10 (b), X-Adapter needs to be used with the source image diffusion model (SDv1.5) during both training and inference, whereas CTRL-Adapter does not require the smaller diffusion model for image or video generation, making it more memory and computationally efficient. SparseCtrl (Guo et al., 2023) guides a video diffusion model with conditional inputs of few frames (instead of full frames), to alleviate the cost of collecting video conditions. Since SparseCtrl involves augmenting ControlNet with an additional channel for frame masks, it requires training a new variant of ControlNet from scratch. In contrast, we

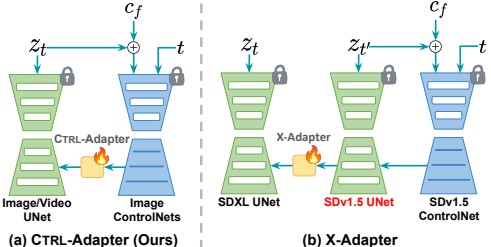

Figure 10: Comparison of giving different inputs to ControlNet, where $z_t, c_f$, and $t$ represent latents, input control features, and timesteps respectively. **(a):** Default CTRL-Adapter design. **(b):** X-Adapter (Ran et al., 2024) needs SDv1.5 U-Net as well as SDv1.5 ControlNet during training and inference, whereas CTRL-Adapter doesn't need SDv1.5 U-Net at all.

leverage existing image ControlNets more efficiently by propagating information through temporal layers in adapters and enabling sparse frame control via skipping the latents from ControlNet inputs.

Furthermore, compared with previous works that are specially designed for specific condition controls on a single modality (image (Zhang et al., 2023c; Qin et al., 2023) or video (Hu & Xu, 2023; Zhang et al., 2024)), our work presents a unified and versatile framework that supports diverse controls, including image control, video control, sparse frame control, with significantly lower computational costs by reusing pretrained ControlNets (outperforms strong baselines in less than 10 GPU hours, see Fig. 2). To the best of our knowledge, we are also the first work that extends multi-condition video control into fine-grained patch-level composition. Table 5 compares CTRL-Adapter with other relevant methods. See Appendix A.1 for extended related works.

## 6 CONCLUSION

We propose CTRL-Adapter, an efficient, powerful, and versatile framework that adds diverse controls to any image/video diffusion model. Training an CTRL-Adapter is significantly more efficient than training a ControlNet for a new backbone, and it can outperform or match strong baselines in visual quality and spatial control. CTRL-Adapter not only provides many useful capabilities including image/video control, sparse frame control, multi-condition control, and zero-shot adaption to unseen conditions, but also can be easily and flexibly integrated into a variety of downstream tasks.

## ACKNOWLEDGMENTS

We thank the reviewers for the thoughtful discussion and feedback. This work was supported by ARO W911NF2110220, DARPA MCS N66001-19-2-4031, DARPA KAIROS Grant FA8750-19-2-1004, NSF-AI Engage Institute DRL211263, ONR N00014-23-1-2356, and Accelerate Foundation Models Research program. The views, opinions, and/or findings contained in this article are those of the authors and not of the funding agency.

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

# Appendix

# A  BACKGROUND

## A.1  EXTENDED RELATED WORKS

**Text-to-video and image-to-video generation models.**  Generating videos from text descriptions or images (*e.g.*, initial video frames) based on deep learning and has increasingly gained much attention. Early works for this task (Li et al., 2017; 2019; Zhao et al., 2018; Skorokhodov et al., 2022) have commonly used variational autoencoders (VAEs) (Kingma & Welling, 2014) and generative adversarial networks (GANs) (Goodfellow et al., 2020), while most of recent video generation works are based on denoising diffusion models (Ho et al., 2020; Sohl-Dickstein et al., 2015). Powered by large-scale training, recent video diffusion models demonstrate impressive performance in generating highly realistic videos from text descriptions (He et al., 2022; Ho et al., 2022; Singer et al., 2023; Zhou et al., 2022; Khachatryan et al., 2023; Wang et al., 2023a; Yin et al., 2023; Wang et al., 2023b; Mullan et al., 2023; OpenAI, 2024; Gupta et al., 2023; Menapace et al., 2024) or initial video frames (*i.e.*, images) (Blattmann et al., 2023; Zhang et al., 2023d; Guo et al., 2024; Xing et al., 2023).

**Adding control to image/video diffusion models.**  While recent image/video diffusion models demonstrate impressive performance in generating highly realistic images/videos from text descriptions, it is hard to describe every detail of images/videos only with text or first frame image. Instead, there have been many works using different types of additional inputs to control the image/video diffusion models, such as bounding boxes (Li et al., 2023c; Yang et al., 2023), reference object image (Ruiz et al., 2023; Gal et al., 2023; Li et al., 2023a), segmentation map (Gafni et al., 2022; Avrahami et al., 2023; Zhang et al., 2023c), sketch (Zhang et al., 2023c), *etc*., and combinations of multiple conditions (Kim et al., 2023; Qin et al., 2023; Zhao et al., 2024; Wang et al., 2024). As finetuning all the parameters of such image/video diffusion models is computationally expensive, several methods, such as ControlNet (Zhang et al., 2023c), have been proposed to add conditional control capability via parameter-efficient training (Zhang et al., 2023c; Ryu, 2022; Mou et al., 2023). X-Adapter (Ran et al., 2024) learns an adapter module to reuse ControlNets pretrained with a smaller image diffusion model (*e.g.*, SDv1.5) for a bigger image diffusion model (*e.g.*, SDXL). While they focus solely on learning an adapter for image control, CTRL-Adapter features architectural designs (*e.g.*, temporal convolution/attention layers) for video generation as well. In addition, X-Adapter needs the smaller image diffusion model (SDv1.5) during training and inference, whereas CTRL-Adapter doesn't need the smaller diffusion model at all (for image/video generation), hence being more memory and computationally efficient (see Fig. 10 for details). SparseCtrl (Guo et al., 2023) guides a video diffusion model with conditional inputs of few frames (instead of full frames), to alleviate the cost of collecting video conditions. Since SparseCtrl involves augmenting ControlNet with an additional channel for frame masks, it requires training a new variant of ControlNet from scratch. In contrast, we leverage existing image ControlNets more efficiently by propagating information through temporal layers in adapters and enabling sparse frame control via skipping the latents from ControlNet inputs (see Sec. 2.2 for details). Furthermore, compared with previous works that are specially designed for specific condition controls on a single modality (image (Zhang et al., 2023c; Qin et al., 2023) or video (Hu & Xu, 2023; Zhang et al., 2024)), our work presents a unified and versatile framework that supports diverse controls, including image control, video control, sparse frame control, and multi-source control, with significantly lower computational costs by reusing pretrained ControlNets (*e.g.*, CTRL-Adapter outperforms baselines in less than 10 GPU hours). Table 5 summarizes the comparison of CTRL-Adapter with related works.

Table 5: Overview of the capabilities supported by controllable image/video generation methods.

| Method | Image Control | Video Control | Video Control w/ Sparse Frames | Multi-Condition Control | Compatible w/ Different Backbones |
|---|---|---|---|---|---|
| *Image Control Methods* | | | | | |
| ControlNet (Zhang et al., 2023c) | ✔ | ✘ | ✘ | ✘ | ✘ |
| Multi-ControlNet (Zhang et al., 2023c) | ✔ | ✘ | ✘ | ✔ | ✘ |
| T2I-Adapter (Mou et al., 2023) | ✔ | ✘ | ✘ | ✔ | ✘ |
| Uni-ControlNet (Zhao et al., 2024) | ✔ | ✘ | ✘ | ✔ | ✘ |
| X-Adapter (Ran et al., 2024) | ✔ | ✘ | ✘ | ✘ | ✔ |
| *Video Control Methods* | | | | | |
| ControlVideo (Zhang et al., 2024) | ✘ | ✔ | ✘ | ✘ | ✘ |
| VideoComposer (Wang et al., 2024) | ✘ | ✔ | ✘ | ✔ | ✘ |
| SparseCtrl (Guo et al., 2023) | ✘ | ✔ | ✔ | ✘ | ✘ |
| CTRL-Adapter (Ours) | ✔ | ✔ | ✔ | ✔ | ✔ |

Figure 11: Detailed architecture of CTRL-Adapter blocks.

## A.2 EXTENDED PRELIMINARIES: LDM AND CONTROLNET

**Latent Diffusion Models.** Many recent video generation works are based on latent diffusion models (LDMs) (Rombach et al., 2022), where a diffusion model learns the temporal dynamics of compact latent representations of videos. First, given a $F$-frame RGB video $\boldsymbol{x} \in \mathbb{R}^{F \times 3 \times H \times W}$, a video encoder (of a pretrained autoencoder) provides $C$-dimensional latent representation (*i.e.*, latents): $\boldsymbol{z} = \mathcal{E}(\boldsymbol{x}) \in \mathbb{R}^{F \times C \times H' \times W'}$, where height and width are spatially downsampled ($H' < H$ and $W' < W$). Next, in the forward process, a noise scheduler such as DDPM (Ho et al., 2020) gradually adds noise to the latents $\boldsymbol{z}$: $q(\boldsymbol{z}_t|\boldsymbol{z}_{t-1}) = N(\boldsymbol{z}_t; \sqrt{1-\beta_t}\boldsymbol{z}_{t-1}, \beta_t \boldsymbol{I})$, where $\beta_t \in (0,1)$ is the variance schedule with $t \in \{1, ..., T\}$. Then, in the backward pass, a diffusion model (usually a U-Net architecture) $\mathcal{F}_{\boldsymbol{\theta}}(\boldsymbol{z}_t, t, \boldsymbol{c}_{\text{text/img}})$ learns to gradually denoise the latents, given a diffusion timestep $t$, and a text prompt $\boldsymbol{c}_{\text{text}}$ (*i.e.*, T2V) or an initial frame $\boldsymbol{c}_{\text{img}}$ (*i.e.*, I2V) if provided. The diffusion model is trained with following objective: $\mathcal{L}_{\text{LDM}} = \mathbb{E}_{\boldsymbol{z}, \boldsymbol{\epsilon} \sim N(0, \boldsymbol{I}), t} \|\boldsymbol{\epsilon} - \boldsymbol{\epsilon}_{\boldsymbol{\theta}}(\boldsymbol{z}_t, t, \boldsymbol{c}_{\text{text/img}})\|_2^2$, where $\boldsymbol{\epsilon}$ and $\boldsymbol{\epsilon}_{\boldsymbol{\theta}}$ represent the added noise to latents and the predicted noise by $\mathcal{F}_{\boldsymbol{\theta}}$ respectively. We apply the same objective for CTRL-Adapter training.

**ControlNets.** ControlNet (Zhang et al., 2023c) is designed to add spatial controls to image diffusion models in the form of different guidance images (*e.g.*, depth, sketch, segmentation maps, *etc*.). Specifically, given a pretrained backbone image diffusion model $\mathcal{F}_{\boldsymbol{\theta}}$ that consists of input/middle/output blocks, ControlNet has a similar architecture $\mathcal{F}_{\boldsymbol{\theta}'}$, where the input/middle blocks parameters of $\boldsymbol{\theta}'$ are initialized from $\boldsymbol{\theta}$, and the output blocks consist of 1x1 convolution layers initialized with zeros. ControlNet takes the diffusion timestep $t$, text prompt $\boldsymbol{c}_{\text{text}}$, control image $\boldsymbol{c}_{\text{f}}$ (*e.g.*, depth image), and the noisy latents $\boldsymbol{z}_t$ as inputs, and provides the features that are merged into middle/output blocks of backbone image model $\mathcal{F}_{\boldsymbol{\theta}}$ to generate the final image. The authors of ControlNet have released a variety of ControlNet checkpoints based on Stable Diffusion (Rombach et al., 2022) v1.5 (SDv1.5) and the user community have also shared many ControlNets trained with different input conditions based on SDv1.5. However, these ControlNets cannot be used with more recently released bigger and stronger image/video diffusion models, such as SDXL (Podell et al., 2024) and I2VGen-XL (Zhang et al., 2023d). Moreover, the input/middle blocks of the ControlNet are in the same size with those of the diffusion backbones (*i.e.*, if the backbone model gets bigger, ControlNet also gets bigger). Due to this, it becomes increasingly difficult to train new ControlNets for each bigger and newer model that is released over time. To address this, we introduce CTRL-Adapter for efficient adaption of existing ControlNets for new diffusion models.

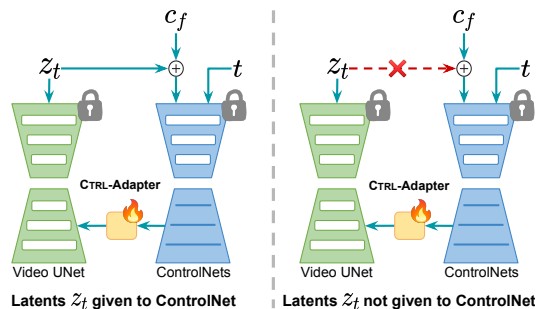

Figure 12: **Left (default):** latent $z_t$ is given to ControlNet. **Right:** latent $z_t$ not given to ControlNet.

# B    CTRL-ADAPTER METHOD AND ARCHITECTURE DETAILS

## B.1    CTRL-ADAPTER ARCHITECTURE DETAILS

In Fig. 11, we illustrate the detailed architecture of CTRL-Adapter blocks. See Fig. 3 for how the CTRL-Adapter blocks are used to adapt ControlNets to image/video diffusion models. Fig. 11 is an extended version of Fig. 3 (right) with more detailed visualizations, including skip connections, normalization layers in each module, and the linear projection layers (*i.e.*, FFN) in each spatial/temporal attention modules.

## B.2    INVERSE TIMESTEP SAMPLING: ROBUST ADAPTATION TO CONTINUOUS DIFFUSION TIMESTEP SAMPLERS

While SDv1.5 samples discrete timesteps $t$ uniformly from $\{0, 1, ...1000\}$, some recent diffusion models (Esser et al., 2024; Ma et al., 2024a; Rombach et al., 2021) sample timesteps from continuous distributions, *e.g.*, SVD (Blattmann et al., 2023) samples timesteps from a LogNormal distribution. This gap between discrete and continuous distributions means that we cannot assign the same timestep $t$ to both the video diffusion model and the ControlNet. Therefore, we propose inverse timestep sampling, an algorithm that *creates a timestep mapping between the continuous and discrete time distributions* The high-level idea of this algorithm is inspired by inverse transform sampling (Estimation lemma, 2010). Given the cumulative distribution functions (CDFs) of the continuous timestep distribution $F_{\text{cont.}}$ and the ControlNet timestep distribution $F_{\text{CNet}}$, we first uniformly sample a value $u$ between $[0, 1]$, and then returns the smallest timesteps $t_{\text{cont.}} \in [0, \infty] \subseteq \mathbb{R}, t_{\text{CNet}} \in \{0, 1, ..., 1000\} \subseteq \mathbb{N}$, such that $F_{\text{cont.}}(t_{\text{cont.}}) \geq u, F_{\text{CNet}}(t_{\text{CNet}}) \geq u$. This procedure naturally creates a mapping between two distributions. In Algorithm 1, we provide the PyTorch (Ansel et al., 2024) implementation of inverse timestep sampling, described in Sec. 2.2. In the example, inverse time stamping adapts to the SVD (Blattmann et al., 2023) backbone.

During each training step, the procedure for this algorithm can be summarized as follows:

- Sample a variable $u$ from Uniform$[0, 1]$. See line 19 in function `inverse_timestamp_sample`.

- Sample noise scale $\sigma_{\text{cont.}}$ via inverse transform sampling (Estimation lemma, 2010); *i.e.*, we derive the inverse cumulative density function of $\sigma_{\text{cont.}}$ and sample $\sigma_{\text{cont.}}$ by sampling $u$: $\sigma_{\text{cont.}} = F_{\text{cont.}}^{-1}(u)$. See function `sample_sigma` and line 21 in function `inverse_timestamp_sample`.

- Given a preconditioning function $g_{\text{cont.}}$ that maps noise scale to timestep (typically associated with the continuous-time noise sampler), we can compute $t_{\text{cont.}} = g_{\text{cont.}}(\sigma_{\text{cont.}})$. See function `sigma_to_timestep` and line 23 in `inverse_timestamp_sample`.

- Set the timesteps and noise scales for both ControlNet and our CTRL-Adapter as $t_{\text{CNet}} = \text{round}(1000u)$ and $\sigma_{\text{CNet}} = u$ respectively, where 1000 represents the denoising timestep range over which the ControlNet is trained. See line 25 in `inverse_timestamp_sample`.

During inference, we follow the similar sampling strategy, with the only change in the first step. Instead of uniformly sample a single value for $u$, we uniformly sample $k$ equidistant values for $u$

within $[0,1]$ and derive corresponding $t_{\text{cont./CNet}}$ and $\sigma_{\text{cont./CNet}}$ as inputs for denoising steps, where $k$ here is the number of denoising steps during inference.

---

**Algorithm 1** PyTorch Implementation for Inverse Timestep Sampling

```python
import torch

def sample_sigma(u, loc=0., scale=1.):
    """Draw a noise scale (sigma) following Karras et al. (2022)"""
    sigma_min, sigma_max, rho = 0.002, 700, 7 # values used in the paper
    min_inv_rho, max_inv_rho = sigma_min ** (1 / rho), sigma_max ** (1 / rho)
    sigma = (max_inv_rho + (1-u) * (min_inv_rho - max_inv_rho)) ** rho
    return sigma

def sigma_to_timestep(sigma):
    """Map noise scale to timestep. Here we use the function used in SVD."""
    timestep = 0.25 * sigma.log()
    return timestep

def inverse_timestamp_sample():
    """Sample noise scales and timesteps for ControlNet and diffusion models
    trained with continuous noise sampler. Here we use the setting used for SVD."""
    # 1) sample u from Uniform[0,1]
    u = torch.rand(1)
    # 2) calculate sigma_svd from pre-defined log-normal distribution
    sigma_svd = sample_sigma(u, loc=0.7, scale=1.6)
    # 3) calculate timestep_svd from sigma_svd via pre-defined mapping function
    timestep_svd = sigma_to_timestep(sigma_svd)
    # 4) calculate timestep and sigma for controlnet
    sigma_cnet, timestep_cnet = u, round(1000 * u)
    return sigma_svd, timestep_svd, sigma_cnet, timestep_cnet
```

---

## C  TRAINING AND INFERENCE DETAILS

**Model architectures.**    Detailed illustration of our CTRL-Adapter architecture has been provided across several parts of our paper, including Sec. 2, Appendix B.1, Appendix E, and Appendix F. In addition, for all the backbone models used in this paper, we kept all their parameters frozen and made no modifications.

**Training details.**    We use a learning rate of $5 \times e^{-5}$; AdamW (Loshchilov & Hutter, 2018) optimizer with values for $\beta_1$, $\beta_2$, $\epsilon$, and weight decay as 0.9, 0.999, $1 \times e^{-8}$, and $1 \times e^{-2}$ respectively. We set the max gradient norm as 1. All our experiments are trained on 4 A100 80GB GPUs with batch size of 1. *Please note that other than mixed-precision training with data type* `bfloat16`, *we didn't use any additional methods to speed up the training/inference clock time, or to save GPU memory.* To be more specific, **we didn't use any of the following methods**: xformers (Lefaudeux et al., 2022), gradient checkpointing, 8bit Adam optimizer, and DeepSpeed (Rasley et al., 2020). In addition, to make our framework easy to use directly from raw input images/videos, we extract all control condition images/frames **on-the-fly** during training. We train the image and video CTRL-Adapters for 80k and 40k steps respectively, which can be finished in 24 hours measured by training clock time. The fast convergence of our method is shown in Fig. 2.

**Inference details.**    All inference can be done on a single A6000 GPU with 48GB memory. During inference, we use the default hyper-parameters for each backbone model, including the number of frames to generate, the number of denoising steps, and classifier-free guidance scale, *etc.*.

**Safeguards.**    When we generate images during inference, we also activate the NSFW filter of the backbone models. This ensures that users are protected from unnecessary exposure to explicit or objectionable materials. For training, the datasets we used (Chen et al., 2024d; Schuhmann & Bevan, 2023) both filter out the image/video samples with harmful contents. For example, as stated in the "Risk mitigation" section of Panda70M paper, they used the internal automatic pipeline to filter out the video samples with harmful or violent language and texts that include drugs or hateful speech. They also use the NLTK framework to replace all people's names with "person". LAION-POP dataset is also created by filtering out samples based on the safety tags (using a customized trained NSFW classifier that they built).

## D   EXPERIMENTAL SETUP

### D.1   CONTROLNETS AND TARGET DIFFUSION MODELS

**ControlNets.**   We use ControlNets trained with SDv1.5.[1] SDv1.5 has the most number of publicly released ControlNets and has a much smaller training cost compared to recent image/video diffusion models. Note that unlike X-Adapter (Ran et al., 2024), CTRL-Adapter does not need to load the source diffusion model (SDv1.5) during training or inference (see (a) and (b) in Fig. 10 for model architecture comparison).

**Target diffusion models (where ControlNets are to be adapted).**   For video generation models, we experiment with two text-to-video generation models – Latte (Ma et al., 2024b) and Hotshot-XL (Mullan et al., 2023), and two image-to-video generation models – I2VGen-XL (Zhang et al., 2023d) and Stable Video Diffusion (SVD) (Blattmann et al., 2023). For image generation model, we experiment with PixArt-$\alpha$ (Chen et al., 2024c) and the base model in SDXL (Podell et al., 2024). For all models, we use their default settings during training and inference (*e.g.*, number of output frames, resolution, number of denoising steps, classifier-free guidance scale, *etc.*).

### D.2   TRAINING DATASETS FOR CTRL-ADAPTER

**Video datasets.**   For training CTRL-Adapter for video diffusion models, we download around 1.5M videos randomly sampled from the Panda-70M training set (Chen et al., 2024d). Following recent works (Blattmann et al., 2023; Dai et al., 2023), we filter out videos of static scenes by removing videos whose average optical flow (Farnebäck, 2003b; Bradski, 2000) magnitude is below a certain threshold. Concretely, we use the Gunnar Farneback's algorithm[2] (Farnebäck, 2003a) at 2FPS, calculate the averaged the optical flow for each video and re-scale it between 0 and 1, and filter out videos whose average optical flow error is below a threshold of 0.25. This process gives us a total of 200K remaining videos.

**Image datasets.**   For training CTRL-Adapter for image diffusion models, we use 300K images randomly sampled from LAION POP,[3] which is a subset of LAION 5B (Schuhmann et al., 2022) dataset and contains 600K images in total with aesthetic values of at least 0.5 and a minimum resolution of 768 pixels on the shortest side. As suggested by the authors, we use the image captions generated with CogVLM (Wang et al., 2023c).

### D.3   INPUT CONDITIONS

We extract various input conditions from the video and image datasets described above.

- **Depth map:**   As recommended in Midas[4] (Ranftl et al., 2020), we employ `dpt_swin2_large_384` for the best speed-performance trade-off.
- **Canny edge, surface normal, line art, softedge, and user sketching/scribbles:** Following ControlNet (Zhang et al., 2023c), we utilize the same annotator implemented in the `controlnet_aux`[5] library.
- **Semantic segmentation map:**   To obtain higher-quality segmentation maps than UPer-Net (Xiao et al., 2018) used in ControlNet, we employ SegFormer (Xie et al., 2021) `segformer-b5-finetuned-ade-640-640` finetuned on ADE20k dataset at 640×640 resolution.
- **Human pose:** We employ ViTPose (Xie et al., 2021) `ViTPose_huge_simple_coco` to improve both processing speed and estimation quality, compared to OpenPose (Cao et al., 2017) used in ControlNet.

---

[1]https://huggingface.co/lllyasviel/ControlNet
[2]https://docs.opencv.org/4.x/d4/dee/tutorial_optical_flow.html
[3]https://laion.ai/blog/laion-pop/
[4]https://github.com/isl-org/MiDaS
[5]https://github.com/huggingface/controlnet_aux

Table 6: Comparison of different architecture of CTRL-Adapter for image and video control, measured with visual quality (FID) and spatial control (MSE/optical flow error) metrics. The metrics are calculated from 1000 randomly selected COCO val2017 images and 150 videos from DAVIS 2017 dataset respectively. **Top:** image control on SDXL backbone. **Bottom:** video control on I2VGen-XL backbone. SC, TC, SA, and TA: Spatial Convolution, Temporal Convolution, Spatial Attention, and Temporal Attention. $^*N$: the number of blocks in CTRL-Adapter. We denote the default configurations (SC+SA for image / SC+TC+SA+TA for video) with bold font.

| SDXL | SC | SA | **SC+SA** | SC*2 | SA*2 | (SC+SA)*2 | SC*3 | SA*3 | (SC+SA)*3 |
|---|---|---|---|---|---|---|---|---|---|
| FID ($\downarrow$) | 12.15 | **11.52** | 11.57 | 11.74 | 12.36 | 10.91 | 11.66 | 12.11 | 12.07 |
| MSE ($\downarrow$) | 88.37 | 89.60 | **87.97** | 88.59 | 88.14 | 89.62 | 88.22 | 88.39 | 90.26 |

| I2VGen-XL | SC+SA | SA+TA | SC+TA | SC+TC | TC+TA | SA+TA*2 | SC+TA*2 | (TC+TA)*2 | **SC+TC+SA+TA** |
|---|---|---|---|---|---|---|---|---|---|
| FID ($\downarrow$) | 13.15 | 13.91 | 13.52 | **12.87** | 13.58 | 13.53 | 13.73 | 13.88 | 13.49 |
| Optical Flow Err. ($\downarrow$) | 3.27 | 3.33 | 3.30 | 3.29 | 3.26 | 3.37 | 3.25 | 3.23 | **3.22** |

## D.4 EVALUATION DATASETS

**Video datasets.** Following previous works (Hu & Xu, 2023; Zhang et al., 2024), we evaluate our video ControlNet adapters on DAVIS 2017 (Pont-Tuset et al., 2017), a public benchmark dataset also used in other controllable video generation works (Hu & Xu, 2023). We first combine all video sequences from `TrainVal`, `Test-Dev 2017` and `Test-Challenge 2017`. Then, we chunk each video into smaller clips, with the number of frames in each clip being the same as the default number of frames generated by each video backbone (*e.g.*, 8 frames for Hotshot-XL, 16 frames for I2VGen-XL, and 14 frames for SVD). This process results in a total of 1281 video clips of 8 frames, 697 clips of 14 frames, and 608 video clips of 16 frames.

**Image datasets.** We evaluate our image ControlNet adapters on COCO `val2017` split (Lin et al., 2014), which contains 5k images that cover diverse range of daily objects. We resize and center crop the images to 1024 by 1024 for SDXL evaluation.

## D.5 EVALUATION METRICS

**Visual quality.** Following previous works (Qin et al., 2023; Hu & Xu, 2023), we use Frechet Inception Distance (FID) (Heusel et al., 2017) to measure the distribution distance between our generated images/videos and input images/videos.[6]

**Spatial control.** For video datasets, following VideoControlNet (Hu & Xu, 2023), we report the L2 distance between the optical flow (Ranjan & Black, 2017) of the input video and the generated video (Optical Flow Error). For image datasets, following Uni-ControlNet (Zhao et al., 2024), we report the Structural Similarity (SSIM) (Wang et al., 2004)[7] and mean squared error (MSE)[8] between generated images and ground truth images.

# E VARIANTS OF CTRL-ADAPTER ARCHITECTURE DESIGN

## E.1 CTRL-ADAPTER DESIGN ABLATIONS

### E.1.1 DIFFERENT COMBINATIONS OF CTRL-ADAPTER COMPONENTS

As described in Sec. 2.2, each CTRL-Adapter module consists of four components: spatial convolution (SC), temporal convolution (TC), spatial attention (SA), and temporal attention (TA). We experiment

---

[6]To be consistent with the numbers reported in Uni-ControlNet (Zhao et al., 2024), we use pytorch-fid (`https://github.com/mseitzer/pytorch-fid`) in Table 2. For other results, we use clean-fid (Parmar et al., 2022) (`https://github.com/GaParmar/clean-fid`) which is more robust to aliasing artifacts.

[7]`https://scikit-image.org/docs/stable/auto_examples/transform/plot_ssim.html`

[8]`https://scikit-learn.org/stable/modules/classes.html#module-sklearn.metrics`

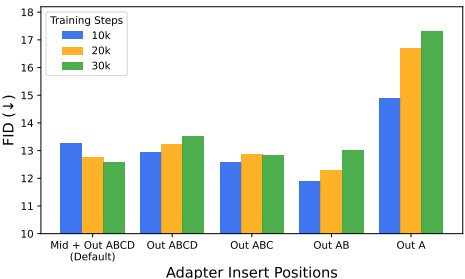 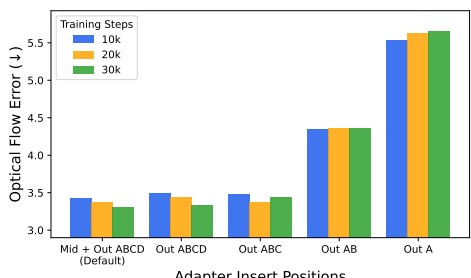

Figure 13: Comparison of inserting CTRL-Adapter to different U-Net blocks. 'Mid' represents the middle block, whereas 'Out ABCD' represents output blocks A, B, C, and D. The metrics are calculated from 150 videos from DAVIS 2017 dataset.

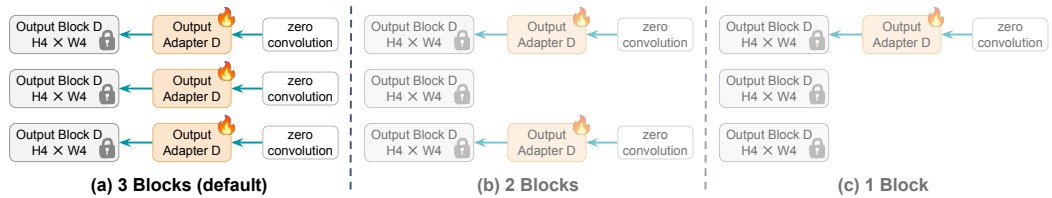

Figure 14: Comparison of inserting different numbers of CTRL-Adapters to the backbone diffusion U-Net's output blocks. We use output block D here for illustration. We insert three CTRL-Adapters to the output blocks of the same feature map size by default.

with different architecture combinations of the adapter components for image and video control, and show the results in Table 6. Compared to X-Adapter (Ran et al., 2024), which uses a stack of three spatial convolution modules (*i.e.*, ResNet (He et al., 2016) blocks) for adapters, and VideoComposer (Wang et al., 2024), which employs spatial convolution + temporal attention for spatiotemporal condition encoder, we explore a richer combination that enhances global understanding of spatial information through spatial attention and improves temporal ability via a combination of temporal convolution and temporal attention. For **image control** (Table 6 top), we find that the combining of SC+SA is more effective than stacking SC or SA layers only. Stacking SC+SA twice further improves the visual quality (FID) slightly but hurts the spatial control (MSE) as a tradeoff. Stacking SC+SA three times hurts the performance due to insufficient training. We use the single SC+SA layer for image CTRL-Adapter by default. For **video control** (Table 6 bottom), we find that SC+TC+SA+TA shows the best balance of visual quality (FID) and spatial control (optical flow error). Notably, we find that the combinations with both temporal layers, SC+TC+SA+TA and (TC+TA)*2, achieve the lowest optical flow error. We use SC+TC+SA+TA for video CTRL-Adapter by default.

### E.1.2 WHERE TO FUSE CTRL-ADAPTER OUTPUTS IN BACKBONE DIFFUSION

We compare the integration of CTRL-Adapter outputs at different positions of video diffusion backbone model. As illustrated in Fig. 3, we experiment with integrating CTRL-Adapter outputs to different positions of I2VGen-XL's U-Net: middle block, output block A, output block B, output block C, and output block D. Specifically, we compared our default design (Mid + Out ABCD) with four other variants (Out ABCD, Out ABC, Out AB, and Out A) that gradually remove CTRL-Adapters from the middle block and output blocks at positions from B to D. As shown in Fig. 13, removing the CTRL-Adapters from the middle block and the output block D does not lead to a noticeable increase in FID or optical flow error (*i.e.*, the performances of 'Mid+Out ABCD', 'Out ABCD', and 'Out ABC' are similar in both left and right plots). However, Fig. 13 (right) shows that removing CTRL-Adapters from block C causes a significant increase in optical flow error. Therefore, we recommend users retain our CTRL-Adapters in the mid and output blocks A/B/C to ensure good performance.

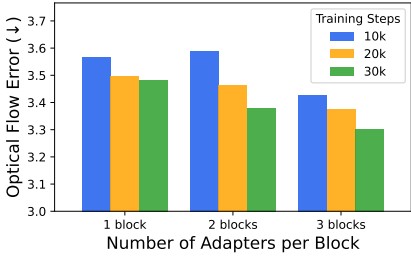

Figure 15: Comparison of inserting different numbers of CTRL-Adapters to each U-Net output block. The metrics are calculated from 150 videos from DAVIS 2017 dataset. We insert 3 CTRL-Adapters to each output block by default.

### E.1.3 NUMBER OF CTRL-ADAPTERS IN EACH OUTPUT BLOCK POSITION

As illustrated in Fig. 3, there are three output blocks for each feature map dimension in the video diffusion model (represented by $\times 3$ in each output block). Here, we conduct an ablation study by adding CTRL-Adapters to only one or two of the three output blocks of the same feature size. The motivation is that using fewer CTRL-Adapters can almost linearly decrease the number of trainable parameters, thereby reducing GPU memory usage during training. We visualize the architectural changes with output block D as an example in Fig. 14. We insert CTRL-Adapters for three blocks as our default setting. As observed in Fig. 15, reducing the number of CTRL-Adapters increases the optical flow error. Therefore, we recommend adding CTRL-Adapters to each output block to maintain optimal performance.

### E.2 ADAPTATION TO DiT-BASED BACKBONES

As illustrated in Sec. 2, we have observed that the spatial features encoded in the U-Net of ControlNets and the DiT blocks are structurally different (see Fig. 22 for visualization of such observation). Therefore, mapping all middle/output blocks of ControlNet to DiT blocks might not be the optimal solution. In Fig. 16, we implement three different strategies to insert CTRL-Adapters to the DiT blocks. Specifically, variant (a) inserts CTRL-Adapters interleavingly into the DiT blocks, while variant (b) and (c) insert CTRL-Adapters to the first 14 and the last 14 DiT blocks respectively. In Table 7, we perform quantitative analysis of these three variants on the DiT-based video generation model, Latte (Ma et al., 2024b), with soft edge as control condition. As we can see, inserting CTRL-Adapters interleavingly into the DiT blocks gives the best performance. This is consistent with our finding: since all DiT blocks encode global information of the generated objects, it is optimal to treat these blocks equally, rather than inserting CTRL-Adapters only at the beginning or end. Between locations A and B, we use location A as our default setting because its feature map size ($64 \times 64$) directly matches the features of the DiT blocks (also $64 \times 64$) without resizing.

After finalizing where to insert CTRL-Adapters, the next question is which block(s) of the ControlNet we should create CTRL-Adapters to map from. In Table 8, we implement several variants on the DiT-based image generation model, PixArt-$\alpha$ (Chen et al., 2024c), including mapping from the block(s) at location A, location B, location C, and location D, respectively (see Fig. 3 for the definitions of these locations). As we can see, mapping from location A or location B gives the best performance. Again, this is consistent with our findings in Fig. 22, since feature maps at locations C and D are too coarse to be informative. Moreover, we implemented two additional variants: (1) combining the ControlNet features from locations A and B (*i.e.*, Output Blocks A+B), and (2) mapping more blocks from the same location (*i.e.*, the second and third columns in Table 8). However, neither of these approaches provides sufficient gain compared to mapping a single block from location A or B. Therefore, we use mapping one block from location A as our default setting in our main paper.

[Han: add rebuttal table]

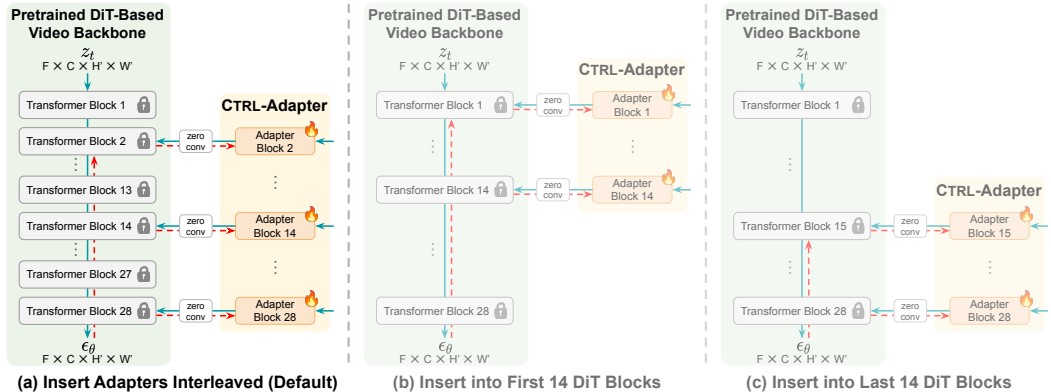

Figure 16: Visualization of different routing methods for combining multiple ControlNet outputs. We use (a) as our default setting, and show the settings (b) and (c) as ablations.

Table 7: Ablation of inserting CTRL-Adapters to different DiT blocks in Latte (Ma et al., 2024b). Visualization of the three architecture variants (interleaved, first half, and second half) are shown in Fig. 16. We use soft edge as control condition here for evaluation.

| (a) Interleave (default) | | (b) First Half | | (c) Second Half | |
|---|---|---|---|---|---|
| FID ($\downarrow$) | Optical Flow Error ($\downarrow$) | FID ($\downarrow$) | Optical Flow Error ($\downarrow$) | FID ($\downarrow$) | Optical Flow Error ($\downarrow$) |
| **18.32** | **2.98** | 19.66 | 3.09 | 23.18 | 3.31 |

### E.3 SKIPPING LATENT FROM CONTROLNET INPUTS

We find skipping the latent $z$ from ControlNet inputs can help CTRL-Adapter to more robustly handle (1) adaption to the backbone with noise scales different from SDv1.5, such as SVD and (2) video control with sparse frame conditions. For the first scenario, we can see from Table 10 that skipping latents in SVD leads to better visual quality (lower FID), but slightly worse spatial control (higher optical flow error). This is reasonable since skipping the noisy latents can avoid introducing large noise into the ControlNet, but it also risks losing information encoded in the latents. For the second scenario, skipping latents results in both better visual quality and better spatial control, as adding dense noisy latents can make the sparse control conditions less informative.

### E.4 DIFFERENT WEIGHING MODULES FOR MULTI-CONDITION GENERATION

For multi-condition generation described in Sec. 2.3, in addition to the simple unconditional global weights, we also experimented with learning a router module that takes additional inputs such as diffusion time steps and image/text embeddings and outputs weights for different ControlNets. Specifically, we introduce three variants based on (a.1) unconditional global weights, which are (a.2) MLP router - taking timestep as inputs; (a.3) MLP router - taking image/text embedding as inputs; and (a.4) MLP router - taking timestep and image/text embedding as inputs. The MoE router in these variants are constructed as a 3-layer MLP. We illustrate the five methods in Fig. 17.

Table 9 show that all four global weighting schemes for fusing different ControlNet outputs perform effectively, and no specific method outperforms other methods with significant margins in all settings. With no surprise, patch-level MoE router performs consistently better than global MoE router in all control settings. Testing the effectiveness of incorporating text/image/timestep embeddings to patch-level MoE routers are left for future work.

Table 8: Ablation study of mapping ControlNet features from different locations, and mapping different number of blocks from the same location to the DiT blocks. The best numbers in each row are **bolded**, and the best numbers in each column are underscored.

| Insert Locations | 1 Block (default) | | 2 Blocks | | 3 Blocks | |
|---|---|---|---|---|---|---|
| | FID ($\downarrow$) | SSIM ($\downarrow$) | FID ($\downarrow$) | SSIM ($\downarrow$) | FID ($\downarrow$) | SSIM ($\downarrow$) |
| Output Block A (default) | **17.90** | **0.6802** | 19.08 | 0.6971 | 19.28 | 0.6855 |
| Output Block B | **18.23** | 0.6712 | 18.61 | 0.6720 | 21.47 | **0.6549** |
| Output Blocks A+B | 17.52 | 0.6812 | - | - | - | - |
| Output Block C | 22.22 | 0.5273 | - | - | - | - |
| Output Block D | 34.16 | 0.3506 | - | - | - | - |

Table 9: Comparison of global weighting methods for multi-condition video generation (see Fig. 17 for visualization of the additional weighting methods (a.2, a.3, and a.4) developed based on (a.1) unconditional global weights). The control sources are abbreviated as D (depth map), C (canny edge), N (surface normal), S (softedge), Seg (semantic segmentation map), L (line art), and P (human pose).

| | D+C | | D+P | | D+C+N+S | | D+C+N+S+Seg+L+P | |
|---|---|---|---|---|---|---|---|---|
| | FID ($\downarrow$) | Flow Error ($\downarrow$) | FID ($\downarrow$) | Flow Error ($\downarrow$) | FID ($\downarrow$) | Flow Error ($\downarrow$) | FID ($\downarrow$) | Flow Error ($\downarrow$) |
| *Baseline* | | | | | | | | |
| Equal Weights | 8.50 | 2.84 | 11.32 | 3.48 | 8.75 | 2.40 | 9.48 | 2.93 |
| *Global MoE Router* | | | | | | | | |
| (a.1) Unconditional Global Weights | 9.14 | 2.89 | 10.98 | 3.32 | 8.39 | 2.36 | 8.18 | 2.48 |
| (a.2) Timestep Emb. Weights | 9.41 | 3.51 | 11.13 | 3.35 | 9.51 | 2.78 | 8.17 | 2.45 |
| (a.3) Text/Image Emb. Weights | 8.73 | 3.16 | 11.35 | 3.37 | 7.91 | 2.76 | 8.83 | 2.48 |
| (a.4) Timestep + Text/Image Emb. Weights | 8.64 | 3.31 | 10.69 | 3.43 | 8.09 | 2.69 | 8.51 | 2.43 |
| *Patch-Level MoE Router* | | | | | | | | |
| (b) MLP Weights | 8.40 | **2.34** | 9.37 | **3.17** | 7.87 | **2.11** | 8.26 | **2.00** |
| (c) Q-Former Weights | **7.54** | 2.39 | **9.22** | 3.22 | **7.72** | 2.31 | **8.00** | 2.08 |

Table 10: Skipping latent from ControlNet inputs helps CTRL-Adapter for (1) adaptation to backbone models with different noise scales and (2) video control with sparse frame conditions. We evaluate SVD and I2VGen-XL on depth maps and scribbles as control conditions respectively.

| Method | Latent $z$ is given to ControlNet | FID ($\downarrow$) | Optical Flow Error ($\downarrow$) |
|---|---|---|---|
| *Adaptation to different noise scales* | | | |
| SVD (Blattmann et al., 2023) + CTRL-Adapter | ✔ | 4.48 | 2.77 |
| SVD (Blattmann et al., 2023) + CTRL-Adapter | ✘ | **3.82** | **2.96** |
| *Sparse frame conditions* | | | |
| I2VGen-XL (Zhang et al., 2023d) + CTRL-Adapter | ✔ | 7.20 | 5.13 |
| I2VGen-XL (Zhang et al., 2023d) + CTRL-Adapter | ✘ | **5.98** | **4.88** |

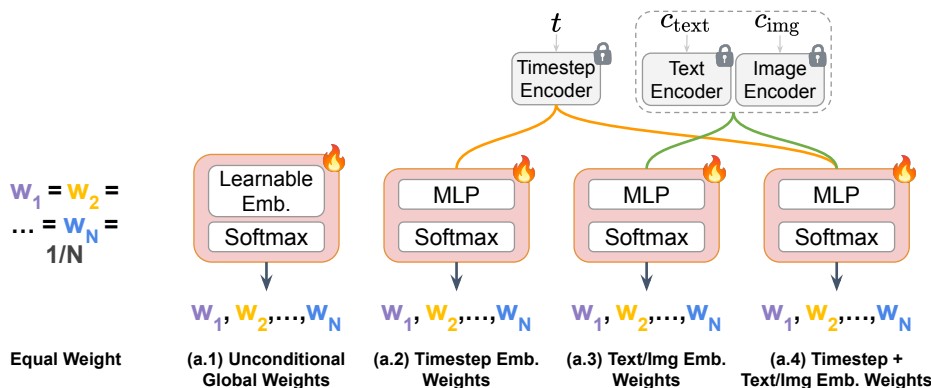

Figure 17: Visualization of different **global** MoE routing methods.

# F ADDITIONAL QUANTITATIVE ANALYSIS

## F.1 TRADE-OFF BETWEEN VISUAL QUALITY AND SPATIAL CONTROL

In Fig. 18, Fig. 19, and Fig. 20, we show the visual quality (FID) and spatial control (SSIM/optical flow error) metrics with different numbers of denoising steps with spatial control (with the fusion of CTRL-Adapter outputs) on SDXL, SVD, and I2VGen-XL backbones respectively. Specifically, suppose we use $N$ denoising steps during inference, a control guidance level of $x\%$ means that we fuse CTRL-Adapter features to the video diffusion U-Net during the first $x\% \times N$ denoising steps, followed by $(100 - x)\% \times N$ regular denoising steps. In all experiments, we find that increasing the number of denoising steps with spatial control improves the spatial control accuracies (SSIM/optical flow error) but hurts visual quality (FID).

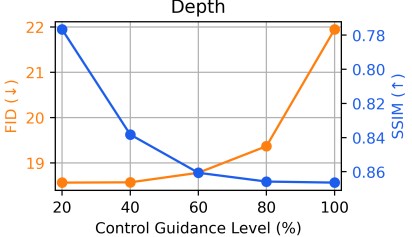 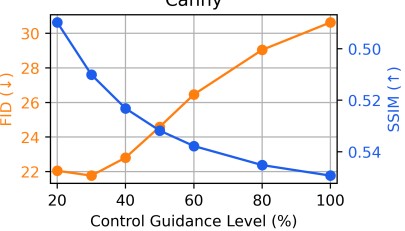

Figure 18: Trade-off between generated visual quality (FID) and spatial control accuracy (SSIM) on **SDXL**. Control guidance level of $x$ represents that we apply CTRL-Adapter in the first $x\%$ of the denoising steps during inference. A control guidance level between $30\%$ and $60\%$ usually achieves the best balance between image quality and spatial control accuracy.

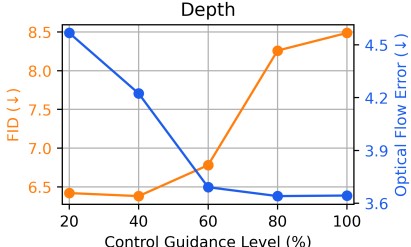 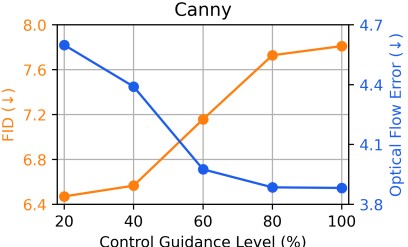

Figure 19: Trade-off between generated visual quality (FID) and spatial control accuracy (Optical Flow Error) on **SVD**. Control guidance level of $x$ represents that we apply CTRL-Adapter in the first $x\%$ of the denoising steps during inference. A control guidance level between $40\%$ and $60\%$ usually achieves the best balance between image quality and spatial control accuracy

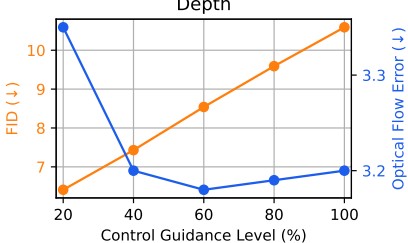 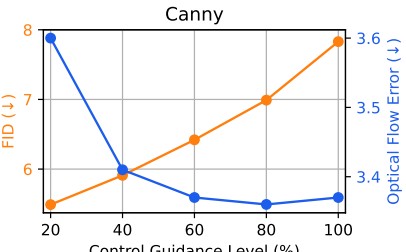

Figure 20: Trade-off between generated visual quality (FID) and spatial control accuracy (Optical Flow Error) on **I2VGen-XL**. Control guidance level of $x$ represents that we apply CTRL-Adapter in the first $x\%$ of the denoising steps during inference. A control guidance level between $40\%$ and $60\%$ usually achieves the best balance between image quality and spatial control accuracy.

## F.2 CLIPScore as an Evaluation Metric

We also experiment with using CLIPScore on a total of 13 different image control training settings, since it is a common metric for visual-text alignment. However Fig. 21 shows that CLIPScore does not change much as the training proceeds, even though there are clear improvements in visual quality and control accuracy. This indicates that CLIPScore is not a meaningful metric for measuring accuracy of controlled image/video generation.

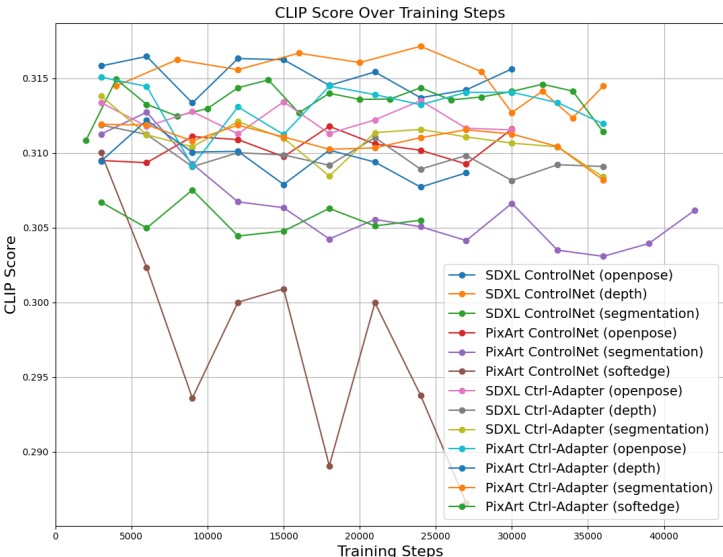

Figure 21: CLIPScore-based evaluation on 13 different image control training settings. CLIPScore does not change much as the training proceeds, while there are clear improvements in visual quality and control accuracy.

## F.3 Comparison with Control-LoRA

We use the latest Control-LoRA[9] (e.g., version 3), and evaluated all their available official checkpoints for a total of 5 control conditions (canny for SDXL as well as depth, normal, segmentation, openpose for SDv1.5). We follow the same evaluation setting by randomly sampling 1000 images from COCO2017 val, and use FID for image quality metric and SSIM/mAP for control quality metrics. As we can see from Table 11, CTRL-Adapter is strictly better than Control-LoRA on both generated image quality and control quality. Therefore, compared with Control-LoRA which sacrifices image quality and control quality for higher efficiency, our CTRL-Adapter achieves efficient training without trade-off in performance. We would like to point out that the main contribution of our paper is efficient training without trade-off in performance instead of only focusing on training/inference efficiency. The lightweight design of Control-LoRA makes it achieves higher efficiency compared with ControlNet, but suffers from large image quality and control quality drop.

Table 11: Comparison of CTRL-Adapter with Control-LoRA

| Method | Depth | | Canny | | Segmentation | | Normal | | Openpose | |
|---|---|---|---|---|---|---|---|---|---|---|
| | FID (↓) | SSIM (↑) | FID (↓) | SSIM (↑) | FID (↓) | SSIM (↑) | FID (↓) | SSIM (↑) | FID (↓) | SSIM (↑) |
| Control-LoRA | 21.71 | 0.747 | 16.02 | 0.451 | 19.95 | 0.595 | 24.29 | 0.775 | 31.07 | 0.312 |
| CTRL-Adapter | **15.12** | **0.839** | **13.97** | **0.560** | **15.67** | **0.673** | **14.94** | **0.818** | **18.74** | **0.521** |

---

[9]https://huggingface.co/stabilityai/control-lora/tree/main

F.4    A UNIFIED MULTI-TASK ADAPTER V.S. INDIVIDUAL TASK-SPECIFIC ADAPTERS

In Table 12, we provide an extended version of the results as shown in Sec. 4.7. Specifically, we added one row that trained solely on depth but applied to other conditions, and a row with plain diffusion without ControlNet as a baseline for comparison. Following Table 12, we report FID and SSIM as evaluation metrics for visual quality and spatial control. As we can see from the tables below, zero-shot results with SDXL+Ctrl-Adapter trained on depth and applied to other conditions achieve worse performance compared to individual and unified adapters (which is expected), but are much better than pure SDXL without ControlNet. This proves the effectiveness of zero-shot transferability of Ctrl-Adapter.

Table 12:  Training a unified CTRL-Adapter with SDXL backbone achieves comparable FID/SSIM to training individual CTRL-Adapters; evaluated on 1K samples from COCO val2017.

| Method | Depth | | Canny | | Softedge | | Lineart | | Segmentation | | Normal | |
|---|---|---|---|---|---|---|---|---|---|---|---|---|
| | FID ($\downarrow$) | SSIM ($\uparrow$) | FID ($\downarrow$) | SSIM ($\uparrow$) | FID ($\downarrow$) | SSIM ($\uparrow$) | FID ($\downarrow$) | SSIM ($\uparrow$) | FID ($\downarrow$) | SSIM ($\uparrow$) | FID ($\downarrow$) | SSIM ($\uparrow$) |
| Individual CTRL-Adapters | 14.87 | 0.8398 | 14.00 | 0.5600 | 14.13 | 0.5123 | 11.26 | 0.5216 | 16.03 | 0.6732 | 14.94 | 0.8182 |
| Unified CTRL-Adapter | 15.13 | 0.8358 | 13.97 | 0.5454 | 14.25 | 0.4934 | 12.99 | 0.5117 | 15.68 | 0.6682 | 14.94 | 0.8143 |
| Trained Solely on Depth | 14.87 | 0.8398 | 16.17 | 0.3685 | 15.79 | 0.3822 | 15.32 | 0.4128 | 16.49 | 0.5623 | 15.76 | 0.7716 |
| Plain SDXL w/o ControlNet | 17.49 | 0.7079 | 17.49 | 0.2916 | 17.49 | 0.2831 | 17.49 | 0.3241 | 17.49 | 0.4624 | 17.49 | 0.6306 |

# G  ADDITIONAL QUALITATIVE ANALYSIS

## G.1  VISUALIZATION OF SPATIAL FEATURE MAPS

As mentioned in Sec. 2 and Appendix E.2, the spatial features encoded in the U-Net of ControlNets and the DiT blocks are structurally different. We visualize this difference in Fig. 22. For the DiT-based model, we use PixArt-$\alpha$ as a representative. We follow the visualization method mentioned in Tumanyan et al. (2023). Specifically, we first extract the spatial features from different DiT blocks and U-Net middle/output blocks at the last denoising step during inference. For each block, we applied PCA to the extracted features and visualized the top three leading components.

As shown in Fig. 22, almost all 28 DiT blocks capture global and semantic information about the object "cactus". This observation is consistent with the findings in Guo & Yue (2024). On the other hand, the U-Net blocks in ControlNet demonstrate a coarse-to-fine pattern as the feature map size increases. This indicates that mapping output blocks A/B of ControlNet to DiT blocks is a better option compared to using middle or output blocks C/D of the ControlNet.

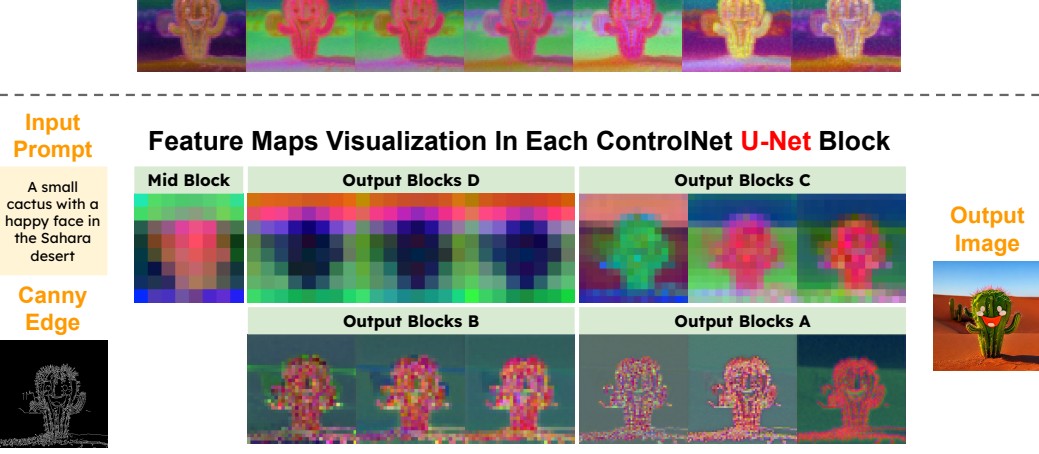

Figure 22: Visualization of spatial feature maps in PixArt-$\alpha$ and canny edge ControlNet. We first extract the spatial features from different DiT blocks and U-Net middle/output blocks at the last denoising step during inference. For each block, we applied PCA to the extracted features and visualized the top three leading components. Almost all 28 DiT blocks capture global and semantic information about the object "cactus", while the U-Net blocks in ControlNet demonstrate a coarse-to-fine pattern as the feature map size increases.

### G.2 FAST TRAINING CONVERGENCE

In addition to the quantitative results shown in Fig. 2 and Fig. 9, and the qualitative comparison on PixArt-$\alpha$ in Fig. 8, we provide additional visualization for SDXL depth ControlNet + CTRL-Adapter training. The training speed test is performed on 4 A100 80GB GPUs, with a batch size of 1 per GPU. As shown in Fig. 23, for relatively easy examples (*i.e.*, bedroom, sandwich, bus), our CTRL-Adapter training can converge within 4.5 GPU hours (which is equivalent to around 1.125 hours measured in training clock time). For complex examples and those requiring fine details (*i.e.*, surfing man, group of kids), our CTRL-Adapter can also converge within around 6 to 7.5 GPU hours (which is equivalent to 1.5 to 1.875 hours measured in training clock time), which proves the training efficiency of our CTRL-Adapter.

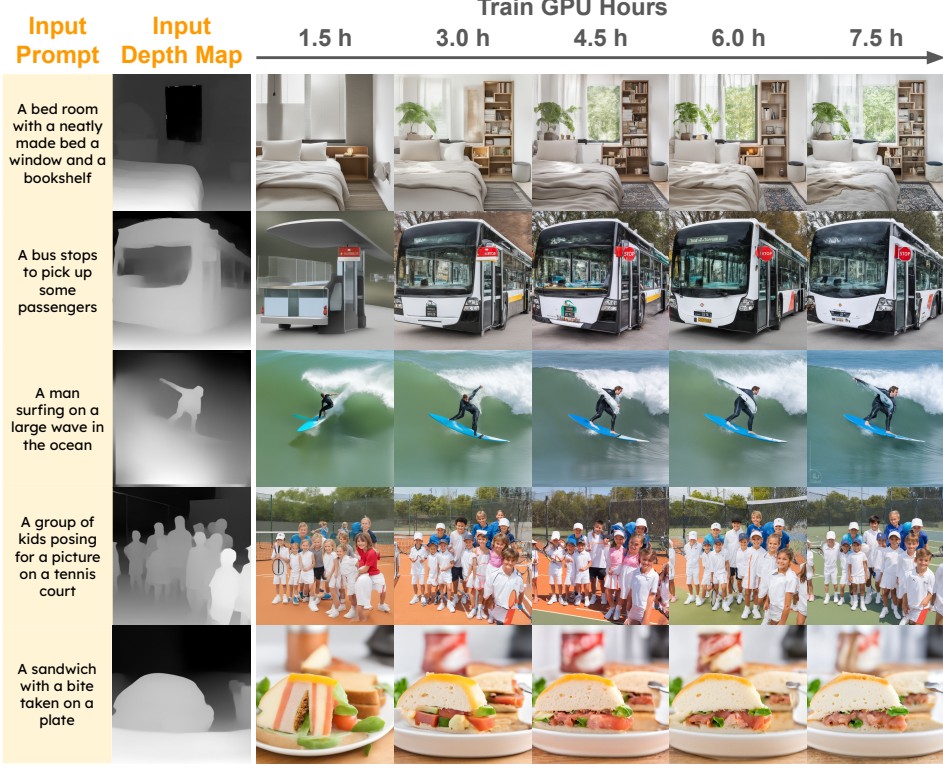

Figure 23: Training efficiency of CTRL-Adapter on SDXL backbone. Total training GPU hours are measured on 4 A100 80GB GPUs, with batch size per GPU equal to 1.

# H  ADDITIONAL VISUALIZATION EXAMPLES

We provide more qualitative examples in this section.

## H.1  VIDEO GENERATION VISUALIZATION EXAMPLES

In Fig. 24, we show video generation results on COCO val2017 split using depth map and canny edge as control conditions. We visualize baseline methods as well as CTRL-Adapters built on top of Hotshot-XL (Mullan et al., 2023), SVD (Blattmann et al., 2023), I2VGen-XL (Zhang et al., 2023d), and Latte (Ma et al., 2024b).

In Fig. 25 and Fig. 26, we show video generation results with I2VGen-XL using depth map and canny edge extracted from videos from Sora[10] and the internet.

In Fig. 27, we show video generation results with Latte using soft edge extracted from videos from Sora[11] and the internet.

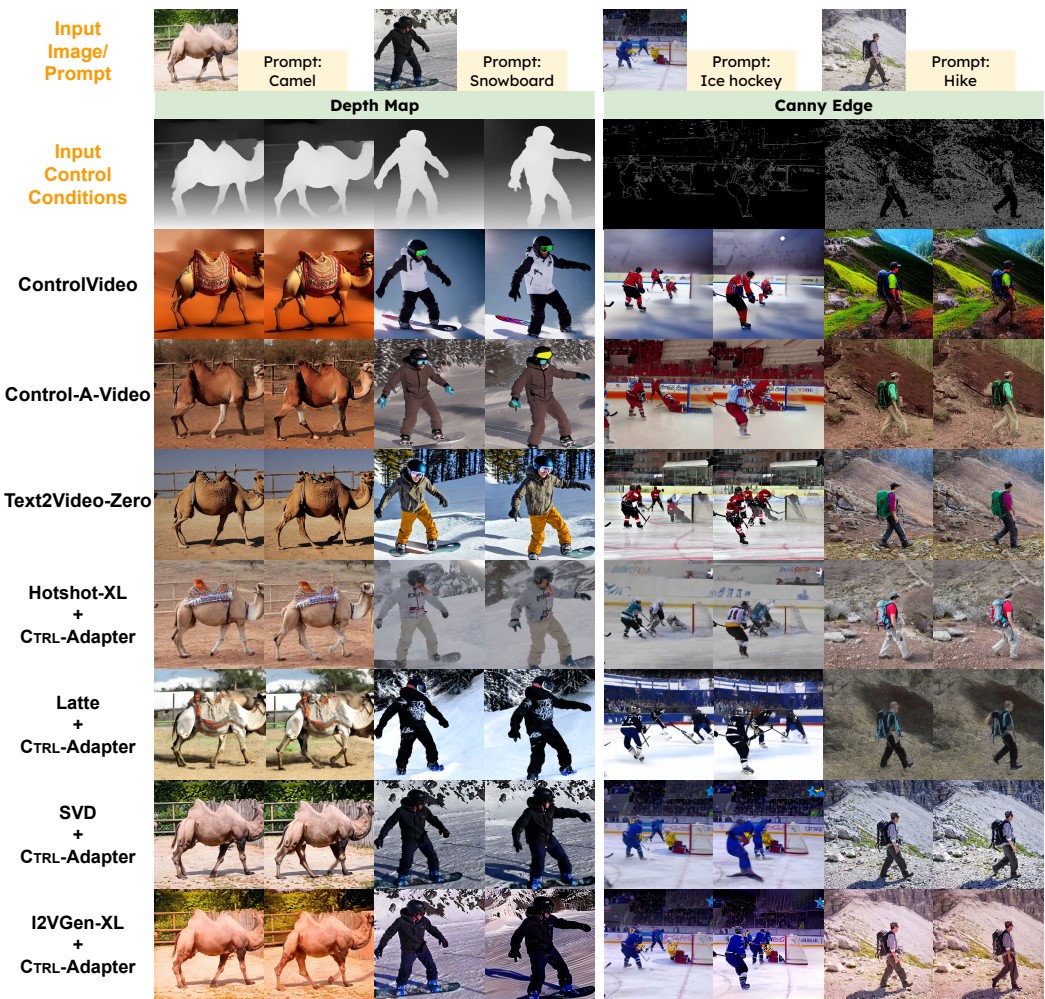

Figure 24: Videos generated from different video control methods and CTRL-Adapter on DAVIS 2017, using depth map (left) and canny edge (right) conditions.

---

[10] https://openai.com/sora
[11] https://openai.com/sora

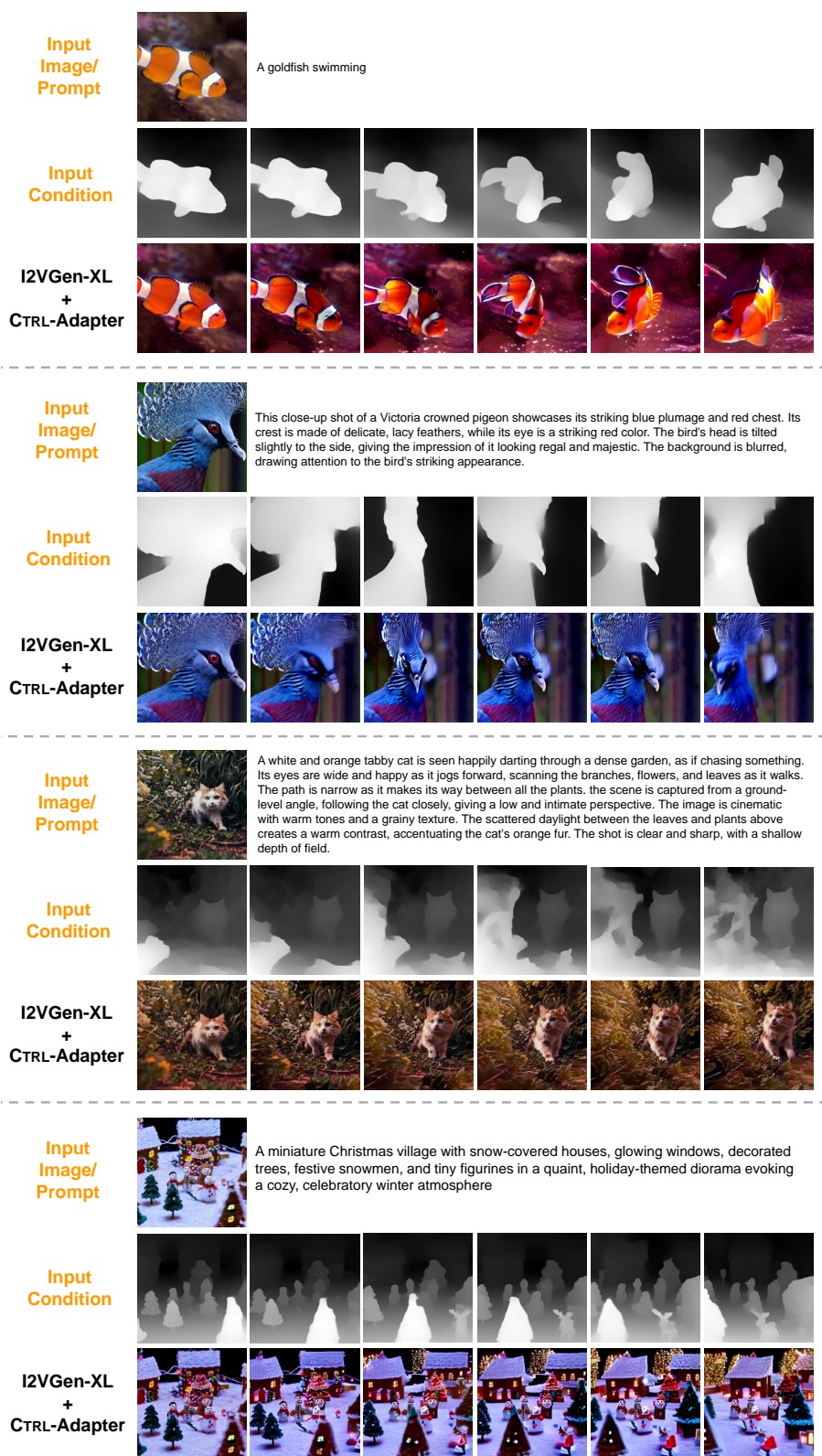

Figure 25: Video generation with **I2VGen-XL + CTRL-Adapter** using **depth map** as a control condition.

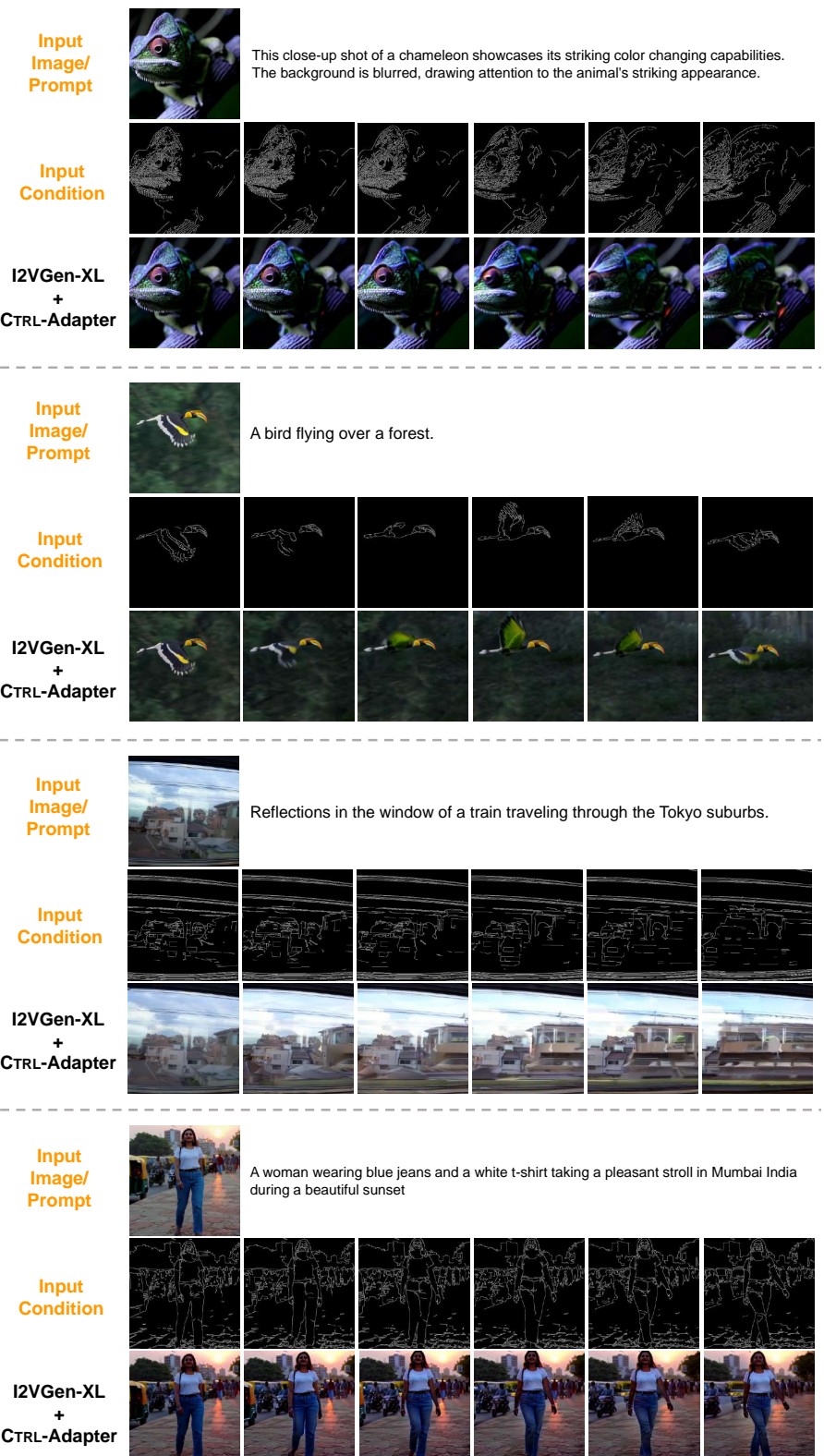

Figure 26: Video generation with **I2VGen-XL + CTRL-Adapter** using **canny edge** as control condition.

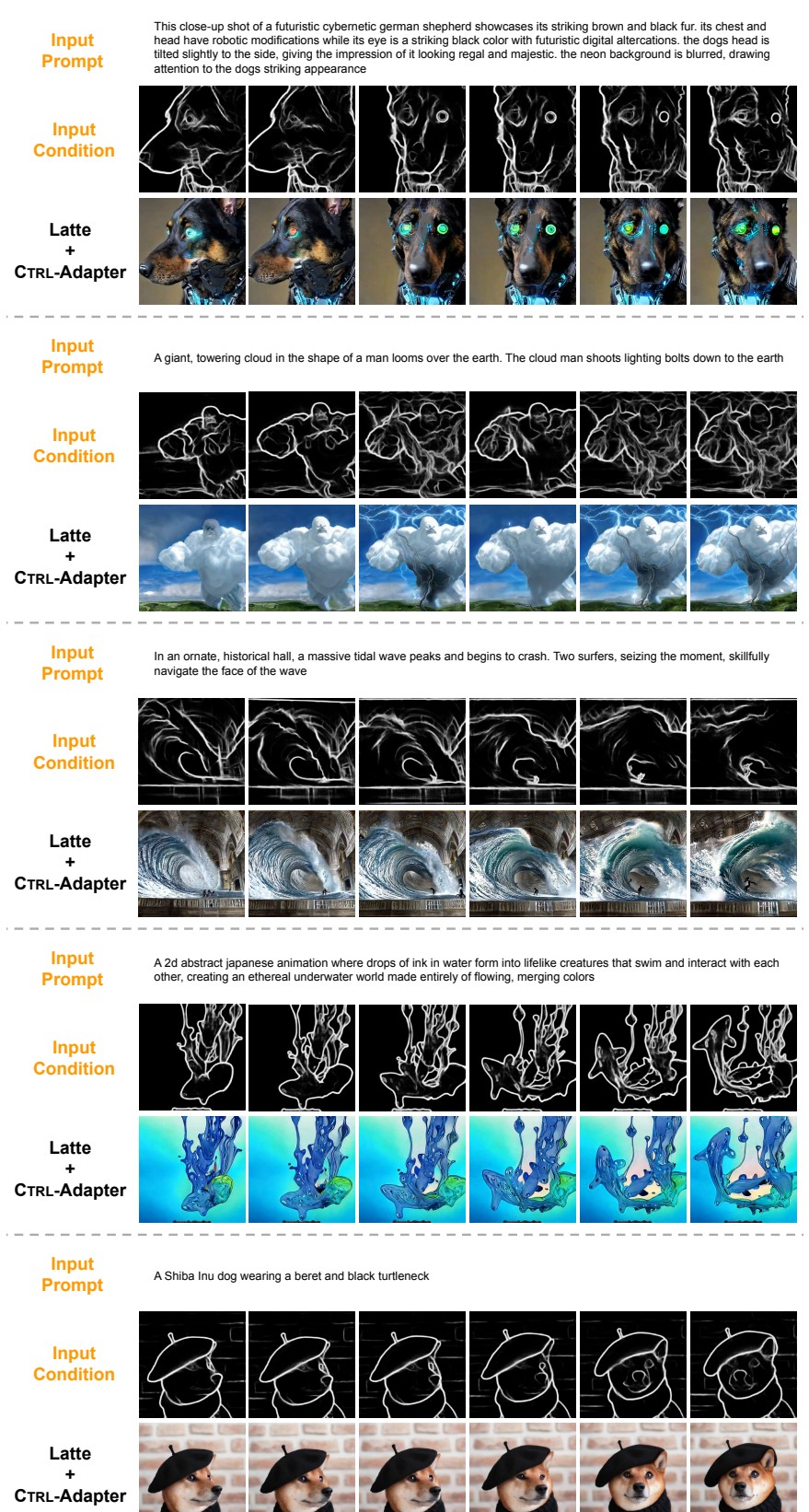

Figure 27: Video generation with **Latte + CTRL-Adapter** using **soft edge** as control condition.

Fig. 28 shows example videos generated with single and multiple conditions. While all videos correctly capture the high-level dynamics of 'a woman wearing purple strolling during sunset', the videos generated with more conditions show more robustness in several minor artifacts. For example, when only depth map is given (Fig. 28 a), the building behind the person is blurred. When depth map and human pose are given (Fig. 28 b), the color of the purse changes from white to purple. When four conditions (depth map, canny edge, human pose, and semantic segmentation) are given, such artifacts are removed (Fig. 28 c). In Fig. 29, we show multi-condition control examples with I2VGen-XL.

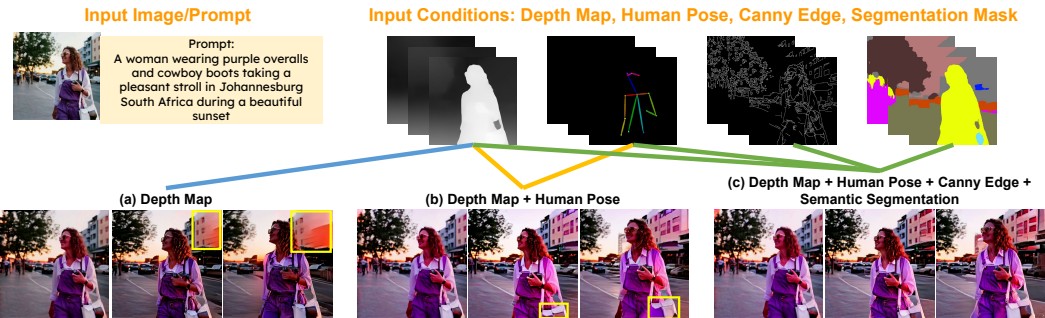

Figure 28: Video generation from single and multiple conditions with CTRL-Adapter on I2VGen-XL. (a) single condition: depth map; (b) 2 conditions: depth map + human pose; (c) 4 conditions: depth map + human pose + canny edge + semantic segmentation. Adding more conditions can help fix several minor artifacts (*e.g.*, in (a) – building is blurred; in (b) – purse color changes).

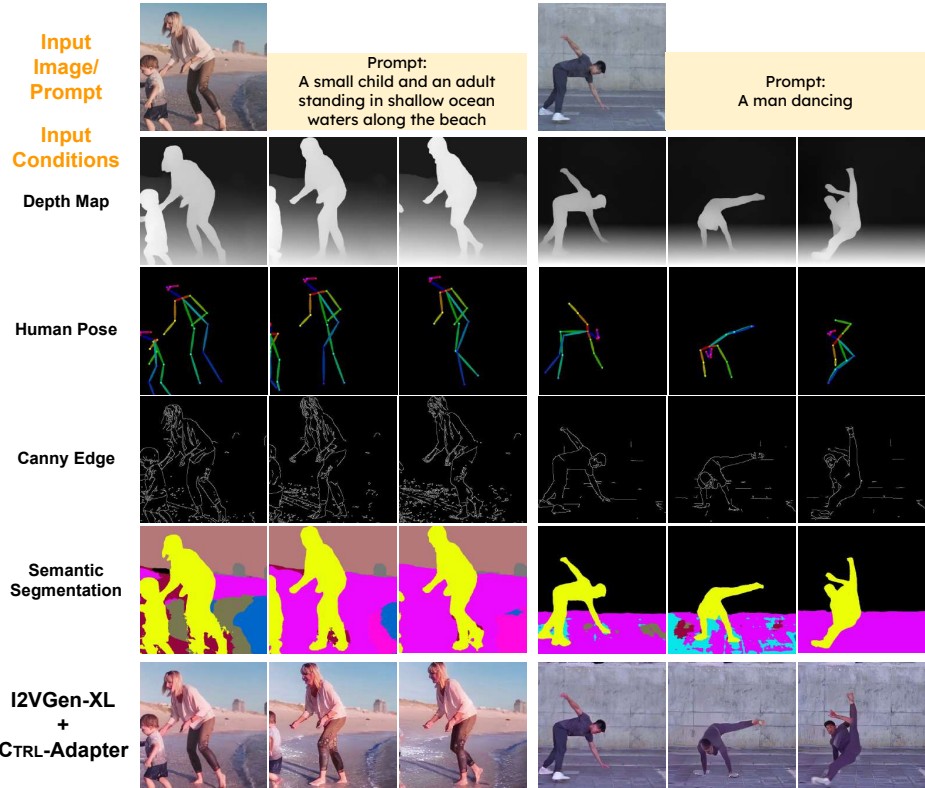

Figure 29: Video generated with **I2VGen-XL + CTRL-Adapter** from 4 control conditions: depth map + human pose + canny edge + semantic segmentation map.

## H.2 IMAGE GENERATION VISUALIZATION EXAMPLES

In Fig. 30 and Fig. 31, we show image generation results on COCO val2017 split using depth map and canny edge as control conditions.

In Fig. 32, Fig. 33, and Fig. 34, we show image generation results on prompts from Lexica[12] using depth map, canny edge, and soft edge as control conditions.

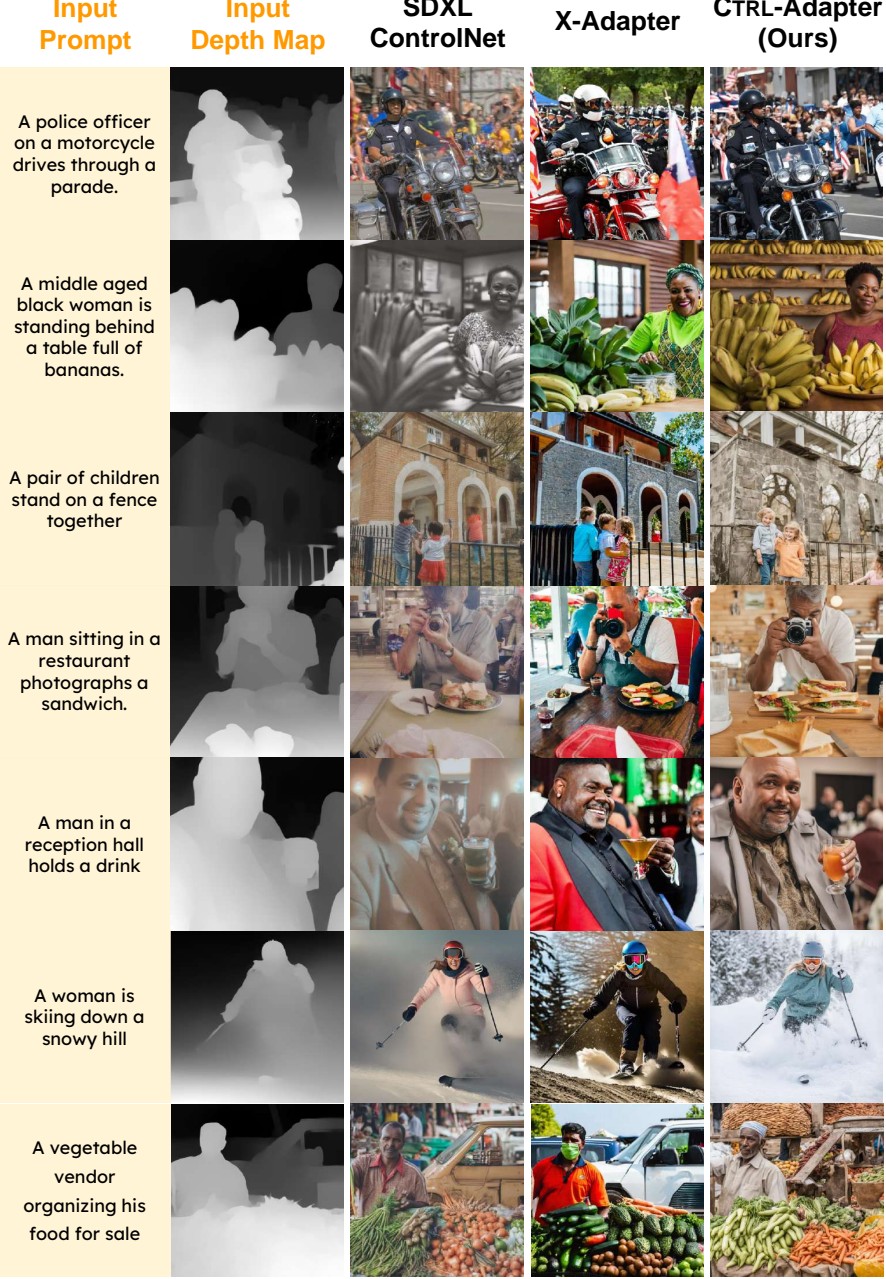

Figure 30: Image generation from different **SDXL**-based image control methods and CTRL-Adapter on COCO val2017 split using **depth map** as control condition.

---

[12] https://lexica.art/

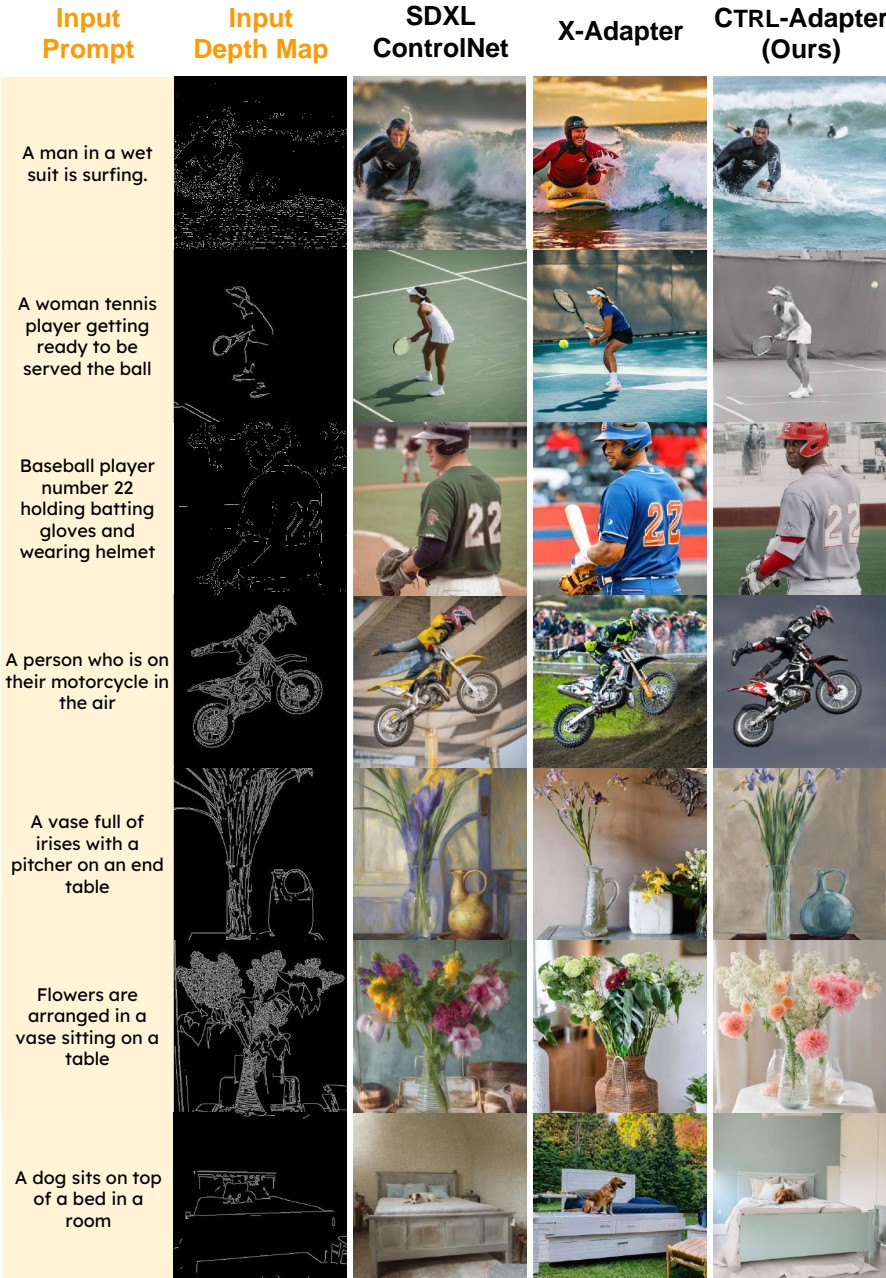

Figure 31: Image generation from different **SDXL**-based image control methods and CTRL-Adapter on COCO val2017 split using **canny edge** as control condition.

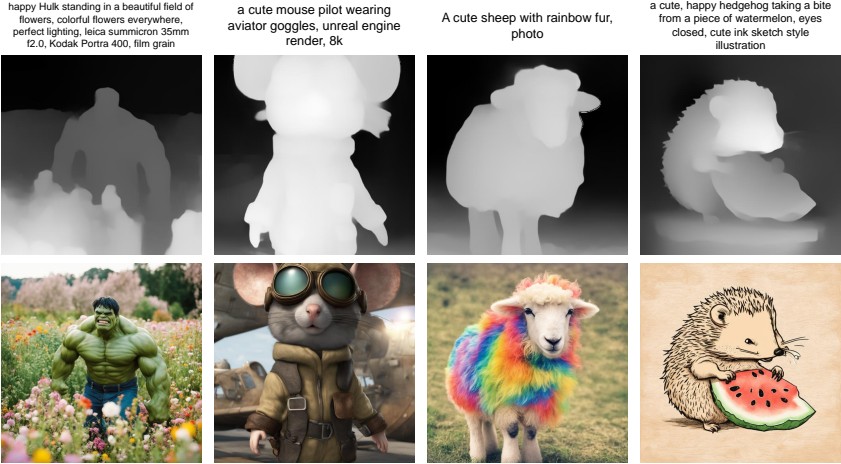

Figure 32: Image generation with **SDXL + CTRL-Adapter** using **depth map** as a control condition.

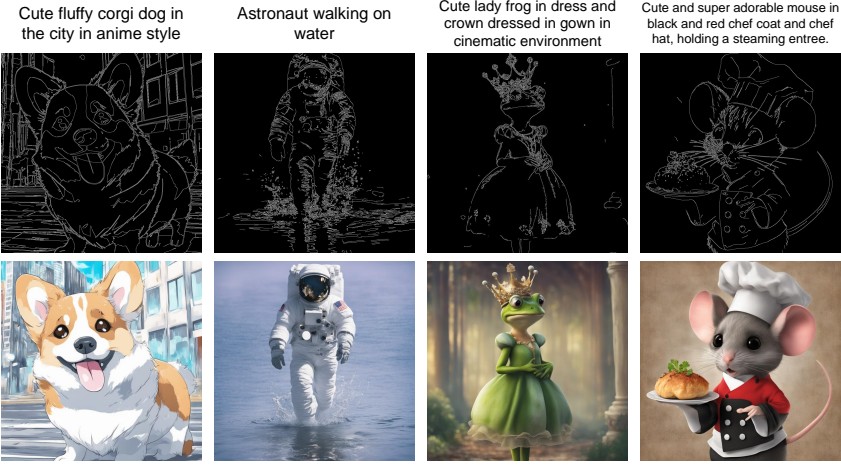

Figure 33: Image generation with **SDXL + CTRL-Adapter** using **canny edge** as a control condition.

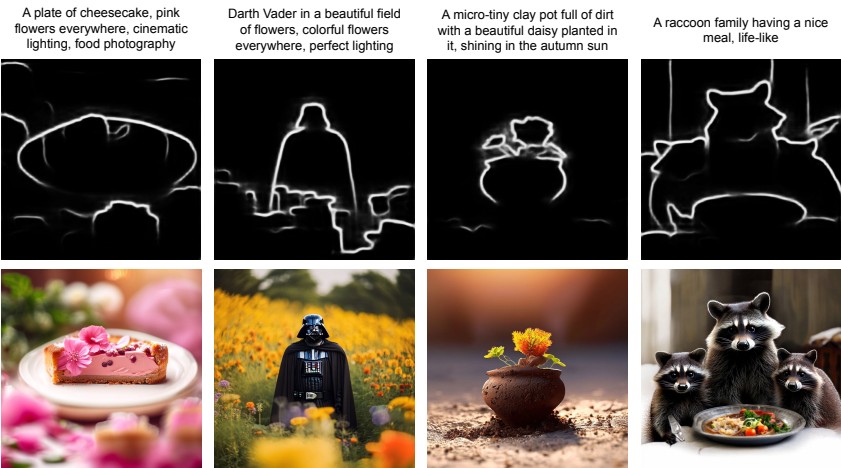

Figure 34: Image generation with **PixArt-α + CTRL-Adapter** using **soft edge** as a control condition.

### H.3 VISUALIZATION EXAMPLES FOR ADDITIONAL DOWNSTREAM TASKS

Here, we describe in detail how our CTRL-Adapter can be seamlessly integrated into a wide variety of downstream tasks including video editing, video style transfer, and text-guided motion control.

**Video editing.** Video editing can be achieved by combining image and video CTRL-Adapters. The procedure is as follows:

- Firstly, given a source video, we first extract the control condition(s). We can either extract a single control condition (*e.g.*, depth map), or multiple control conditions (*e.g.*, depth map, canny edge, segmentation, *etc.*) to enhance performance (as we observe in Tab. 3, Fig. 28 and Fig. 29 that multi-condition control improves spatial control accuracy).

- Next, given a user-provided prompt together with the extracted control condition(s), we can use image CTRL-Adapter (*i.e.*, SDXL + CTRL-Adapter) to generate the first frame of the video.

- Finally, we can use video CTRL-Adapter (*i.e.*, I2VGen-XL + CTRL-Adapter), with the generated first frame image, text prompt, and extracted control conditions as inputs for final video generation.

In Fig. 35, we provide additional visualizations of the camel example in our main paper.

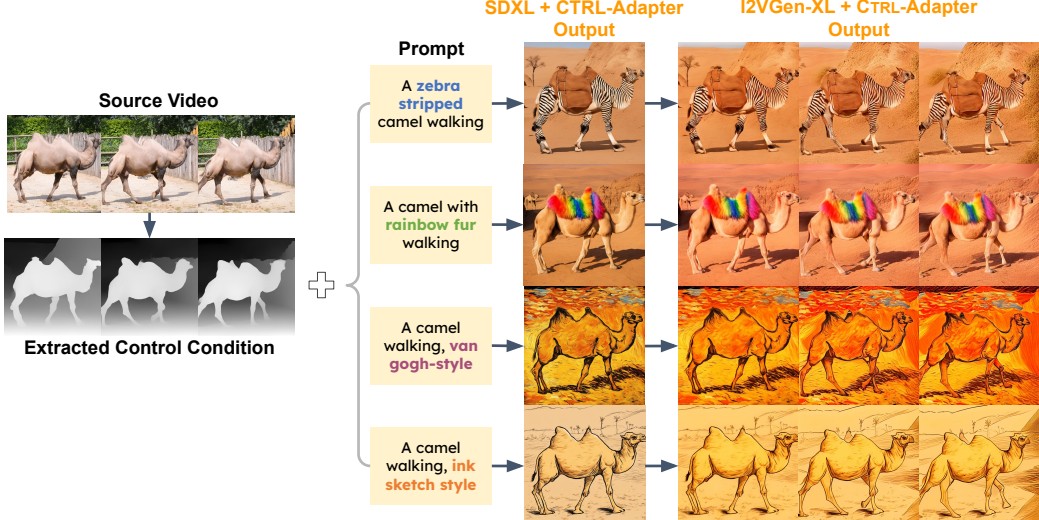

Figure 35: Video editing by combining **SDXL** and **I2VGen-XL**, where both models are equipped with spatial control via CTRL-Adapter. First, we extract conditions (*e.g.*, depth map) from the original video. Next, we create the initial frame with SDXL + CTRL-Adapter. Lastly, we provide the newly generated initial frame and frame-wise conditions to I2VGen-XL + CTRL-Adapter to obtain the final edited video. This video editing framework can edit both object and background.

**Text-Guided Motion Control.** This task can be achieved by combining video CTRL-Adapter with inpainting ControlNet. We train such CTRL-Adapter as follows:

- Firstly, for each training video, we randomly select a random block in the image, with the width and height of the block uniformly sampled from 0.25 to 0.75 of the image size.
- Next, we color the block area of the video frames as black color (these processed frames can be regarded as control condition sequences like depth maps).
- Finally, we can train CTRL-Adapter with the frozen inpainting ControlNet similar as other types of CTRL-Adapters mentioned in our main paper.

In Fig. 36, we provide additional visualizations of text-guided image amination.

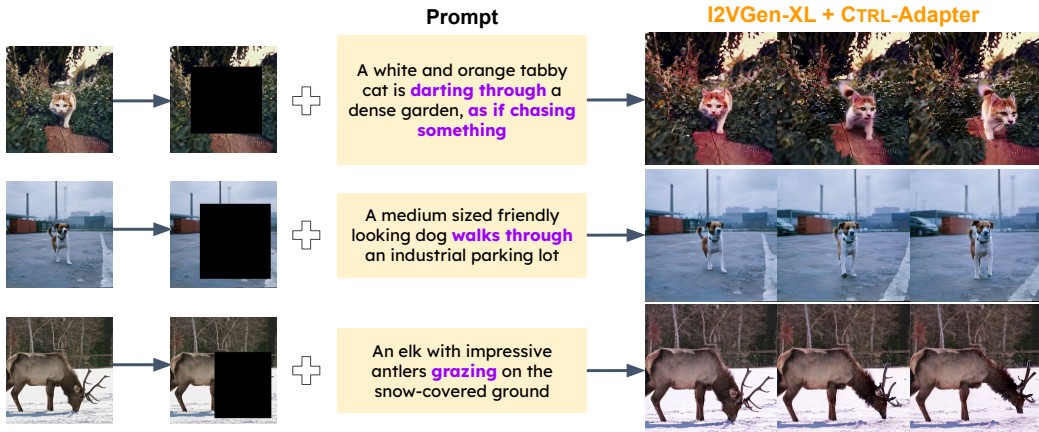

Figure 36: Text-guided motion control by combining **inpainting ControlNet** with **I2VGen-XL + CTRL-Adapter**. Specifically, inpainting ControlNet takes the masked frames as well as text prompt as inputs. The output feature maps of inpainting ControlNet are then given to CTRL-Adapter built on top of I2VGen-XL to generate the final video. Object(s) in the masked area can follow the motion described in the text prompt. The unmasked area can be either static or dynamic.

**Video style transfer.** This task can be achieved by combining video CTRL-Adapter with shuffle ControlNet. We train such CTRL-Adapter as follows:

- Firstly, for the first frame of each training video, we apply the content shuffle detector implemented in the `controlnet_aux`[13] library, to get a shuffled image.
- Next, we repeat this shuffled image $N$ times, with $N$ equal to the number of output frames of the backbone video diffusion model. These repeated images can be regarded as control condition sequences like depth maps.
- Finally, we can train CTRL-Adapter with the frozen shuffle ControlNet similar as other types of CTRL-Adapters mentioned in our main paper.

in Fig. 37, we provide additional visualizations of video style transfer.

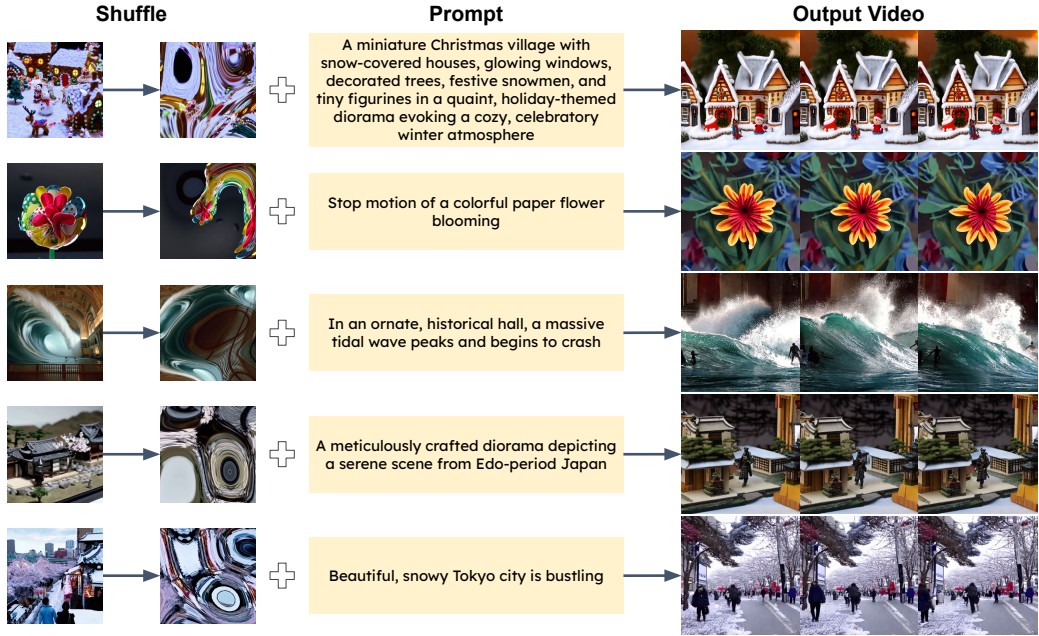

Figure 37: Video style transfer by combining **shuffle ControlNet** with **Latte + CTRL-Adapter**. Specifically, shuffle ControlNet takes the shuffled image as well as text prompt as inputs. The output feature maps of shuffle ControlNet are then given to CTRL-Adapter built on top of Latte to generate the final video. The generated videos maintain similar style as the input image before shuffling.

---

[13]https://github.com/huggingface/controlnet_aux

## I  ETHICS STATEMENT

CTRL-Adapter is motivated by the fact that training ControlNets for new diffusion models, especially video diffusion models that need to consider temporal consistency, can be a huge burden for many users. As shown in Fig. 2, by adopting pretrained ControlNets, training CTRL-Adapter can be significantly faster than training other controllable image or video generation methods. For example, with the same type of compute (*i.e.*, A100 80GB GPUs), CTRL-Adapter trained on SDXL depth ControlNet for 10 GPU hours can outperform SDXL ControlNet trained for 700 GPU hours. This drastically reduces the carbon emissions footprint by over 70 times. Therefore, we believe that our work can be a strong contribution to efficient and controllable image and video generation.

While our framework can benefit numerous applications in controllable generation, similar to other image and video generation frameworks, it can also be used for potentially harmful purposes (e.g., creating false information or misleading videos). Therefore, it should be used with caution in real-world applications.

Note that since CTRL-Adapter is a method to equip current open-source image and video diffusion models with better control, its performance, quality, and potential visual artifacts largely depend on the capabilities (e.g., motion styles and video length) of the backbone models used. For example, if a diffusion model cannot handle complex motions, CTRL-Adapter built on top of this backbone might lead to non-optimal complex motion control.

## J  REPRODUCIBILITY STATEMENT

We include our training and inference code in the supplementary materials. We explained in detail our architecture design in Sec. 2 and Appendix B, experimental setup details in Appendix D, and training and inference details in Appendix C.

## K  LICENSE

We use standard licenses from the community and provide the following links to the licenses for the datasets, codes, and models that we used in this paper. For further information, please refer to the specific link.

PyTorch (Ansel et al., 2024): BSD-style

HuggingFace Transformers (Wolf et al., 2020): Apache License 2.0

HuggingFace Diffusers (von Platen et al., 2022): Apache License 2.0

ControlNet (Zhang et al., 2023c): Apache License 2.0

SDXL (Podell et al., 2024): MiT License

PixArt-$\alpha$ (Chen et al., 2024c): AGPL-3.0 License

I2VGen-XL (Zhang et al., 2023d): MiT License

Stable Video Diffusion (SVD) (Blattmann et al., 2023): MiT License

Latte (Ma et al., 2024b): Apache License 2.0

Hotshot-XL (Mullan et al., 2023): Apache License 2.0

LAION dataset (Schuhmann & Bevan, 2023): MiT License

Panda70M dataset (Chen et al., 2024d): License

COCO dataset (Lin et al., 2014): CC BY 4.0

DAVIS 2017 dataset (Pont-Tuset et al., 2017): CC BY 4.0

