# OpenReview forum: "Ctrl-Adapter: An Efficient and Versatile Framework for Adapting Diverse Controls to Any Diffusion Model"
_ICLR.cc/2025/Conference — ICLR 2025 Oral_

### Official Review · Reviewer_pa78 · 2024-10-28

**Soundness:** 3
**Presentation:** 2
**Contribution:** 3
**Rating:** 8
**Confidence:** 3

**Summary:**

This work Ctrl-Adapter extends X-Adapter to a more general setting: the pretrained ControlNet and downstream diffusion models can now differ in modality (images, videos) and architecture (e.g., UNet or Transformers). The authors show in extensive experiments that Ctrl-Adapter works well in many scenarios and outperforms X-Adapter.

**Strengths:**

- The major technical contribution is to extend X-Adapter by involving temporal convolutions for the support of adapting from image to video.

- There are a bunch of engineering contributions to improve performance: (1) the utilization of Q-former; (2) skipping latents; (3) handling noise scheduling shift.

- A really extensive experiments to demonstrate the strength of their work in various scenarios: single condition, multi conditions, zero-shot, from image to image, from image to video.

**Weaknesses:**

1.	Overall, the writing could be improved to enhance clarity. The methodology section should be more focused and closely aligned with the key contributions. Currently, it is cluttered with engineering details and lacks mathematical formulation, which makes it harder to grasp the technical novelty.

2.	Does skipping latent require retraining ControlNet?

3.	The Multi-ControlNet setup is unclear:
	•	Are six conditions trained simultaneously in each iteration?
	•	In Section 4.5, Figure 8 (zero-shot results), the zero-shot inferences under different conditions do not appear to align accurately with the intended conditions, which is understandable since they were not trained on these. However, could you provide an example without ControlNet for comparison? Could you also add quantitative results for the zero-shot experiment in Table 4? (Consider adding a row for models trained solely on depth but applied to other conditions, and a row with plain diffusion without ControlNet as a baseline for comparison.)

4.  What is architecture difference with X-Adapter? What is the training speed improvement over X-Adapter come from?

**Questions:**

See weakness part. Overall, I lean to accept the paper. I need more explanation compared to X-Adapter in the technical side: (1) the architecture difference; (2) the training difference; (3) explain what is the performance and training speed come from.

---

> ### Author Response · Authors · 2024-11-21
> **Author Response to Reviewer pa78 (Part 1/3)**
>
> We would like to sincerely thank the Reviewer for the insightful and detailed comments.
>
>
> > ### W1: Overall, the writing could be improved to enhance clarity. The methodology section should be more focused and closely aligned with the key contributions. Currently, it is cluttered with engineering details and lacks mathematical formulation, which makes it harder to grasp the technical novelty.
>
>
> Thank you very much for your feedback. Following the Reviewer’s suggestion, we restructured the method section **in the updated PDF** by adding more details to highlight our technical novelties, including (1) architecture design for the adaptation to DiT-based image/video backbones, (2) skipping latents for robust adaptation to different noise scales and sparse frame inputs, and (3) inverse timestep sampling for robust adaptation to continuous diffusion timestamp samplers. In addition, we added more details about the MoE router in Sec. 2.2, and emphasized our training objectives in Sec. 2.1.
>
> We sincerely thank the Reviewer for understanding the challenge of presenting extensive content within the limited space constraints of this paper. Please let us know if there is any content or design aspect that remains unclear. We will do our best to address your concerns and provide detailed illustrations as needed.
>
>
> > ### W2: Does skipping latent require retraining of ControlNet?
>
> Thank you very much for this excellent question. Skipping latent does not require retraining of ControlNet (actually, none of our settings requires any update of ControlNet) because the ConrolNet is already familiar with such scenarios during training. Below we explain the intuition behind this in more detail:
>
> During ControlNet training, ControlNet takes the latent $z$ as one of the inputs, where Gaussian noises of different scales (from different noise scheduling timestamps $t$; e.g., $t ~ [1, 1000]$) are added to $z$. When timestamp $t$ is big (e.g., $t=1000$), the added Gaussian noise has a large scale, and when the time timestamp $t$ is small (e.g., $t=10$), the added Gaussian noise is close to zero and almost negligible. Therefore, when we skip latent to ControlNet, it is already a familiar scenario for ControlNet (almost equivalent to the scenario when time timestamp $t$ is small).

---

> > ### Comment · Reviewer_pa78 · 2024-11-25
> > **Reply to Part 1/3**
> >
> > Hi,
> > thanks for the response. Could you explain skipping latent more clear in the methodology part, like add an inline equation in the methodology.
> > Does it mean $c_f$ as the input to controlnet rather than $c_f + z_t$ at all timesteps? Or use $c_f + z_0$ as the input? Or you mean anything else.

---

> > > ### Author Response · Authors · 2024-11-25
> > > **Reply to new question from reviewer pa78**
> > >
> > > Dear Reviewer pa78,
> > >
> > > Thanks for your additional question! With latent skipping, the input to the ControlNet now is only $c_f$ instead of $c_f + z_t$ or $c_f + z_0$. We have clarified this point in the **newly updated PDF** colored in **red**.

---

> > > > ### Comment · Reviewer_pa78 · 2024-11-26
> > > > **skipping latent reply**
> > > >
> > > > Thanks. It was what I understood. But why in Fig 3, you still added $z_t$ to $c_f$, your latent skipping only happens in some noise steps?

---

> > > > > ### Author Response · Authors · 2024-11-26
> > > > >
> > > > > Thank you for your question!
> > > > >
> > > > > Fig. 3 is a general pipeline of our architecture not specifically focusing on latent skipping. The figure illustrating latent skipping is shown in Fig. 13 in the appendix (as mentioned in L221-L222 of our PDF). Latent skipping, when used, is applied to all denoising steps.

---

> > > > > > ### Comment · Reviewer_pa78 · 2024-11-26
> > > > > > **replace to Fig.3 latent skipping**
> > > > > >
> > > > > > Do you mean Fig. 16? Fig. 13 is a histogram.
> > > > > > Any reason not removing latent skipping if you use latent skipping always.

---

> > > > > > > ### Author Response · Authors · 2024-11-26
> > > > > > >
> > > > > > > Dear reviewer pa78,
> > > > > > >
> > > > > > > Thanks for your comment, and we are happy to provide more clarifications about this point. In the latest PDF, it's "_Figure 12: Left (default): latent $z_t$ is given to ControlNet. Right: latent $z_t$ not given to ControlNet_". If the figure number in your PDF version doesn't match, you can find this figure by its caption. Sincerely apologize for any potential confusions.
> > > > > > >
> > > > > > > **1. When we skip latents**
> > > > > > >
> > > > > > > We would like to kindly clarify that as we mentioned in L247 in our PDF "_...we find that skipping $z_t$ from ControlNet inputs is effective for CTRL-Adapter in certain settings..._", we suggest skipping latent only for the following two specific scenarios: (1) different noise scales, and (2) sparse frame conditions. We do not skipping the latent for other use cases mentioned in our paper.
> > > > > > >
> > > > > > > To quantitatively prove that skipping latents works better under these two scenarios, we show that skipping improves both FID and optical flow scores in Table 10 with caption "_Skipping latent from ControlNet inputs helps CTRL-Adapter for (1) adaptation to backbone models with different noise scales and (2) video control with sparse frame conditions. We evaluate SVD and I2VGen-XL on depth maps and scribbles as control conditions respectively._",
> > > > > > >
> > > > > > > **2. When we don't skip latents**
> > > > > > >
> > > > > > > In the common ControlNet usage scenarios, skipping-latents causes slight decrease in spatial control ability and lower visual quality (we have quantitative experiments in the reply "Official Comment by Authors" to reviewer iSfd).
> > > > > > >
> > > > > > > ---
> > > > > > > Once more, thank you very much for your reviews and feedback! If you have any additional questions, please feel free to let us know, and we are very willing to provide more explanations.

---

> ### Author Response · Authors · 2024-11-21
> **Author Response to Reviewer pa78 (Part 2/3)**
>
> > ### W3-1: Are six conditions trained simultaneously in each iteration?
>
> Thanks for this question. We combine the dataloader of six conditions in a round-robin fashion. For each training step, we randomly sample a condition and calculate the gradient for that condition. As the reviewer mentioned, including multiple conditions simultaneously in each mini-batch could be another alternative strategy that could also work for Ctrl-Adapter training.
>
> > ### W3-2: In Section 4.5, Figure 8 (zero-shot results), the zero-shot inferences under different conditions do not appear to align accurately with the intended conditions, which is understandable since they were not trained on these. However, could you provide an example without ControlNet for comparison?
>
> Thank you for raising this comment. In [this link](https://anonymous.4open.science/r/Ctrl-Adapter_ICLR2025_Rebuttal-27C9), we compare the video examples generated with the following settings on the cat walking example shown in Fig. 8. For each of the following variants, we generated 10 samples with different random seeds **without cherry-picking**. Note: you can download this repo by clicking “download repository” on the top right of the website (the whole zip file is 120MB).
> - (a) I2VGen-XL (no ControlNet)
> - (b) I2VGen-XL + Depth Ctrl-Adapter
> - (c) I2VGen-XL + Depth Ctrl-Adapter (zero-shot inference on normal map)
> - (d) I2VGen-XL + Depth Ctrl-Adapter (zero-shot inference on softedge)
>
> As we can see, for variant (a) I2VGen-XL (no ControlNet), the movement of the cat is often unnatural in the generated videos (e.g., the movement is not continuous, and the cat’s movement is not physical). In contrast, variants (b), (c), and (d) all generate much more natural and physically realistic walking movements of the cat. Therefore, from these examples, we can conclude that backbone models with control provide better motion guidance for objects compared to the variant without any control.
>
> > ### W3-3: Could you also add quantitative results for the zero-shot experiment? Consider adding models trained solely on depth but applied to other conditions, and a model with plain diffusion without ControlNet as a baseline for comparison.
>
> Thank you very much for this valuable suggestion! Firstly, we would like to kindly clarify that the results in Table 4 of our main paper is SDXL + Ctrl-Adapter for controllable **image** generation, while the zero-shot experiment in Sec. 4.5 of our main paper is I2VGen-XL + Ctrl-Adapter for controllable **video** generation.
>
> In order to show the zero-shot performance quantitatively as a complementary to the visualizations in Sec. 4.5, we conducted experiments that used the I2VGen-XL + Ctrl-Adapter trained on depth condition, then inference on the same depth condition, as well as zero-shot inference on canny edge and softedge. Following the suggestion, we added a plain diffusion without ControlNet as a baseline for comparison. We report the FID and Optical Flow Error to measure the visual quality and spatial control, respectively.
>
> As we can see from the two tables below, zero-shot inference on canny or softedge results in an increase in FiD score. This is expected due to the gap between training condition (depth) and the inference conditions (canny, softedge). In addition, we observe that the baseline FID score (6.95) is better than the scores derived from our models with spatial control. Such visual quality - spatial control trade-off is also expected and discussed in Appendix Sec. F.1 in our paper. As for the Optical Flow Error, we can also observe a similar increasing trend for zero-shot inference. In addition, we would like to kindly emphasize that the absolute value of this Optical Flow Error score is not directly comparable across different control conditions, because the score is calculated between the **conditions extracted from input and generated videos** (as we mentioned in “Evaluation Metrics” in Sec. 3 of our main paper), and the absolute score of this metric for different control conditions (depth, canny, softedge) are usually different.
>
> **FID:**
>
> |||||
> |-|-|-|-|
> |  | Inference with Depth | Inference with Canny | Inference with Softedge |
> | Trained with Corresponding Conditions | 7.43 | 6.81 | 7.18|
> | Trained with Depth Condition | 7.43 | 7.77 (**zero-shot**) | 8.99 (**zero-shot**) |
> | Baseline: I2VGen-XL without ControlNet = 6.95 |
>
> **Optical Flow Error:**
>
> |||||
> |-|-|-|-|
> |  | Inference with Depth | Inference with Canny | Inference with Softedge |
> | Trained with Corresponding Conditions | 3.20 | 2.44 | 2.49 |
> | Trained with Depth Condition | 3.20 | 3.10 (**zero-shot**) | 2.94 (**zero-shot**) |
> | Baseline: I2VGen-XL without ControlNet = 5.94 |

---

> ### Author Response · Authors · 2024-11-21
> **Author Response to Reviewer pa78 (Part 3/3)**
>
> > ### W4 & Q1: I need more explanation compared to X-Adapter in the technical side: (1) the architecture difference; (2) the training difference; (3) explain what is the performance and training speed come from.
>
> **(1) architecture difference**
>
> - As described in Sec 6 (L506-516), X-Adapter only tackles image control, Ctrl-Adapter tackles both image and video control.
> - As described in Appendix Sec. E.1.1 and Table 6, we have explored various combinations of Ctrl-Adapter modules: spatial convolution (SC), temporal convolution (TC), spatial attention (SA), and temporal attention (TA). Ctrl-Adapter uses SC+SA for image / SC+TC+SA+TA for video control. X-Adapter uses SC*3 modules, which we found our SC+SA is more effective in Table 6.
>
> **(2) training difference & (3) where performance and training speed improvement comes from**
>
> - As described in Sec 6 (L506-516) and Fig. 11, X-Adapter needs to be used with the source image diffusion model (SDv1.5) **during both training and inference**, whereas Ctrl-Adapter does not require to load SDv1.5 at all, making it more memory and computationally efficient.
>
> Once more, we sincerely thank you for all the comments and very useful feedback. We think that we have addressed all the questions in depth. If the reviewer has any additional questions, please let us know, and we will be more than happy to answer them.

---

> > ### Comment · Reviewer_pa78 · 2024-11-25
> > **Reply to Part 2 and 3**
> >
> > The replies addressed my concerns.

---

> > > ### Author Response · Authors · 2024-11-25
> > >
> > > We are glad that the Reviewer’s concerns in parts 2 and 3 were resolved in our previous response (please feel free to revisit your score!).
> > >
> > > We have also spent effort by collecting additional positive results for the following question from Reviewer:
> > > >"Could you also add quantitative results for the zero-shot experiment in Table 4? (Consider adding a row for models trained solely on depth but applied to other conditions, and a row with plain diffusion without ControlNet as a baseline for comparison.)?"
> > >
> > > in addition to the new zero-shot controllable **video** generation experiments with I2VGen-XL + Ctrl-Adapter (as shown in our original rebuttal), here, we show a new extra experiment for zero-shot controllable **image** generation with SDXL + Ctrl-Adapter. Specifically, following Reviewer’s suggestion on Table 4, we added one row that trained solely on depth but applied to other conditions, and a row with plain diffusion without ControlNet as a baseline for comparison. Following Table 4, we report FID and SSIM as evaluation metrics for visual quality and spatial control. As we can see from the tables below, zero-shot results with SDXL+Ctrl-Adapter trained on depth and applied to other conditions achieve worse performance compared to individual and unified adapters (which is expected), but are much better than pure SDXL without ControlNet. This proves the effectiveness of zero-shot transferability of Ctrl-Adapter.
> > >
> > > **FID**
> > >
> > > ||||||||
> > > |-|-|-|-|-|-|-|
> > > |  | Inference with Depth | Zero-Shot Inference with Canny | Zero-Shot Inference with Softedge |  Zero-Shot Inference with Lineart | Zero-Shot Inference with Segmentation |  Zero-Shot Inference with Normal |
> > > | Individual Ctrl-Adapters           | 14.87 | 14.00 | 14.13 | 11.26 | 16.03 | 14.94 |
> > > | Unified Ctrl-Adapter                 | 15.13 | 13.97 | 14.25 | 12.99 | 15.68 | 14.94 |
> > > | Trained Solely on Depth           | 14.87 | 16.17 | 15.79 | 15.32 | 16.49 | 15.76 |
> > > | Plan SDXL without ControlNet | 17.49 | 17.49 | 17.49 | 17.49 | 17.49 | 17.49 |
> > >
> > >
> > >
> > >
> > > **SSIM**
> > >
> > > ||||||||
> > > |-|-|-|-|-|-|-|
> > > |  | Inference with Depth | Zero-Shot Inference with Canny | Zero-Shot Inference with Softedge |  Zero-Shot Inference with Lineart | Zero-Shot Inference with Segmentation |  Zero-Shot Inference with Normal |
> > > | Individual Ctrl-Adapters           | 0.8398 | 0.5600 | 0.5123 | 0.5216 | 0.6732 | 0.8182 |
> > > | Unified Ctrl-Adapter                 | 0.8358 | 0.5454 | 0.4934 | 0.5117 | 0.6682 | 0.8143 |
> > > | Trained Solely on Depth           | 0.8398 | 0.3685 | 0.3822 | 0.4128 | 0.5623 | 0.7716 |
> > > | Plan SDXL without ControlNet | 0.7079 | 0.2916 | 0.2831 | 0.3241 | 0.4624 | 0.6306 |
> > >
> > >
> > > We believe that we have addressed all your previous questions, in particular: (1) restructured the method section to be more focused and closely aligned with the key contributions (in revised PDF) to improve presentation, (2) visualization examples without ControlNet for comparison (in anonymous github link), (3) more intuition and explanation compared to X-Adapter regarding architecture difference, training difference, and where the performance and training speed come from, and (4) some clarifications about latent skipping and multi-condition training details.
> > >
> > >
> > > Thank you again for your feedback. If the Reviewer has any additional questions, please let us know so that we can address them before the end of the discussion period.
> > >
> > > Once more, thank you very much for your reviews and feedback!
> > >
> > > Yours sincerely,
> > >
> > > The Authors

---

> ### Comment · Reviewer_pa78 · 2024-11-25
> **Concerns in technical contributions and paper writing**
>
> Dear Authors,
>
> I am maintaining my original score because I still do not fully grasp the technical contributions of this work. *The writing remains overly focused on minor engineering details rather than addressing the key technical challenges and the novel solutions proposed.*
>
> Let me outline my understanding and concerns:
>
> **1. Motivation:**
>
> My understanding of the motivation is as follows: X-Adapter is limited to the same base model, whereas this work extends ControlNet (originally developed for SD15) to other models (e.g., DiT, SDXL) and domains beyond image generation (e.g., video).
>
>
> **2. Technical Challenges:**
> Unfortunately, this part remains unclear in the current writing. The paper seems to skip over the challenges and instead jumps directly into engineering solutions like “skip latents” and “inverse noise scheduler.” From the motivation, there are two key issues, specifically:
>
> *(1) From Image to Video:*
>
> Transferring from image generation to video generation introduces an additional temporal dimension, which should pose significant challenges. However, the paper does not discuss these challenges in depth and instead directly describes the ablated designs for adding a temporal layer. A stronger paper would articulate the problem in terms of how the pretrained ControlNet lacks temporal reasoning, while the video base model possesses this ability. It would then analyze the challenge of merging per-frame controllability from ControlNet with the temporal reasoning of the base model. Your proposed solution—adding a simple temporal layer to ControlNet and merging with the base model through addition—might serve as a reasonable baseline, but it feels underexplored. For example, in your ablations, it would have been insightful to explore not just different temporal layers in the ControlNet but also merging ControlNet and the base model. Also, you could analyze how the simple addition of temporal layer changes the visualization of latents. This gives readers confidence that your solution is very reasonable.
>
>
> *(2) From SD15 to DiT:*
>
> Adapting from SD15 to DiT also presents significant challenges. However, the paper again bypasses the problem description and jumps to a straightforward solution: adapting only Block A due to its large feature size. While this might work as a baseline, it feels suboptimal as it discards most of the ControlNet blocks. If you believe this approach is indeed ideal, the paper should include a deeper analysis to substantiate this claim. For example, conducting an attention map analysis or exploring other strategies (e.g., upsampling features from other blocks, merging them, and injecting them into the base model) could strengthen the contribution. These aspects are currently missing from the paper.
>
> In summary, the paper lacks a clear articulation of the technical challenges and sufficient exploration or analysis how your proposals address chanllenges. While the motivation is clear and valuable, the focus on engineering solutions without deeper analysis or justification leaves the work feeling incomplete. I encourage the authors to revisit these aspects to make the contributions more compelling and impactful.

---

> > ### Comment · Reviewer_pa78 · 2024-11-25
> > **Questions to Multi-Control training**
> >
> > I am new to multi-control. You said you train multiple controls in a round-robin fashion, and only trains one control at one time. In this case, how your patch-level Q-former is learned to handle multiple controls in inference? The weights for each ControlNet are independent?

---

> ### Author Response · Authors · 2024-11-25
>
> Dear Reviewer pa78,
>
> Thanks for your additional question, and sorry for the confusion in the earlier response, where we thought you were referring to the unified Ctrl-Adapter setup (in Table 4), as you mentioned 'six' control conditions. Below we clarify the settings.
>
> - In the Multi Ctrl-Adapter setup with 7 control conditions (in Sec. 2.3 & Table 3), in each training step, we first sampling K uniformly from {1,2,3,4} and sampling K control conditions out of 7 total video control conditions (code implementation is in our supplementary .zip file). We added this detail in the red color in Sec. 2.3 of the newly revised PDF.
> - In the unified Ctrl-Adapter setup (in Table 4), in each training step, we select one control condition out of 6 total image control conditions in a round-robin fashion.
>
> Please feel free to let us know if there are any questions that are unclear to you, and we will try our best to address them. Again, thank you for your detailed feedback and valuable questions!

---

> ### Comment · Reviewer_pa78 · 2024-11-26
> **Response to Authors**
>
> Thanks. Now I understand the multi adapter.
>
> How about the deeper analysis to technical challenges, motivations of your solutions, and justify why your solution is non-trival, and why your design will be better than others?
>
> For example, for the challenge of extending to different base models, is different feature size the only issue to tackle? For the current solution, seems using only block A when adapting image controlnet to video generation is suboptimal? Is it good enough? Why not using more blocks?
>
> For the challenges of extending image to video, is extra temporal dimension the only issue to tackle? Is adding temporal alone in adapter, without learnable interaction between adapter and base model good enough? Why not learnable interaction between features from main branch and features from adapter?
>
> Overall, I am happy to raise my score if you can dig deeper into technical challenges and analyze more into your contributions. I would like to see more analysis in the big challenges you met to extend X-Adapters beyond base model and modality, push your simple extension of existing work to the highest quality as you can.

---

> > ### Author Response · Authors · 2024-11-26
> > **Author Response to Reviewer pa78 (Part 1/2)**
> >
> > Dear Reviewer pa78,
> >
> > Really appreciate your detailed response!
> >
> > ### **Key technical challenges and the novel solutions proposed**
> >
> > Firstly, thanks for your rewriting suggestions. We added bullet points at the end of our introduction section in the newly updated PDF to highlight the key technical challenges and the novel solutions proposed. We also summarize here for the ease of your reading:
> >
> > **Technical contribution:**
> > - We are the first to propose fine-grained patch-level MoE router for ControlNet composition via Multi-condition control
> > - Adaptation to UNet-based and DiT-based image/video backbones (via our novel adapter architecture design)
> > - We are the first to discuss and provide solution for the adaptation to diffusion models with continuous timestep samplers (via inverse noise sampling algorithm)
> > - We are the first to discuss and provide solution for adaptation to different noise scales (via latent-skipping strategy)
> >
> > **Performance, efficiency, and flexibility:**
> > - Ctrl-Adapter matches the performance of pretraining ControlNets for both image and video control, with significantly better efficiency
> > - Ctrl-Adapter can be applied to a variety of downstream tasks including video editing, video style transfer, and text-guided object motion control
> >
> > ### **1. Motivation of our work, and further explanation of our difference with X-Adapter**
> >
> > Here we would like to provide some more explanations and pointers to your concerns, which we hope could better help you grasp the technical contributions of our work, and the several key novelties that distinguish our work from X-Adapter.
> >
> > **Motivation**
> >
> > Our motivation is to design a _novel, flexible framework that enables the efficient reuse of pretrained ControlNets for diverse controls with any new image/video diffusion models, by adapting pretrained ControlNets and improving temporal alignment for videos._
> >
> > We would like to kindly highlight a few key points that make our paper well-distinguished from X-Adapter:
> >
> > - Ctrl-Adapter architecture is novel, and is not only more performant than X-Adapter (see Table 2 for comparison), but also more efficient for both training and inference  (as illustrated in W4 & Q1 under “Author Response to Reviewer pa78 (Part 3/3)”
> > - X-Adapter is only for image control, while Ctrl-Adapter is designed to work for both images and videos.
> > - As mentioned above, the major technical contributions of patch-level MoE router for ControlNet composition, inverse noise sampling for adaptation to continuous timestep samplers, as well as latent-skipping strategy for adaptation to different noise scales are not discussed in X-Adapter.

---

> > ### Author Response · Authors · 2024-11-26
> > **Author Response to Reviewer pa78 (Part 2/2)**
> >
> > ### **2. Technical Challenges**
> >
> > > “The paper seems to skip over the challenges and instead jumps directly into engineering solutions like “skip latents” and “inverse noise scheduler.”
> >
> > We would like to kindly remind the reviewer that we have already discussed the challenges of these two points before providing solutions in our PDF. Specifically,
> >
> > - In L221-L230, we mentioned intuition for skipping-latents: “some recent diffusion models sample noisier of much bigger scale….We find that adding larger-scale $z_t$ from the new backbone models to image conditions $c_f$ dilutes the $c_f$ and makes the ControlNet outputs less informative, whereas skipping $z_t$ enables the adaptation of such new backbone models”, as well as “when the image conditions are provided only for the subset of video frames, ControlNet could rely on the information from zt and ignore $c_f$ during training“.
> >
> > - In L232-L243, we mentioned intuition for inverse noise sampling: “some recent diffusion models sample timesteps from continuous distributions…. This gap between discrete and continuous distributions means that we cannot assign the same timestep to both the video diffusion model and the ControlNet”.
> >
> >
> > **(1) From Image to Video**
> >
> >
> > >“How the pretrained ControlNet lacks temporal reasoning?”
> >
> > Thank you very much for this comment. We have addressed this question quantitatively through the following experiments in our paper:
> >
> > - The Hotshot-XL + SDXL ControlNet baseline achieves much worse visual quality and spatial control scores because it's applying image ControlNet frame-by-frame directly to the video generation backbone without any temporal modeling (see Table 1).
> > - Ctrl-Adapter with temporal layers achieve better spatial control (measured by optical flow error) than variants without temporal layers on video generation model (see Table 6).
> >
> > >“ It would have been insightful to explore not just different temporal layers in the ControlNet but also merging ControlNet and the base model?”
> >
> > We have addressed this question quantitatively by comparing our adapter-based strategy with LoRA-based strategy (which is equivalent to merging ControlNet and the base model) in Appendix Sec. F.3 (see Table 11).
> >
> > Intuitively, merging ControlNet and the base model (like LoRA) requires backpropagation through the ControlNet. While LoRA-style parameter-efficient finetuning can reduce the number of parameters to learn for a new control condition (so that disk storage for saving checkpoints will be reduced), it does not save much GPU memory during training. In contrast, our Ctrl-Adapter approach can keep the whole ControlNet frozen during training without passing any gradients. In addition, for the two scenarios that latent-skipping works better (e.g., different noise scales & sparse frame conditions, see Appendix Sec. E.3 for details), since no latents are input to the ControlNet, the users are further able to pre-extract the ControlNet features, entirely skipping the loading ControlNet on memory during Ctrl-Adapter training.
> >
> >
> >
> > **(2) From SD15 to DiT**
> >
> > Thanks for your valuable feedback! We would like to kindly emphasize that we indeed already covered attention map analysis in Appendix Sec. G.1 (see Fig. 22), and Ablation of adapting different blocks (including composition of multiple blocks) in Appendix Sec. E.2 (see Fig. 16, Table 7 and Table 8).
> >
> >
> >
> > ---
> > Finally, please let us express again our sincere gratitude for your detailed response and valuable feedback. These suggestions are indeed very valuable to improve our manuscript. We will follow your paper writing suggestions, and highlight the key contributions of our paper to make it stronger in the camera-ready version.
> >
> >
> > In addition, since the discussion period deadline has been extended, we will try our best to provide some extra quantitative experiments that delve deeper into our architecture design choice. Thank you again!

---

> ### Author Response · Authors · 2024-12-01
>
> Dear Reviewer pa78,
>
> Hope you had a wonderful Thanksgiving holiday week! Thank you very much for updating your evaluation scores after our previous responses. We hope our detailed responses have adequately addressed your concerns. We sincerely appreciate the time and effort you devoted to reviewing our paper, as well as your valuable comments and suggestions.
>
> ---
> This week, we spent some additional effort conducting new experiments that we would like to share. We believe that these new experiments can further support our design choice of passing block A in ControlNet to Ctrl-Adapter. The experiment below was conducted under the same settings as Table 8 in our revised PDF. The rows in the table are sorted in descending order of SSIM scores for better readability.
>
>
> ||||
> |-|-|-|
> |  | SSIM | FID |
> | block A | 0.6802 | 17.90 |
> | block A + B | 0.6812 | 17.52 |
> | block B | 0.6712 | 18.23 |
> | block A + B + C | 0.6711 | 18.09 |
> | block A + B + C + D | 0.6478 | 26.27 |
> | block B + C | 0.6373 | 18.23 |
> | block B + C + D | 0.6008 | 22.84 |
> | block C | 0.5273 | 22.22 |
> | block D | 0.3506 | 34.16 |
>
> As we can see, incorporating block A is the most beneficial for overall spatial control (SSIM) and visual quality (FID). Additionally, we observed that combining block A with block B only slightly increases SSIM and FID scores (only increase SSIM from 0.6802 to 0.6812, and decrease FID from 17.90 to 17.52). We believe this increase is marginal compared to the additional computational cost required. Furthermore, due to the small feature map size, the information contained in blocks C and D is significantly less effective compared to blocks A and B. Therefore, using block A remains a strong choice, offering good performance and lower computational cost compared to combining multiple blocks.
>
> ---
> Finally, we would like to reiterate our sincere gratitude for your detailed comments and valuable feedback. Your suggestions are incredibly helpful in improving our manuscript. We will follow your recommendations for refining the paper's key contributions, adding extra experiments from this rebuttal, and strengthening it further in the camera-ready version.

---

### Official Review · Reviewer_QcBv · 2024-11-03

**Soundness:** 3
**Presentation:** 3
**Contribution:** 3
**Rating:** 6
**Confidence:** 4

**Summary:**

CTRL-Adapter is an efficient and versatile framework designed to integrate diverse controls into image and video diffusion models by adapting pretrained ControlNets. This method achieves comparable performance to training ControlNet from scratch while significantly reducing training costs. It introduces fine-grained, patch-level MoE (Mixture of Experts) routing to compose ControlNet features more effectively, as opposed to prior methods that operate only at the image or frame level. CTRL-Adapter is compatible with both UNet-based and DiT-based backbones, continuous timestep samplers, and varying noise scales. Extensive experiments demonstrate that CTRL-Adapter performs well on image and video generation and can be applied to various tasks, including video editing, style transfer, and text-guided object motion control.

**Strengths:**

1. The proposed method stands out for its simplicity and efficiency. It presents a cost-effective approach to adapting image-based ControlNet models for video-based applications. It also offers valuable insights into merging multiple conditions using three strategic choices. It incorporates a fine-grained, patch-level Mixture of Experts (MoE) routing to enhance the generation process by enabling precise and adaptive control over features.

2. The evaluation is thorough and well-executed, providing extensive comparisons across both single-condition and multi-condition video generation. Additionally, it demonstrates the model's versatility by showcasing its performance on tasks such as video editing, style transfer, and text-guided object motion control, highlighting its wide applicability.

3. The paper is exceptionally well-written, with a logical and coherent structure that ensures ease of understanding.

**Weaknesses:**

1. The comparison of training costs might not be entirely fair. Since the CTRL-Adapter leverages a pre-trained image ControlNet and then adapts it for video tasks, it benefits from the substantial pre-training cost in the image control task. Given that the image ControlNet is already proficient at extracting information from various conditions, this adaptation should inherently be faster. Therefore, when evaluating training efficiency, it is crucial to account for the advantages gained from the pre-trained ControlNet.

2. The explanation of how multiple conditions are integrated in the proposed method could be clearer and more detailed. The paper briefly outlines three possible integration strategies in a single paragraph, accompanied by a figure, which might be insufficient for comprehending the intricacies involved. This section deserves more space and elaboration in the main text rather than being relegated to the appendix.

**Questions:**

1. What is the additional inference cost associated with using multiple condition ControlNets along with their corresponding control adapters, for example, when handling seven different conditions in Table 3? While the paper mentions significant reductions in training costs by focusing on adapting image-based ControlNets for video tasks, how does the increase in inference complexity, due to maintaining multiple controls, impact overall efficiency and performance?

2. Would integrating control adapters directly inside each ControlNet simplify the adaptation process and make it more efficient, like a fine-tuning task from image to video ControlNet? Could this approach be a viable and efficient baseline, particularly for single-condition scenarios?

---

> ### Author Response · Authors · 2024-11-21
> **Author Response to Reviewer QcBv (Part 1/3)**
>
> We would like to sincerely thank the Reviewer for the insightful and detailed comments.
>
>
> > ### W1: The comparison of training costs might not be entirely fair. Since the CTRL-Adapter leverages a pre-trained image ControlNet and then adapts it for video tasks, it benefits from the substantial pre-training cost in the image control task. Given that the image ControlNet is already proficient at extracting information from various conditions, this adaptation should inherently be faster. Therefore, when evaluating training efficiency, it is crucial to account for the advantages gained from the pre-trained ControlNet.
>
> Thank you very much for the comment. As you mentioned, it is true that __image ControlNet is already proficient at extracting information from various conditions, this adaptation should inherently be faster__.  We would like to kindly bring your attention to first three paragraphs of Sec 1, where we explain our main motivation __“we design Ctrl-Adapter, a novel, flexible framework that enables the **efficient reuse** of pretrained ControlNets for diverse controls with any new image/video diffusion models, by adapting pretrained ControlNets (and improving temporal alignment for videos).”__
>
> In many transfer learning scenarios where one adapts a foundation model to a downstream task, we do not take into account the training cost of the foundation model. Since Ctrl-Adapter framework is about efficient transfer learning of ControlNet, instead of training ControlNet from scratch, we would like to respectfully disagree that the training cost of image ControlNet should be part of Ctrl-Adapter training cost.
>
> In addition, as we can see in Fig. 2 left, I2VGen-XL + Ctrl-Adapter outperforms all previous baselines (e.g., Video Composer, ControlVideo, and Control-A-Video) for controllable video generation with less than 10 hours of training. As far as we know, these baseline models don't report their computational cost explicitly in their paper or GitHub repo. For example, Video Composer mentioned in one of their [github issues](https://github.com/ali-vilab/videocomposer/issues/6) that _“We recommend using at least 8/16 A100-80G GPUs to fine-tune our model, and the training time depends on the specific task which may take a week or more.”_, so a reasonable guess of their computational cost would be more than 8$\sim$16 GPUs * 24 hours * 7 days, which is more than 1344~2688 GPU hours. In contrast, the official SD1.5 image ControlNet we used in our paper is trained in 500 GPU hours (see hugging face page (here)[https://huggingface.co/lllyasviel/sd-controlnet-depth]). Therefore, even if we take the GPU hours of pretrain an image ControlNet into consideration, our method is still more efficient than the baseline models.

---

> > ### Author Response · Authors · 2024-12-01
> >
> > Dear Reviewer QcBv,
> >
> > Hope you had a wonderful Thanksgiving holiday week! We would like to sincerely thank you once again for your detailed feedback and apologize for taking up your time. We believe that we have addressed all your questions, particularly: (1) clarification of the training cost of Ctrl-Adapter, with GPU hours comparison with VideoComposer, (2) restructure of the MoE Ctrl-Adapter section with more implementation details (see: updated PDF), (3) quantitative results of the inference cost of multiple conditions control, and (4) comparison of computational cost of LoRA-style and adapter-style ControlNet integration strategies.
> >
> > As the deadline for the extended discussion period ends tomorrow, we would greatly appreciate your response soon. If our rebuttal adequately addresses your questions, we kindly request an update to your evaluations. Once again, thank you very much for your reviews and feedback!
> >
> > Yours sincerely,
> >
> > The Authors

---

> ### Author Response · Authors · 2024-11-21
> **Author Response to Reviewer QcBv (Part 2/3)**
>
> > ### W2: The explanation of how multiple conditions are integrated in the proposed method could be clearer and more detailed. The paper briefly outlines three possible integration strategies in a single paragraph, accompanied by a figure, which might be insufficient for comprehending the intricacies involved. This section deserves more space and elaboration in the main text rather than being relegated to the appendix.
>
>
> Thank you very much for your feedback. Following the suggestion, we have added more details about the MoE router details in Sec. 2.3 **in the updated PDF**. Additionally, we have moved the qualitative experiment results demonstrating adaptation to different downstream tasks from the main paper, and instead put the implementation details in the appendix.
>
> We sincerely thank the reviewer for understanding the challenge of presenting extensive content within the limited space constraints of this paper. We kindly draw your attention to Appendix B and Appendix E, where we provide comprehensive descriptions and analyses of the method details, including the adapter architectures and MoE method.
>
> Please let us know if there is any content or design aspect that remains unclear. We will do our best to address your concerns and provide detailed illustrations as needed.
>
>
> > ### Q1: What is the additional inference cost associated with using multiple condition ControlNets along with their corresponding control adapters, for example, when handling seven different conditions in Table 3? While the paper mentions significant reductions in training costs by focusing on adapting image-based ControlNets for video tasks, how does the increase in inference complexity, due to maintaining multiple controls, impact overall efficiency and performance?
>
>
> Thank you very much for this excellent comment. We break down the components to run with their corresponding inference time when using Ctrl-Adapter with 7 conditions (e.g., depth, canny, softedge, segmentation, normal, lineart, openpose) mentioned in Table 3 as follows. Specifically, we report the second per component for each inference denoising step:
> - (a) SDv1.5 ControlNet of 7 different conditions - (0.081 sec per condition)
> - (b) MoE router - (0.096 sec)
> - (c) Ctrl-Adapter - (0.146 sec)
> - (d) Target backbone model - (0.402 sec for I2VGen-XL backbone)
>
> Since you can run (a) 7 ControlNets in parallel in practice, compared to using a single condition, (0.081 + 0.146 + 0.402 = 0.629), using multiple conditions (0.081 + 0.096 + 0.146 + 0.402 = 0.725) takes only the **0.096** sec (**15.2%** increase) additional inference time, while increasing the video control accuracy (**3.20** for single depth condition -> **2.08** for all 7 conditions in Optical Flow Error metric, as reported in Table 1 and Table 3). Users can choose whether to use multiple conditions due to their hardware constraints.

---

> ### Author Response · Authors · 2024-11-21
> **Author Response to Reviewer QcBv (Part 3/3)**
>
> > ### Q2: Would integrating control adapters directly inside each ControlNet simplify the adaptation process and make it more efficient, like a fine-tuning task from image to video ControlNet? Could this approach be a viable and efficient baseline, particularly for single-condition scenarios?
>
> Thank you very much for this excellent question! This idea has been recently explored in several diffusion model community projects, such as [stabilityai/control-lora](https://huggingface.co/stabilityai/control-lora), where they use LoRA finetuning of existing ControlNet.
>
> Note that **finetuning ControlNet requires backpropagation through the ControlNet**. While parameter-efficient finetuning can reduce the number of parameters to learn for a new control condition (so that disk storage for saving checkpoints will be reduced), **it does not save much GPU memory during training.** In contrast, our Ctrl-Adapter approach can keep the whole ControlNet frozen during training without passing any gradients. In addition, for the two scenarios that latent-skipping works better (e.g., different noise scales & sparse frame
> conditions, see Appendix Sec. E.3 for details), since no latents are input to the ControlNet, the users are further able to pre-extract the ControlNet features, entirely skipping the loading ControlNet on memory during Ctrl-Adapter training.
>
>
> Once more, we sincerely thank you for all the comments and very useful feedback. We think that we have addressed all the questions in depth. If the reviewer has any additional questions, please let us know, and we will be more than happy to answer them.

---

> ### Author Response · Authors · 2024-11-25
> **Official Comment by Authors**
>
> Dear Reviewer QcBv,
>
> We would like to once more thank you for your effort providing feedback. We believe that we have addressed all Reviewer's questions, in particular: (1) clarification of the training cost of Ctrl-Adapter, with GPU hours comparison with VideoComposer, (2) restructure of the MoE Ctrl-Adapter section with more implementation details (see: updated PDF), (3) quantitative results of the inference cost of multiple conditions control, and (4) comparison of computational cost of LoRA-style and adapter-style ControlNet integration strategies.
>
> Since the end of the discussion period is approaching, we would like to sincerely ask the Reviewer to comment on the rebuttal and update the score accordingly. If the Reviewer has any additional questions, please let us know so that we can address them before the end of the discussion period.
>
> Once more, thank you very much for your reviews and feedback!
>
> Yours sincerely,
>
> The Authors

---

### Official Review · Reviewer_iSfd · 2024-11-04

**Soundness:** 3
**Presentation:** 4
**Contribution:** 3
**Rating:** 6
**Confidence:** 3

**Summary:**

This paper introduces a new framework called CTRL-Adapter, aimed at enhancing the control capabilities of image and video diffusion models. By adapting pretrained ControlNets, CTRL-Adapter allows for the efficient integration of various control conditions into new diffusion models, supporting image control, video control, and sparse frame control. The results demonstrate that CTRL-Adapter maintains high performance on various datasets with significantly reduced computational costs compared to training ControlNets from scratch.

**Strengths:**

+ The presentation and writing quality of this manuscript are are generally strong and well-structured.
+ The framework supports a wide range of applications, from image and video control to video editing and style transfer.
+ CTRL-Adapter requires fewer computational resources. This method significantly reduces the training time, outperforming baselines in less than 10 GPU hours
+ The authors provide comprehensive experiments that validate the performance of CTRL-Adapter.

**Weaknesses:**

+ While skipping latents can increase efficiency, it may inadvertently lead to loss of fine-grained detail in frames without direct conditioning. I'm wondering if there are any issues with this method that I'm concerned about?
+ Providing more detailed results from an ablation study about Weaknesses 1 could offer valuable insights into the impact of this strategy. It could reveal conditions where skipping latents is beneficial versus scenarios where it may compromise quality.
+ Whether skipping latents reduces the ability of the CTRL-Adapter to generalize between certain control types.
+ In my experience, the most significant advantage of this work is the value of a wide range of applications. In terms of technical contributions, it seems to extend multiple adapters in various generative models. Could the authors provide more elaboration on the technical contribution or insights?

**Questions:**

Please see weaknesses

---

> ### Author Response · Authors · 2024-11-21
> **Author Response to Reviewer iSfd (Part 1/3)**
>
> We would like to sincerely thank the Reviewer for the insightful and detailed comments.
>
>
> > ### W1 & W2: While skipping latents can increase efficiency, it may inadvertently lead to loss of fine-grained detail in frames without direct conditioning. I'm wondering if there are any issues with this method that I'm concerned about? Providing more detailed results from an ablation study about Weaknesses 1 could offer valuable insights into the impact of this strategy. It could reveal conditions where skipping latents is beneficial versus scenarios where it may compromise quality.
>
> Great question! **We would like to kindly clarify that we suggest skipping latent $z_t$ only for the following two specific scenarios, and not skipping the latent for most use cases.**
>
> - **We suggest skipping latent $z_t$ for**:
>   - (1) adaption to the backbone with noise scales different from SDv1.5, such as SVD
>   - (2) video control with sparse frame conditions, as described in the introduction (L83-85) and Appendix E.3.
> In Table 10, we show that skipping $z_t$ improves both FID and optical flow scores in these scenarios.
>
> - As you mentioned, **skipping latents $z_t$ can hurt performance**, in the common ControlNet usage scenarios (i.e., not using sparse frame conditions + not using Ctrl-Adapter), as reported by the ControlNet author [here](https://github.com/lllyasviel/ControlNet/discussions/216). The reason behind this is likely because the fine-grained details are lost, as you mentioned.

---

> ### Author Response · Authors · 2024-11-21
> **Author Response to Reviewer iSfd (Part 2/3)**
>
> > ### W3: Whether skipping latents reduces the ability of the CTRL-Adapter to generalize between certain control types.
>
> Thanks for asking this Interesting question! We conducted experiments that used the I2VGen-XL + Ctrl-Adapter trained on depth condition **without skipping latents**, and zero-shot inference on depth, canny edge, and softedge **with and without skipping latents**. Following our main paper, we report the Optical Flow Error to measure the spatial control quality. As we can see from the table below, skipping latents indeed increases the Optical Flow Error when generalizing from depth to canny edge (**18%** increase in Optical Flow Error), and generalizing from depth to softedge (**14%** increase in Optical Flow Error), which is as expected since skipping latents result in some information lost **under this default setting**. On the other hand, we would like to kindly emphasize that as we answered in the previous question, we show quantitatively in Appendix Table 10 that skipping latent helps **only for the two specific scenarios**, (1) adaptation to different noise scales and (2) sparse control, which is not contradictory to the result in the table below.
>
> Please let us know if there is any content or design aspect that remains unclear. We will do our best to address your concerns and provide detailed illustrations as needed.
>
>
> |||||
> |-|-|-|-|
> | Training Condition | Depth | Depth | Depth |
> | Inference Condition | Depth | Canny | Softedge |
> | Optical Flow Error (without skipping latents) | 3.20 | 3.10 | 2.94 |
> | Optical Flow Error (with skipping latents) | 3.36 | 3.68 | 3.38 |
>
>
> > ### W4: In my experience, the most significant advantage of this work is the value of a wide range of applications. In terms of technical contributions, it seems to extend multiple adapters in various generative models. Could the authors provide more elaboration on the technical contribution or insights?
>
> Thanks for appreciating the advantages of Ctrl-Adapter! Designing a versatile adapter that can work for different applications is indeed the major motivation for this project. Below we summarize our technical contributions:
>
> - **Good performance with high training efficiency.** Training Ctrl-Adapter matches the performance of pretrained ControlNets with significantly lower training costs.
>
> - **Multi-condition control via MoE routers.** To the best of our knowledge, we are the first to propose fine-grained patch-level MoE composition for ControlNet features, while previous works (Multi-ControlNet, UniControl, Uni-ControlNet, VideoComposer) fuse the features of different control conditions only at the image/frame level.
>
> - **Inverse timestep sampling to support backbones with continuous noise schedulers.** With the gradual adoption of continuous noise schedulers in the image/video generation community (e.g., DiT, SD3, etc.), reusing the many existing plug-ins developed for traditional discrete noise schedulers has become an important problem. We are the first to provide a decent solution for continuous/discrete noise scheduler matching through algorithmic design.
>
> - **Skip latents for sparse control and large noise scale.** Training diffusion models with significantly large noise scales is becoming a trend (e.g., SVD). We show empirically that skipping latents to ControlNet is a useful solution to ensure good generation quality.
>
> - **Versatility.** Through extensive experiments, we show that Ctrl-Adapter supports both image and video generation backbones, both UNet-based and DiT-based backbones, and multiple widely-adopted models such as SDXL, PixArt-$\alpha$, SVD, I2VGen-XL, HotshotXL, and Latte. In addition, Ctrl-Adapter can be flexibly applied to various downstream tasks beyond spatial control, including video editing, video style transfer, and text-guided object motion control.

---

> ### Author Response · Authors · 2024-11-21
> **Author Response to Reviewer iSfd (Part 3/3)**
>
> ### **Additional Insights**
>
> In addition to the technical contributions, we would also like to provide our insights about Ctrl-Adapter in the following.
>
>
>
> **Do we need to train individual task-specific adapters for each control condition, or just train a unified multi-task adapter?**
>
> - In Sec. 4.5 and Sec. 4.7, we show that Ctrl-adapter shares good zero-shot generalization ability across different control conditions, and thus enables us to train a unified multi-task adapter without an obvious drop in performance.
>
> **Can we make some architecture/technical improvements over previous adapter works?**
>
> - In related works (Sec. 6), we discussed that X-Adapter [A] needs to be used with the source image diffusion model (SDv1.5) during both training and inference, whereas Ctrl-Adapter does not require to load SDv1.5 at all, making it more memory and computationally efficient.
>
> **For different backbones (UNet v.s. DiT), should the corresponding blocks map from the same or different blocks from ConrtrolNet?**
>
> - As illustrated in Appendix G.1. “Visualization of Spatial Feature Maps”, we provide insights that U-Net blocks in Controlnet demonstrate a coarse-to-fine pattern as the feature map size increases, while such a pattern is not observed in DiT models. This motivated us to study different mappings from ControlNet layers to target diffusion models’ layers (as detailed in Appendix Sec. E.1.2 and Appendix Sec. E.2).
>
>
> [A] Ran, Lingmin, et al. "X-adapter: Adding universal compatibility of plugins for upgraded diffusion model", CVPR 2024.
>
> Once more, we sincerely thank you for all the comments and very useful feedback. We think that we have addressed all the questions in depth. If the reviewer has any additional questions, please let us know, and we will be more than happy to answer them.

---

> ### Author Response · Authors · 2024-11-25
>
> Dear Reviewer iSfd,
>
> We would like to once more sincerely thank your effort to provide detailed feedback. It would be great if you could check our previous responses and help revisit your score.
>
> In addition to the experiments in our previous reply showing whether skipping latents reduces the generalizability of Ctrl-Adapter to other control types on controllable **video** generation, we spent effort conducting a new extra experiment for skipping latents with **image** generation baseline (i.e., SDXL + Ctrl-Adapter). Specifically, we use the model trained solely on depth condition (without skipping latents), and evaluate its generalization ability to other conditions under both with and without skipping latents. We  report FID and SSIM as evaluation metrics for visual quality and spatial control, and add evaluation scores with images generated with pure SDXL without ControlNet as a “lower bound” baseline.
>
> As we can see from the tables below, results with SDXL+Ctrl-Adapter trained on depth and applied to other conditions **with** skipping latents achieve slightly worse performance compared to SDXL+Ctrl-Adapter trained on depth and applied to other conditions **without** skipping latents, but are much better than pure SDXL without ControlNet. This proves that skipping latents have minimal impact on the performance of Ctrl-Adapter.
>
> **FID**
>
> ||||||||
> |-|-|-|-|-|-|-|
> |  | Inference with Depth | Zero-Shot Inference with Canny | Zero-Shot Inference with Softedge |  Zero-Shot Inference with Lineart | Zero-Shot Inference with Segmentation |  Zero-Shot Inference with Normal |
> | Trained Solely on Depth without skipping latents | **14.87** | **16.17** | 15.79 | **15.32** | **16.49** | 15.76 |
> | Trained Solely on Depth with skipping latents     | 15.81 | 16.39 | **15.54** | 15.72 |  16.73 | **15.66** |
> | Plan SDXL without ControlNet | 17.49 | 17.49 | 17.49 | 17.49 | 17.49 | 17.49 |
>
>
>
> **SSIM**
>
> ||||||||
> |-|-|-|-|-|-|-|
> |  | Inference with Depth | Zero-Shot Inference with Canny | Zero-Shot Inference with Softedge |  Zero-Shot Inference with Lineart | Zero-Shot Inference with Segmentation |  Zero-Shot Inference with Normal |
> | Trained Solely on Depth without skipping latents | **0.8398** | **0.3685** | **0.3822** | 0.4128 | **0.5623** | **0.7716** |
> | Trained Solely on Depth with skipping latents | 0.8235 | 0.3429 | 0.3458 | **0.4177** |  0.5547 | 0.7684 |
> | Plan SDXL without ControlNet | 0.7079 | 0.2916 | 0.2831 | 0.3241 | 0.4624 | 0.6306 |
>
> We believe that we have addressed all Reviewer's questions, in particular: (1) clarification of the scenarios when skipping latent could be beneficial and harmful with pointer to quantitative results (see: Table 10), (2) elaboration of technical contributions and insights of Ctrl-Adapter, and (3) new experiments showing whether skipping latents reduces the generalizability of Ctrl-Adapter to other control types on controllable **video** generation.
>
> Since the end of the discussion period is approaching, we would like to sincerely ask the Reviewer to comment on the rebuttal. If the Reviewer has any additional questions, please let us know so that we can address them before the end of the discussion period.
>
> Once more, thank you very much for your reviews and feedback!
>
> Yours sincerely,
>
> The Authors

---

> ### Author Response · Authors · 2024-12-01
>
> Dear Reviewer iSfd,
>
> Hope you had a wonderful Thanksgiving holiday week! We would like to sincerely thank you once again for your detailed feedback and apologize for taking up your time. We believe that we have addressed all your questions, particularly: (1) clarification of the scenarios when skipping latent could be beneficial and harmful with pointer to quantitative results (see: Table 10), (2) elaboration of technical contributions and insights of Ctrl-Adapter, and (3) new experiments showing whether skipping latents reduces the generalizability of Ctrl-Adapter to other control types with both image and video generation backbones.
>
> As the deadline for the extended discussion period ends tomorrow, we would greatly appreciate your response soon. If our rebuttal adequately addresses your questions, we kindly request an update to your evaluations. Once again, thank you very much for your reviews and feedback!
>
> Yours sincerely,
>
> The Authors

---

> > ### Comment · Reviewer_iSfd · 2024-12-02
> >
> > I thank the authors for their detailed responses and recommend this work for acceptance.

---

### Official Review · Reviewer_6nvp · 2024-11-04

**Soundness:** 4
**Presentation:** 2
**Contribution:** 4
**Rating:** 8
**Confidence:** 3

**Summary:**

The paper proposes an efficient method for adapting pretrained ControlNets to various image and video diffusion frameworks. The authors also proposed a mixture of expert method for multi conditioning, and a mapping between discrete and continuous diffusion timesteps. Overall, the paper address an important problem (adapting pre-trained ControlNets efficiently), provide useful insights and achieves superior results.

**Strengths:**

- The results proposed in the paper are interesting and convincing.
- Adapting pre-trained ControlNets to new diffusion frameworks including videos is interesting and the method is novel.
- The paper is well written and the experiments to validate the proposed method are significant.
- The code is provided.

**Weaknesses:**

- Poor presentation: the main paper is packed with experiments but lacks many essential details about the method such as the adapter architecture, more details about the MoE method, etc.
- Unjustified architecture choices: the authors proposed to use a 1-1 mapping between the ControlNet layer to ones with similar dimension in the new diffusion model. This assumes that these layers learn similar features but the authors did not touch on this topic or justified their choices.
- What is the rational behind conditioning the CTRL-Adapter with the first frame? How does the proposed method perform without such conditioning?

**Questions:**

Additional comments (no need to address):
- typo in in L069 "Controlnet a cannot"

---

> ### Author Response · Authors · 2024-11-21
> **Author Response to Reviewer 6nvp**
>
> We would like to sincerely thank the Reviewer for the insightful and detailed comments.
>
> > ### W1: Details about the method, such as the adapter architecture and the MoE method, etc.
>
> Thank you very much for your feedback. Following the suggestion, we have added more details about the **adapter architecture design** in Sec. 2.2 and the **MoE router details** in Sec. 2.3 **in the revised PDF**. Additionally, we have removed the qualitative experiment results demonstrating adaptation to different downstream tasks from the main paper, instead with the details illustrated in the appendix. We also kindly draw your attention to Appendix B and Appendix E, where we provide comprehensive descriptions and analyses of the method details, including the adapter architectures and MoE method.
>
> Please let us know if there is any content or design aspect that remains unclear. We will do our best to address your concerns and provide detailed illustrations as needed.
>
>
> > ### W2: the authors proposed to use a 1-1 mapping between the ControlNet layer to ones with similar dimensions in the new diffusion model. This assumes that these layers learn similar features but the authors did not touch on this topic or justified their choices.
>
> As you mentioned, it is indeed important to choose which layers of different target backbones we plug into our Ctrl-Adapter, because different architectures learn different representations across layers. Based on our analysis, we use the _1-1 mapping_ for diffusion backbones with U-Net architectures but not for backbones with DiT architectures. We would like to kindly explain more details as follows:
>
> **1. We found that different architectures learn different features.**
>
> - As illustrated in Appendix G.1. “Visualization of Spatial Feature Maps”, we find that U-Net blocks in Controlnet demonstrate a coarse-to-fine pattern as the feature map size increases, while such a pattern is not observed in DiT models. This motivated us to study different mappings from ControlNet layers to target diffusion models’ layers.
>
> **2. We analyzed various design choices for U-Net based backbones.**
>
> - As described in Appendix E.1.2 “Where to fuse Ctrl-Adapter outputs in backbone diffusion” and Appendix E.1.3 “Number of Ctrl-Adapters in each output block position”, we compared various design choices and reached our current design (i.e., “1-1 mapping”).
>
> **3. We analyzed various design choices for DiT backbones and found different adapter mapping designs.**
>
> - As described in Appendix E.2. “Adaptation to DiT-based Backbones,” we compared different combinations of ‘which ControlNet features to use’ and ‘where in DiT to fuse Ctrl-Adapter outputs.’ As summarized in Table 8, we found that inserting the ControlNet output block A (the last block) into DiT blocks in an interleaved manner gives the best performance. Similar mapping has been used in transformer encoder-decoder architectures in the NLP domain, where a decoder attends the last layer of encoder hidden states.
>
>
>
> > ### W3: What is the rationale behind conditioning the CTRL-Adapter with the first frame? How does the proposed method perform without such conditioning?
>
> Thank you for this question. We would like to kindly clarify that Ctrl-Adapter uses the first frame only for the following scenarios: (1) image-to-video generation (e.g., I2VGen-XL, SVD), and (2) video editing and text-guided motion control developed on I2VGen-XL backbone (see Appendix H.3).
>
> - For (1), in our main paper, we present Sec. 4.3, Sec. 4.4, and Sec. 4.5 with I2VGen-XL as the default backbone, which by its definition as an image-to-video generation model, needs the first frame as input condition. However, the results in these sections are also applicable to text-to-video backbone models that don't need image conditioning, such as Latte and Hotshot-XL. We utilized these two text-to-video models in multiple places in our paper, including Figs. 5, 25, 28, and 38.
>
> - In addition, for (2) video editing and text-based motion control (in Appendix H.3), where we extract the first frame to modify/mask, then use the modified/masked first frame to re-generate a video using an image-to-video model. The usage of modifying/masking only the first frame is under the assumption that the objects/background in the first frame will consistently appear in the subsequent frames; such video editing/inpainting based on first frame editing is widely used in many video generation papers that use image-to-video models, such as [A].
>
> [A] Singer et al., Video Editing via Factorized Diffusion Distillation, 2024
>
> Once more, we sincerely thank you for all the comments and useful feedback. We think that we have addressed all the questions in depth. If you have any additional questions, please let us know, and we will be more than happy to answer them.

---

### Author Response · Authors · 2024-11-21
**General comments**

We would like to sincerely thank all the reviewers for their valuable feedback and suggestions. We have provided responses to all of the reviewers' questions in the individual rebuttals.

Following the reviewers’ suggestions, we restructure the method section in the **revised PDF (updated content is colored blue)** with the following main changes, with all other content **unchanged** if not colored in blue:
- (1) more details about the **adapter architecture** in Sec. 2.2, and **mathematical formulation** of our training loss in Sec. 2.1.
- (2) expanded the paragraph “Flexible adaptation to different backbones and settings” into three new independent paragraphs (i.e., adaptation to DiT, skipping latents, and inverse timestep sampling) **to align with our key technical contributions** in Sec. 2.2.
- (3) More details on the architecture and intuition of our design choice for the three **MoE router** variants in Sec. 2.3.

Additionally, we have removed the qualitative experimental results demonstrating adaptation to different downstream tasks from the main paper, and instead put the implementation details in the appendix.

---

### Meta-Review · Area_Chair_Skft · 2024-12-19

**Metareview:**

The paper introduces the CTRL adapter that adapts the pre-trained ControlNets to the new backbones in image and video generation. It allows to use a mixture of ControlNets for conditioning on multiple control signals. The authors provide comprehensive experiments on multiple applications, such as video editing, style transfer, and text-guided object motion control. The paper addresses an important problem of incorporating controls into image/video generative models in the computationally-efficient manner. I recommend the paper for Accept (Oral).

**Additional Comments On Reviewer Discussion:**

The authors revised the description of the method details during the rebuttal and added extensive ablation studies for using different for training and testing, as well as design decisions like skipping latents.

---

### Decision · Program_Chairs · 2025-01-22

Accept (Oral)